# ADAPTIVE DOMAIN SHIFT IN DIFFUSION MODELS FOR CROSS-MODALITY IMAGE TRANSLATION

**Zihao Wang[1]**[*] **Yuzhou Chen[2], Shaogang Ren [3]**

[1] Laplace Lab at University of Tennessee
[2] University of California Riverside [3] University of Tennessee at Chattanooga
`zihao.wang@tennessee.edu`
`Project Page: https://laplace.center/CDTSDE/`

## ABSTRACT

Cross-modal image translation remains brittle and inefficient. Standard diffusion approaches often rely on a single, global linear transfer between domains. We find that this shortcut forces the sampler to traverse off-manifold, high-cost regions, inflating the correction burden and inviting semantic drift. We refer to this shared failure mode as fixed-schedule domain transfer. In this paper, we embed domain-shift dynamics directly into the generative process. Our model predicts a spatially varying mixing field at every reverse step and injects an explicit, target-consistent restoration term into the drift. This in-step guidance keeps large updates on-manifold and shifts the model's role from global alignment to local residual correction. We provide a continuous-time formulation with an exact solution form and derive a practical first-order sampler that preserves marginal consistency. Empirically, across translation tasks in medical imaging, remote sensing, and electroluminescence semantic mapping, our framework improves structural fidelity and semantic consistency while converging in fewer denoising steps. The source code is in `https://github.com/LaplaceLab/CDTSDE`.

## 1 INTRODUCTION

Cross-modal image translation seeks to transform an image from one sensing modality into another while preserving task-relevant semantics. This capability underpins multimodal analysis in medical imaging and remote sensing, where translating between, for example, MRI modal or between SAR (synthetic-aperture radar) and optical imaging enables downstream tools to operate across modalities and improves interpretability (Conte et al., 2021; Arslan et al., 2025; Shin et al., 2021; Wang et al., 2019; Toriya et al., 2019; Luo et al., 2024; Wei et al., 2025; Aydin et al., 2025; Wang et al., 2024c). Yet, substantial differences in texture statistics, intensity distributions, and modality-specific structures make semantic alignment highly nonlinear and spatially heterogeneous, which renders simple domain-transition assumptions brittle.

Diffusion-based methods have recently advanced image translation with improved stability and sample fidelity compared to GANs, but efficiency and semantic faithfulness under highly nonlinear domain shifts like cross-modal translation remain challenging (Wang et al., 2024b; Cui et al., 2025; Lin et al., 2025; Wu et al., 2023). Existing acceleration tactics usually act outside the true translation dynamics: some precondition the initial state (Wu et al., 2023; Wang et al., 2024b)), others swap the numerical solver to higher-order ODE solvers for 15–20 steps (Lu et al., 2022), some inject ad-hoc guidance each step to stabilize larger step sizes (Lin et al., 2025), and still others distill long trajectories into few steps via teacher–student schemes (Sauer et al., 2024; Wang et al., 2024b). While useful, these tactics either decouple the domain-shift physics from the generative dynamics or compress it into surrogates, and on genuinely nonlinear cross-modal mappings they can still incur semantic drift or instability when taking large steps (Wang et al., 2024b).

In this work, we introduce a Cross-Domain Translation SDE (CDTSDE) that directly embeds domain-shift forces into every reverse-time update of the diffusion process. Concretely, the forward

---

[*]Corresponding author: zihao.wang@ieee.org

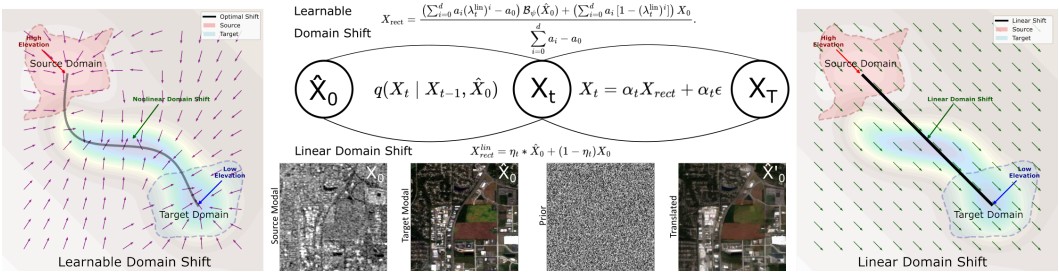

Figure 1: **fixed vs. geometry-aware domain mixture comparison on manifold space** *Left*: The proposed spatially varying schedule $\Lambda_t$ traces a geometry-aware, low-energy path between source and target while preserving endpoint and monotone constraints. *Center*: One-step schematic—given source $\hat{X}_0^{\text{src}}$ and target $X_0$, the field $\Lambda_t$ (predicted from $\lambda_t$ and position encoding) forms $d_t = \Lambda_t \odot \hat{X}_0^{\text{src}} + (1 - \Lambda_t) \odot X_0$ to guide denoising; insets show $\hat{X}_0^{\text{src}}$, $X_0$, and the translated output $\hat{X}_0^{\text{gen}}$. *Right*: Global linear interpolation $\eta_t \mathbf{1}$ forces a straight path through high-energy regions, leaving larger correction requirement to the diffusion model.

marginals are centered on a learned domain-mix signal and the reverse SDE drift contains an explicit, data-driven term that encodes the target-consistent shift at each time. This in-step injection of translation dynamics makes the sampler tolerant to significantly larger step sizes without sacrificing stability: the update direction always carries domain-aware correction, so coarse integration remains on-manifold. The continuous-time formulation yields an exact solution form for the reverse dynamics and a practical first-order sampler derived therefrom, enabling accurate updates with markedly fewer denoising steps. In contrast to external shortcuts, CDTSDE improves both semantic alignment and efficiency by aligning the numerical update with the true cross-modal transformation each step.

In summary, we advance cross-modal translation by embedding domain-shift dynamics into the generative process and adding an explicit restoration drift in the reverse-time SDE so that each update is pulled toward the target modality's image features. **First,** we center the forward marginals on a domain-aware mixture and provide stepwise, target-aligned guidance that keeps larger integration steps stable, reduces denoising iterations, and reframes the score model as a residual corrector. **Second,** we present a continuous-time formulation with a closed-form expression of the reverse dynamics and derive a first-order sampler that preserves marginal consistency, combining theoretical soundness with practical efficiency. **Finally,** on SAR → optical image, Electroluminescence → Semantic-mask, and MRI contrast translation tasks, our method is faster and more accurate than linear or fixed-schedule diffusion baselines, establishing shift-aware restoration as an effective inductive bias for efficient and faithful cross-modal synthesis.

## 2 RELATED WORK

### 2.1 CROSS-MODALITY IMAGE TRANSLATION

Early methods for cross-modality image translation largely relied on direct intensity mappings and voxel-wise regression models, which demanded accurate spatial correspondence between modalities (Huynh et al., 2016; Han, 2017; Nie et al., 2017). GAN-based approaches such as CycleGAN had more become popular due to their ability to learn complex transformations without strict spatial alignment requirements, showing substantial success in medical imaging and multi-spectral domain adaptation tasks (Cai et al., 2019; Armanious et al., 2020; Arar et al., 2020). Despite these successes, GANs still encounter training instability and architectural complexities.

Diffusion models (Ho et al., 2020; Song et al., 2021) have recently emerged as stable alternatives to GANs, achieving superior image synthesis quality. Methods such as SDEdit (Meng et al., 2022) and ILVR (Choi et al., 2021) leverage diffusion models for image-to-image translation, incorporating source-domain conditions into diffusion sampling. Recent advancements in Bridge-based Diffusion models like DDIB DDBM, I2SB, DBIM, ABridge (Su et al., 2023; Zhou et al., 2023; Liu et al., 2023; Zheng et al., 2025; Xiao et al., 2025) and energy-guided diffusion models (EGSDE) (Zhao et al., 2022) introduce additional guidance to enhance cross-modality translation. Medical imaging-

specific adaptations integrate frequency-domain constraints (Li et al., 2024) and local mutual information guidance (Wang et al., 2024d) to improve fidelity. Hybrid strategies, such as adversarially constrained diffusion models (Özbey et al., 2023), further enforce anatomical realism. Despite these advances, most existing models implicitly rely on fixed or linear interpolation between modalities in either the signal or latent space. Such rigid schedules fail to accommodate the non-uniform semantic complexity of cross-modal translation (Li et al., 2023), often requiring the diffusion model itself to compensate for suboptimal guidance paths. By explicitly learning a domain-aware shift trajectory adapted to translation difficulty, our method avoids sampling overhead and under-constrained synthesis to produce efficient, semantically aligned results.

## 2.2 CROSS DOMAIN SHIFT

Domain shift in diffusion-based generation refers to the transformation required to carry samples from a *source* distribution $\mathcal{D}_s$ to a *target* distribution $\mathcal{D}_t$. Fixed linear blends (Cui et al., 2025) work for super-resolution, where the source–target relation is roughly affine; yet, they break down in cross-modality translation with non-linear shifts in intensity, texture, and semantics. Early methods address domain shift through simple conditioning mechanisms. SDEdit (Meng et al., 2022) introduces stochastic corruption followed by denoising, relying on a fixed noise level as a control parameter, whereas ILVR (Choi et al., 2021) conditions generation by replacing the low-frequency spectrum with that of the input image.

Recent works embed domain transformation into the generative process itself. BBDM (Li et al., 2023) constructs Brownian bridges between source and target samples to guide diffusion along domain-aware stochastic paths. In the medical imaging domain, LMI-guided diffusion (Wang et al., 2024d) enhances semantic correspondence by enforcing patch-wise mutual information alignment, while Target-Guided Diffusion Models (Luo et al., 2024) modulate reverse-time gradients with perceptual cues from the target modality. Other works adopt cross-conditioned designs (Xing et al., 2024) or domain-aligned augmentations such as Diffuse-UDA (Gong et al., 2024) to promote style and structure alignment. Meanwhile, adaptive scheduling strategies are being introduced to learn non-linear transitions that minimize semantic discrepancies. For instance, ANT (Lee et al., 2024) learns instance-specific noise curves from domain statistics, and Diffusion Bridge (Lee et al., 2025) constructs polynomial flows in latent space to approximate Wasserstein-optimal transitions from $\mathcal{D}_s$ to $\mathcal{D}_t$. Although these works leverage the strong generative power of diffusion models, their efficiency drawbacks limits broader use in cross-modal image translation tasks.

## 3 METHOD

Cross-modal image translation seeks a mapping $\mathcal{T} : \mathcal{X}_{\text{source}} \to \mathcal{X}_{\text{tar}}$ that converts a source-modality observation $\hat{\boldsymbol{x}}_0 \in \mathcal{X}_{\text{source}}$ into a perceptually and semantically consistent target-modality image $\hat{\boldsymbol{x}}_0^{\text{gen}} \in \mathcal{X}_{\text{tar}}$. During training we assume access to paired samples $(\hat{\boldsymbol{x}}_0, \boldsymbol{x}_0)$, with $\boldsymbol{x}_0 \in \mathcal{X}_{\text{tar}}$ providing ground-truth supervision. The objective is to minimize the expected discrepancy $\mathbb{E}\big[\|\hat{\boldsymbol{x}}_0^{\text{gen}} - \boldsymbol{x}_0\|_2^2\big]$ while ensuring fast inference. As illustrated in the center of Fig. 1, a source image $\hat{\boldsymbol{x}}_0$ (e.g., SAR) is transformed through a learned generative process into a translated target image $\hat{\boldsymbol{x}}_0^{\text{gen}}$ (e.g., optical).

We adopt a conditional denoising diffusion model. Given a clean target image $\boldsymbol{x}_0$, the forward process successively corrupts it with Gaussian noise to obtain $\boldsymbol{x}_T \sim \mathcal{N}(\boldsymbol{0}, \boldsymbol{I})$. Generation begins at noise and follows the learned reverse SDE back to a clean sample. In the cross-modal setting direct conditioning on $\hat{\boldsymbol{x}}_0$ is essential for content preservation. A common implementation of conditional diffusion employs a time-varying linear blend of source and target observations at each reverse step, given by $\boldsymbol{x}_{\text{rect}}^{\text{lin}} = \eta_t\, \hat{\boldsymbol{x}}_0 + (1 - \eta_t)\, \boldsymbol{x}_0$, with $\eta_t = t/T$, where $\hat{\boldsymbol{x}}_0$ is the source-modality input and $\boldsymbol{x}_0$ is the paired target ground truth. This linear interpolation implicitly assumes that the latent transition between modalities lies along a straight path: a valid simplification when the appearance gap between source and target domains is relatively minor.

However, as illustrated in Fig. 1 (right), this assumption often breaks down when modality shifts are substantial. In such cases, the straight-line path rapidly departs from the source manifold (pink region), traverses ambiguous high-energy zones that belong to neither modality, and only re-enters the target manifold (blue region) near the end of the reverse trajectory. This suboptimal shortcut forces the diffusion backbone to spend excessive effort correcting deviations from both structural

and semantic priors, resulting in oscillatory updates, increased stochasticity, and blurred outputs. In contrast, Fig. 1 (left) shows that the optimal transition path (the geodesic in latent space) naturally bends around the energy ridge that separates the modalities. To capture this nonlinearity, we introduce a learnable polynomial schedule that adapts the interpolation dynamically. By aligning the reverse SDE with the valley floor connecting source and target statistics, this adaptive shift path reduces the burden of global realignment and allows the model to concentrate its capacity on local, high-frequency reconstruction. The result is improved fidelity with fewer function evaluations.

**Overview:** Before introducing the full stochastic formulation, we summarize the main objects. At each reverse step $t$, the model predicts a mixing field $\Lambda_t \in (0,1)^{C \times H \times W}$ and forms a domain mixture $d_t = \Lambda_t \odot \hat{x}_0^{\mathrm{src}} + (1 - \Lambda_t) \odot x_0$ (Eq. (1)). Intuitively, $\Lambda_t$ controls, per pixel and channel, how much we trust the source input versus the target domain at time $t$. Section 3.1 formalizes this construction and introduces a path-energy functional $E[d]$ (Eq. (4)) that measures how natural the transition path $d(t)$ is between domains. Sections 3.2, 3.3, 3.4 then show how this mixture is embedded into a diffusion process (Eqs. (6)–(9)). Notations are provided in the Appendix 4

## 3.1 ADAPTIVE DYNAMIC DOMAIN SHIFT

To address these challenges, we propose a spatially varying, channel-aware domain-mixture guidance that, at each reverse step $t$, learns a full domain transform field $\Lambda_t \in (0,1)^{C \times H \times W}$, and uses it to form a *domain mixture* between the *source-domain* observation $\hat{x}_0^{\mathrm{src}}$ and the *target-domain* variable $x_0$:

$$d_t = \Lambda_t \odot \hat{x}_0^{\mathrm{src}} + (1 - \Lambda_t) \odot x_0, \tag{1}$$

where $\odot$ denotes the Hadamard product. The resulting $d_t$ serves as the domain-shift guidance used by the sampler at step $t$.

**Spatial Modulation** Let $\mathbf{p} \in \{1, \ldots, H\} \times \{1, \ldots, W\}$ index spatial location and $c \in \{1, \ldots, C\}$ index channels. We build a position encoding $\boldsymbol{\pi}(\mathbf{p}) \in \mathbb{R}^4$ (sine/cosine of normalized coordinates defined in Appendix Sec. 6.1) and feed the broadcasted base step $\lambda_t^{\mathrm{lin}}$ together with $\boldsymbol{\pi}(\mathbf{p})$ into a light convolutional network $\mathcal{S}_\theta$ to produce a mid-term modulation $h_{t,c}(\mathbf{p}) \in (0,1)$:

$$h_{t,c}(\mathbf{p}) = \mathcal{S}_\theta(\lambda_t^{\mathrm{lin}}, \boldsymbol{\pi}(\mathbf{p})).$$

Mapping $h$ to a zero-centered signal $g_{t,c}(\mathbf{p}) = 2h_{t,c}(\mathbf{p}) - 1 \in (-1, 1)$, we form an interpolation that satisfies $f_{t,c}(0) = 0$ and $f_{t,c}(1) = 1$:

$$f_{t,c}(\lambda_t^{\mathrm{lin}}, \mathbf{p}) = \lambda_t^{\mathrm{lin}} \Big[ 1 + g_{t,c}(\mathbf{p})(1 - \lambda_t^{\mathrm{lin}}) \Big]. \tag{2}$$

Finally, we squash $f$ into $(0,1)$ with a calibrated logistic map. Fix a small $\varepsilon = 10^{-4}$ and set

$$\mathrm{logit}(u) := \ln \tfrac{u}{1-u}, \qquad \beta = -\mathrm{logit}(\varepsilon), \qquad \alpha = -2\,\mathrm{logit}(\varepsilon).$$

Define

$$\Lambda_{t,c}(\mathbf{p}) = \sigma\big(\alpha\, f_{t,c}(\lambda_t^{\mathrm{lin}}, \mathbf{p}) - \beta\big), \qquad \sigma(z) = \tfrac{1}{1+e^{-z}}. \tag{3}$$

With this choice, $\Lambda_{t,c}(\mathbf{p}) = \varepsilon$ when $f_{t,c} = 0$ and $\Lambda_{t,c}(\mathbf{p}) = 1 - \varepsilon$ when $f_{t,c} = 1$. To satisfy exact endpoint constraints used elsewhere, we clamp at the boundaries: $\Lambda_{0,c}(\mathbf{p}) = 0$; $\Lambda_{T,c}(\mathbf{p}) = 1$, while for $t \in \{1, \ldots, T-1\}$ we use the Eq 3. This keeps $\Lambda_{t,c} \in (0,1)$ and preserves monotonicity in $t$, while allocating extra degrees of freedom *only in the middle* of the trajectory. Jointly training $\theta$ with the diffusion backbone lets the schedule bend early toward $\hat{x}_0$ in regions where denoising is confident, while deferring and concentrating effort on hard, modality-specific details where needed.

**Geometric View** Consider the trajectory function $d(t) = \Lambda(t) \odot \hat{x}_0^{\mathrm{src}} + (1 - \Lambda(t)) \odot x_0$ for $t \in [0,1]$, and the path energy

$$\mathcal{E}[d] = \int_0^1 \sum_{c,\mathbf{p}} \Big( a_{c,\mathbf{p}}(t) \big|\dot{d}_{c,\mathbf{p}}(t)\big|^2 + U_{c,\mathbf{p}}\big(d_{c,\mathbf{p}}(t), t\big) \Big) dt, \tag{4}$$

where $a_{c,\mathbf{p}}(t) > 0$ is a local metric and $U_{c,\mathbf{p}}(\cdot, t)$ is a strictly convex local potential in its first argument. We compare two admissible path classes (both obey the endpoint and monotonicity induced

by equation 2 and equation 3): $\mathcal{C}_{\text{pix}} = \left\{ \Lambda(t) \in (0,1)^{C \times H \times W} \right\}$, $\mathcal{C}_{\text{glob}} = \left\{ \Lambda(t) \equiv \eta(t)\mathbf{1}, \ \eta : [0,1] \to [0,1] \right\}$. Clearly $\mathcal{C}_{\text{glob}} \subset \mathcal{C}_{\text{pix}}$.

**Remark:** $\mathcal{E}[\boldsymbol{d}]$ is a path energy that evolves over time and is used to measure whether the transition path from the target domain to the source domain is natural and has a lower cost.

**Theorem 1.** *Assume there exists a set of indices* $\mathcal{S} \subset \{(c, \mathbf{p})\}$ *of positive measure and an interval* $I \subset (0,1)$ *such that: (i) (heterogeneous local geometry) the potentials* $\{U_{c,\mathbf{p}}(\cdot, t)\}_{(c,\mathbf{p}) \in \mathcal{S}}$ *or the metrics* $\{a_{c,\mathbf{p}}(t)\}$ *are not identical across* $(c, \mathbf{p})$ *over* $I$; *(ii) (nondegenerate contrast)* $\Delta_{c,\mathbf{p}} := \hat{x}_c^{\text{src}}(\mathbf{p}) - x_{0,c}(\mathbf{p}) \neq 0$ *on* $\mathcal{S}$ *and this contrast does not annihilate the local slope along the global path on a set of positive measure; and (iii) (endpoint & monotonic feasibility) both classes admit paths satisfying the endpoint and monotone schedule constraints. Then*

$$\inf_{\Lambda \in \mathcal{C}_{\text{pix}}} \mathcal{E}[\boldsymbol{d}] \ < \ \inf_{\Lambda \in \mathcal{C}_{\text{glob}}} \mathcal{E}[\boldsymbol{d}] \tag{5}$$

**Remark:** Since $\mathcal{C}_{\text{glob}} \subset \mathcal{C}_{\text{pix}}$, the non-strict inequality is immediate. Intuitively, view $\boldsymbol{d}(t)$ as the evolving translated image with kinetic–potential energy equation 4. (i) *Heterogeneous local geometry* implies different pixels & channels have different "best" mixture to reduce their local potentials $U_{c,\mathbf{p}}$; a global $\eta(t)\mathbf{1}$ must compromise, incurring avoidable potential penalties somewhere. (ii) *Nondegenerate contrast* ($\Delta_{c,\mathbf{p}} \neq 0$) ensures that pixelwise adjustments actually change $d_{c,\mathbf{p}}(t)$ in the potential's descent direction, yielding a genuine decrease of the potential term on a positive-measure subset. (iii) *Endpoint & monotone feasibility* lets these adjustments be implemented as smooth, localized, time-spread perturbations of $\Lambda(t)$ that preserve endpoints/monotonicity. Theorem 1 states that, under mild heterogeneity, allowing $\Lambda(t)$ to vary across pixels admits a path with strictly lower energy than any global schedule. Thus a dynamic domain transform path $\Lambda(t)$ strictly lowers total energy compared to any global path $\eta(t)\mathbf{1}$. Detailed proof is in Appendix 6.2.

## 3.2 CROSS-MODAL DIFFUSION PROCESS

We couple the adaptive dynamic domain shift with a variance–preserving diffusion process. Let the base noise schedule be $\{\alpha_t\}_{t=1}^T \subset (0,1)$, with $\bar{\alpha}_t = \prod_{s=1}^t \alpha_s$ and $\sigma_t^2 = 1 - \bar{\alpha}_t$. At each step we use the spatially varying, channel–aware mixing field $\Lambda_t \in (0,1)^{C \times H \times W}$ to form the domain mixture defined in Eq. 1 and set the forward marginal to

$$q(\boldsymbol{x}_t \mid \boldsymbol{x}_0, \hat{\boldsymbol{x}}_0^{\text{src}}) \ = \ \mathcal{N}\big(\boldsymbol{x}_t; \ \sqrt{\bar{\alpha}_t}\, \boldsymbol{d}_t, \ \sigma_t^2 \boldsymbol{I}\big), \qquad t = 1, \ldots, T. \tag{6}$$

The corresponding one–step Markov transition consistent with equation 6 is

$$q(\boldsymbol{x}_t \mid \boldsymbol{x}_{t-1}, \boldsymbol{x}_0, \hat{\boldsymbol{x}}_0^{\text{src}}) \ = \ \mathcal{N}\Big(\boldsymbol{x}_t; \ \rho_t \boldsymbol{x}_{t-1} + \sqrt{\bar{\alpha}_t}\big(\boldsymbol{d}_t - \boldsymbol{d}_{t-1}\big), \ \big(1 - \rho_t^2\big)\boldsymbol{I}\Big), \tag{7}$$

where $\rho_t := \sqrt{\bar{\alpha}_t / \bar{\alpha}_{t-1}} = \sqrt{\alpha_t}$ and $\boldsymbol{d}_t$ is given by equation 1. Equivalently, using the mixture identity

$$\boldsymbol{d}_t - \boldsymbol{d}_{t-1} \ = \ (\Lambda_t - \Lambda_{t-1}) \odot \big(\hat{\boldsymbol{x}}_0^{\text{src}} - \boldsymbol{x}_0\big),$$

the mean in equation 7 can be written explicitly in terms of the field increment $\Lambda_t - \Lambda_{t-1}$.

We use a middle point $t_1 < T$ and fix $\Lambda_t \equiv \mathbf{1}$ for all $t \geq t_1$.(Cui et al., 2025; Meng et al., 2022) In this regime the forward mean becomes $\sqrt{\bar{\alpha}_t}\,\hat{\boldsymbol{x}}_0^{\text{src}}$, so the diffusion is a pure noise process centered at the source observation. At inference time, one can initialize directly from $\boldsymbol{x}_{t_1} \sim \mathcal{N}\big(\sqrt{\bar{\alpha}_{t_1}}\hat{\boldsymbol{x}}_0^{\text{src}}, \sigma_{t_1}^2 \boldsymbol{I}\big)$ and run the learned reverse process from $t_1$ to 0, effectively skipping approximately $T - t_1$ denoising steps while preserving translation fidelity.

## 3.3 CONTINUOUS-TIME FORMULATION

Let $\Lambda(t) \in (0,1)^{C \times H \times W}$ be a differentiable, spatially varying mixer and define $\boldsymbol{d}(t) = \Lambda(t) \odot \hat{\boldsymbol{x}}_0^{\text{src}} + \big(\mathbf{1} - \Lambda(t)\big) \odot \boldsymbol{x}_0$. With $\bar{\alpha}_t = \prod_{s=1}^t \alpha_s$ and $\sigma_t^2 = 1 - \bar{\alpha}_t$, the forward SDE is

$$d\boldsymbol{x}_t \ = \ \Big[f(t)\,\boldsymbol{x}_t \ + \ \sqrt{\bar{\alpha}_t}\,\dot{\Lambda}(t) \odot \big(\hat{\boldsymbol{x}}_0^{\text{src}} - \boldsymbol{x}_0\big)\Big] dt \ + \ g(t)\,d\boldsymbol{w}_t, \tag{8}$$

with $f(t) = \frac{d}{dt} \ln \sqrt{\bar{\alpha}_t}$, and $g(t) = \sqrt{\frac{d\sigma_t^2}{dt} - 2\sigma_t^2 f(t)}$. In SDE equation 8, the additional drift is $\sqrt{\bar{\alpha}_t} \, \dot{\Lambda}(t) \odot (\hat{x}_0^{\mathrm{src}} - x_0)$. Importantly, solving the moment ODE implied by Eq. 8 shows that the added drift $\sqrt{\bar{\alpha}_t} \, \dot{\Lambda}(t) \odot (\hat{x}_0^{\mathrm{src}} - x_0)$ makes the forward mean track the domain-mixture path $\sqrt{\bar{\alpha}_t} \, d(t)$, thereby realizing on-manifold, low-energy transport that simplifies the subsequent reverse-time inference.

The corresponding reverse-time SDE (see proof in Appendix 6.8) corresponding to the forward model is:

$$d\boldsymbol{x}_t = \left[ f(t)\, \boldsymbol{x}_t + \sqrt{\bar{\alpha}(t)} \, \dot{\Lambda}(t) \odot (\hat{\boldsymbol{x}}_0^{\mathrm{src}} - \boldsymbol{x}_0) - g^2(t)\, \nabla_{\boldsymbol{x}} \log q_t(\boldsymbol{x}_t \mid \boldsymbol{x}_0, \hat{\boldsymbol{x}}_0^{\mathrm{src}}) \right] dt + g(t)\, d\bar{\boldsymbol{w}}_t, \quad (9)$$

### 3.4 SAMPLER

In this section, we present the solution of the SDE 9 under our mixture model and design efficient samplers for fast sampling. We employ a data prediction model $\boldsymbol{\Phi}_\theta(\boldsymbol{x}_t, \hat{\boldsymbol{x}}_0^{\mathrm{src}}, t)$ that estimates the clean target image $\boldsymbol{x}_0$ from noisy samples. The relationship between the score function and the data prediction model under the mixture forward marginal (Eq. 6) is:

$$\nabla_{\boldsymbol{x}} \log q_t(\boldsymbol{x}_t \mid \hat{\boldsymbol{x}}_0^{\mathrm{src}}) \simeq - \frac{\boldsymbol{x}_t - \sqrt{\bar{\alpha}_t} \, \boldsymbol{d}_t^\theta}{\sigma_t^2}, \qquad \boldsymbol{d}_t^\theta := \Lambda_t \odot \hat{\boldsymbol{x}}_0^{\mathrm{src}} + (1 - \Lambda_t) \odot \boldsymbol{\Phi}_\theta(\boldsymbol{x}_t, \hat{\boldsymbol{x}}_0^{\mathrm{src}}, t), \quad (10)$$

where $\sigma_t^2 = 1 - \bar{\alpha}_t$. See the proof in Appendix 6.6.

In practice, we train a noise prediction model $\boldsymbol{\varepsilon}_\theta(\boldsymbol{x}_t, \hat{\boldsymbol{x}}_0^{\mathrm{src}}, t)$. Under the same mixture marginal, the standard VP conversion gives

$$\boldsymbol{\varepsilon}_\theta(\boldsymbol{x}_t, \hat{\boldsymbol{x}}_0^{\mathrm{src}}, t) = \frac{\boldsymbol{x}_t - \sqrt{\bar{\alpha}_t} \, \boldsymbol{d}_t^\theta}{\sigma_t}, \quad \Longleftrightarrow \quad \boldsymbol{d}_t^\theta = \frac{\boldsymbol{x}_t - \sigma_t \, \boldsymbol{\varepsilon}_\theta(\boldsymbol{x}_t, \hat{\boldsymbol{x}}_0^{\mathrm{src}}, t)}{\sqrt{\bar{\alpha}_t}}. \quad (11)$$

By substituting Eq. 10 into the reverse SDE

$$\boldsymbol{\Upsilon}_t := \sqrt{\bar{\alpha}_t}(1 - \Lambda_t), \qquad \boldsymbol{y}_t := \boldsymbol{x}_t \oslash \boldsymbol{\Upsilon}_t, \qquad \boldsymbol{\lambda}_t := \sigma_t \oslash \boldsymbol{\Upsilon}_t,$$

(with $\oslash$ denoting Hadamard division), together with the shorthand

$$d\boldsymbol{w}_{\boldsymbol{\lambda}} := \sqrt{\frac{d\boldsymbol{\lambda}_t}{dt}} \odot d\bar{\boldsymbol{w}}_t, \quad \boldsymbol{x}_{\boldsymbol{\lambda}} := \boldsymbol{x}_{t(\boldsymbol{\lambda})}, \quad \boldsymbol{w}_{\boldsymbol{\lambda}} := \boldsymbol{w}_{\boldsymbol{\lambda}_t},$$

we obtain the reverse dynamics with respect to $\boldsymbol{\lambda}$:

$$
\begin{aligned}
d\boldsymbol{y}_{\boldsymbol{\lambda}} = & \left(\tfrac{2}{\boldsymbol{\lambda}}\right) \odot \boldsymbol{y}_{\boldsymbol{\lambda}} \, d\boldsymbol{\lambda} + \left[ (1 \oslash (1 - \Lambda_{\boldsymbol{\lambda}})^2) \odot d\Lambda_{\boldsymbol{\lambda}} - \left(\tfrac{\Lambda_{\boldsymbol{\lambda}}}{1 - \Lambda_{\boldsymbol{\lambda}}}\right) \odot \left(\tfrac{2}{\boldsymbol{\lambda}}\right) d\boldsymbol{\lambda} \right] \odot \hat{\boldsymbol{x}}_0^{\mathrm{src}} \\
& - \left(\tfrac{2}{\boldsymbol{\lambda}}\right) \odot \boldsymbol{\Phi}_\theta(\boldsymbol{x}_{\boldsymbol{\lambda}}, \hat{\boldsymbol{x}}_0^{\mathrm{src}}, \boldsymbol{\lambda}) \, d\boldsymbol{\lambda} + \left(\tfrac{g(t)}{\boldsymbol{\Upsilon}_t}\right) \oslash \sqrt{\tfrac{d\boldsymbol{\lambda}_t}{dt}} \odot d\boldsymbol{w}_{\boldsymbol{\lambda}}
\end{aligned}
\quad (12)
$$

where all divisions, square-roots, and products are Hadamard operation. The equation 12 can be solved exactly via the variation-of-constants formula (Cui et al., 2025).

**Proposition 1.** *Let $\boldsymbol{\Upsilon}_t := \sqrt{\bar{\alpha}_t}(1 - \Lambda_t)$ and $\boldsymbol{\lambda}_t := \sigma_t \oslash \boldsymbol{\Upsilon}_t$ ($\oslash$ denotes Hadamard division). Given an initial value $\boldsymbol{x}_s$ at time $s > 0$, the solution $\boldsymbol{x}_t$ of the reverse-time SDE Eq. 12 for $t \in [0, s]$ is*

$$\boldsymbol{x}_t = \boldsymbol{\Upsilon}_t \odot \left(\boldsymbol{\lambda}_t^2 \oslash \boldsymbol{\lambda}_s^2\right) \odot \left(\boldsymbol{x}_s \oslash \boldsymbol{\Upsilon}_s\right) - \boldsymbol{\Upsilon}_t \odot \int_{\boldsymbol{\lambda}_s}^{\boldsymbol{\lambda}_t} \left(2\, \boldsymbol{\lambda}_t^2 \oslash \boldsymbol{\lambda}^3\right) \odot \boldsymbol{\Phi}_\theta(\boldsymbol{x}_{\boldsymbol{\lambda}}, \hat{\boldsymbol{x}}_0^{\mathrm{src}}, \boldsymbol{\lambda}) \, d\boldsymbol{\lambda}$$

$$+ \boldsymbol{\Upsilon}_t \odot \left[ \left(\Lambda_t \oslash (1 - \Lambda_t)\right) - \left(\Lambda_s \oslash (1 - \Lambda_s)\right) \odot \left(\boldsymbol{\lambda}_t^2 \oslash \boldsymbol{\lambda}_s^2\right) \right] \odot \hat{\boldsymbol{x}}_0^{\mathrm{src}} \quad (13)$$

$$+ \boldsymbol{\Upsilon}_t \odot \int_{\boldsymbol{\lambda}_s}^{\boldsymbol{\lambda}_t} \left(\boldsymbol{\lambda}_t^2 \oslash \boldsymbol{\lambda}^2\right) \odot \left(\tfrac{g}{\boldsymbol{\Upsilon}}\right) \oslash \sqrt{\tfrac{d\boldsymbol{\lambda}}{dt}} \, d\boldsymbol{w}_{\boldsymbol{\lambda}} \, .$$

Here $g = g(t(\boldsymbol{\lambda}))$ and $\boldsymbol{\Upsilon} = \boldsymbol{\Upsilon}_{t(\boldsymbol{\lambda})}$ along the integration path; all operations are Hadamard. We give the corresponding first-order solver in the Appendix 6.9.

Empirically, as shown in the next experimental section, this guided path enables accurate translation in fewer steps compared other shift methods, suggesting the learned schedule implicitly captures optimal tradeoffs between semantic alignment and spatial refinement.

## 4 EXPERIMENTS

### 4.1 EXPERIMENTAL SETUP

**Tasks and Datasets** We consider: (i) MRI modality translation (T1→T2) on IXI; (ii) SAR→Optical translation on co-registered Sentinel-1/2; (iii) EL-to-semantic mapping for solar-cell defects on PSCDE. The difficulty increases from minor intra-structural contrast changes (IXI), to cross-sensor appearance shifts (Sentinel), to fine-grained defect synthesis under strong acquisition and morphology variability (PSCDE). Full curation protocols, splits, and counts appear in Appendix 6.10.

**Implementation Overview** We use a UNet-based conditional diffusion backbone implemented in PyTorch Lightning with mixed precision. The adaptive domain shift component follows a learnable schedule; optimization and training lengths are task-specific. These three tasks reflect increasing translation difficulty: IXI (minor intra-modality contrast), Sentinel-1/2 (cross-sensor appearance shift), and PSCDE (fine-grained, high-variability defect structures). Complete configurations (optimizer, learning rates, schedule degree/initialization, architectural choices) appear in Appendix 6.11.

**Baselines and Metrics** We compare against Pix2Pix (Isola et al., 2017), BBDM (Li et al., 2023), ABridge (Xiao et al., 2025), DBIM Zheng et al. (2025), and DOSSR (Cui et al., 2025). We report SSIM/PSNR, MSE/MAE; per-method notes are in Appendix 6.12.

### 4.2 QUANTITATIVE RESULTS

Table 1 presents comprehensive quantitative comparisons across three representative cross-modal translation tasks. Our CDTSDE method consistently achieves top performance across nearly all evaluation metrics, validating the effectiveness of its adaptive domain shift mechanism.

Table 1: Quantitative comparison across three tasks with one row per method. Datasets: Sentinel (SAR→Optical), IXI (T1→T2), and IXI (T2→T1). Metrics: SSIM ($\uparrow$), PSNR ($\uparrow$), MSE ($\downarrow$), MAE ($\downarrow$). Best in **bold**, second-best underlined within each dataset block. *Training protocol:* For fair comparison, all models on Sentinel are trained for 20,000 steps, on IXI for 10,000 steps, and on PSCDE for 5,000 steps (except Pix2Pix). *Sampling:* Because BBDM lacks a fast-sampling variant, we use its recommended default of 1,000 sampling steps. Training curves are in the Appendix 9.

| | Sentinel (SAR→Optical) | | | | IXI (T1→T2) | | | | IXI (T2→T1) | | | |
|---|---|---|---|---|---|---|---|---|---|---|---|---|
| Method | SSIM↑ | PSNR↑ | MSE↓ | MAE↓ | SSIM↑ | PSNR↑ | MSE↓ | MAE↓ | SSIM↑ | PSNR↑ | MSE↓ | MAE↓ |
| Pix2Pix | 0.23±0.1 | 15.12±1.4 | 0.03±0.0 | 0.12±0.0 | 0.71±0.0 | 22.17±1.2 | 0.01±0.0 | 0.03±0.0 | 0.71±0.0 | 22.24±1.4 | 0.01±0.0 | 0.03±0.0 |
| ABridge | 0.15±0.1 | 12.50±2.0 | 0.06±0.0 | 0.18±0.1 | 0.44±0.0 | 13.73±2.4 | 0.04±0.0 | 0.12±0.0 | 0.35±0.0 | 13.77±2.5 | 0.04±0.0 | 0.12±0.0 |
| DOSSR | 0.36±0.1 | 17.14±1.5 | 0.02±0.0 | 0.10±0.0 | 0.75±0.0 | 23.28±1.4 | 0.01±0.0 | **0.03±0.0** | 0.80±0.1 | 24.13±2.0 | 0.01±0.0 | 0.03±0.0 |
| BBDM | 0.02±0.0 | 6.25±0.8 | 0.24±0.0 | 0.40±0.0 | 0.01±0.0 | 8.63±0.3 | 0.13±0.0 | 0.25±0.0 | 0.01±0.0 | 8.69±0.4 | 0.13±0.0 | 0.25±0.0 |
| DBIM | 0.14±0.0 | 13.48±1.4 | 0.05±0.0 | 0.16±0.0 | 0.28±0.0 | 20.43±0.8 | 0.01±0.0 | 0.08±0.0 | 0.33±0.0 | 21.65±1.5 | 0.01±0.0 | 0.06±0.0 |
| CDTSDE | **0.38±0.1** | **17.46±1.5** | **0.01±0.0** | **0.09±0.0** | **0.76±0.0** | **23.32±1.3** | 0.01±0.0 | 0.03±0.0 | **0.82±0.0** | **24.33±1.9** | 0.01±0.0 | **0.03±0.0** |

**SAR-to-Optical Translation** On Sentinel, CDTSDE leads all the reported metrics: SSIM ($0.382 \pm 0.077$), PSNR ($17.46 \pm 1.46$, dB), MSE ($0.0190 \pm 0.0068$), and MAE ($0.0982 \pm 0.0198$), with DOSSR consistently second. DBIM and ABRIDGE improve over BBDM but remain well below Pix2Pix/DOSSR/CDTSDE in fidelity; for instance, DBIM attains SSIM ($0.140 \pm 0.044$), PSNR ($13.48 \pm 1.44$, dB), MSE ($0.050 \pm 0.020$), and MAE ($0.160 \pm 0.031$), compared with ABRIDGE's SSIM ($0.1541 \pm 0.0747$), PSNR ($12.50 \pm 1.96$, dB), MSE ($0.0627 \pm 0.0341$), and MAE ($0.1773 \pm 0.0566$). Pix2Pix provides mid-tier performance (e.g., PSNR $\approx 15.12$, dB) yet still trails CDTSDE and DOSSR, while BBDM remains the weakest across metrics.

**Medical Image Translation (IXI: T1↔T2)** Across both directions on IXI, CDTSDE consistently delivers the strongest fidelity. For T1→T2 it attains the highest SSIM ($0.762 \pm 0.042$) and PSNR ($23.32 \pm 1.25$ dB) with the lowest MSE ($0.0049 \pm 0.0016$); for T2→T1 it further improves to SSIM ($0.825 \pm 0.038$), PSNR ($24.33 \pm 1.86$ dB), MSE ($0.0040 \pm 0.0018$), and MAE ($0.0324 \pm 0.0076$). Among diffusion bridge-based methods, DBIM improves over other bridge-based ABridge and BBDM on PSNR/MSE/MAE (e.g., for T2→T1 it reaches SSIM ($0.33 \pm 0.04$), PSNR ($21.65 \pm 1.45$ dB), MSE ($0.010 \pm 0.004$), MAE ($0.060 \pm 0.015$)) yet still lags behind Pix2Pix, DOSSR, and CDTSDE in structural fidelity. DOSSR is a close second in SSIM/PSNR/MSE and attains the lowest MAE on T1→T2 ($0.0321 \pm 0.0060$). Pix2Pix is weaker, and ABridge and BBDM show marked degradation and structural artifacts consistent with their low SSIM/PSNR.

Table 2: PSCDE dataset: semantic translation metrics. Higher is better except Hausdorff.

| Method | Dice ↑ | IoU ↑ | Precision ↑ | Recall ↑ | Hausdorff ↓ | Skeleton F1 ↑ |
|---|---|---|---|---|---|---|
| Pix2Pix | 0.178 ± 0.374 | 0.173 ± 0.363 | 0.186 ± 0.387 | 0.173 ± 0.364 | 156.28 ± 90.84 | 0.053 ± 0.124 |
| ABridge | 0.009 ± 0.029 | 0.005 ± 0.016 | 0.080 ± 0.237 | 0.072 ± 0.144 | 161.89 ± 71.66 | 0.008 ± 0.015 |
| BBDM | 0.024 ± 0.036 | 0.012 ± 0.019 | 0.069 ± 0.139 | 0.228 ± 0.117 | 195.44 ± 61.42 | 0.030 ± 0.028 |
| DBIM | 0.167 ± 0.360 | 0.158 ± 0.342 | 0.427 ± 0.482 | 0.165 ± 0.355 | 101.49 ± 85.28 | 0.004 ± 0.010 |
| DOSSR | 0.460 ± 0.331 | 0.370 ± 0.337 | 0.659 ± 0.321 | 0.413 ± 0.335 | 59.53 ± 62.50 | 0.587 ± 0.328 |
| CDTSDE | **0.488 ± 0.351** | **0.403 ± 0.352** | **0.784 ± 0.258** | **0.432 ± 0.356** | **39.87 ± 50.76** | **0.614 ± 0.341** |

**Industrial Defect Detection** On PSCDE, CDTSDE delivers the strongest pixel-level performance and topology preservation, achieving the highest Dice (0.488) and IoU (0.403), the best Skeleton F1 (0.614), and the lowest boundary error by Hausdorff distance (39.87). It also attains the best precision/recall balance (0.784/0.432), enabling recovery of thin structures under noise. DOSSR is consistently second across all metrics (Dice 0.460, IoU 0.370, Precision 0.659, Recall 0.413, Hausdorff 59.53, Skeleton F1 0.587). In contrast, the bridge-based diffusion methods DBIM, ABridge, and BBDM all lag far behind the top two: DBIM attains moderate region overlap (Dice 0.167, IoU 0.158) and the lowest Hausdorff (101.49) among the bridges but suffers from very poor Skeleton F1 (0.004), while ABridge and BBDM largely fail to produce meaningful semantic translations (Dice ≤ 0.024, IoU ≤ 0.012) and exhibit severe boundary errors (Hausdorff ≥ 161.89). Pix2Pix sits between these extremes, with lower overlap than the proposed methods (Dice 0.178, IoU 0.173) and large boundary deviation (Hausdorff 156.28) alongside weak structural fidelity (Skeleton F1 0.053).

Overall, CDTSDE ranks first or second overall, underscoring robust generalization.

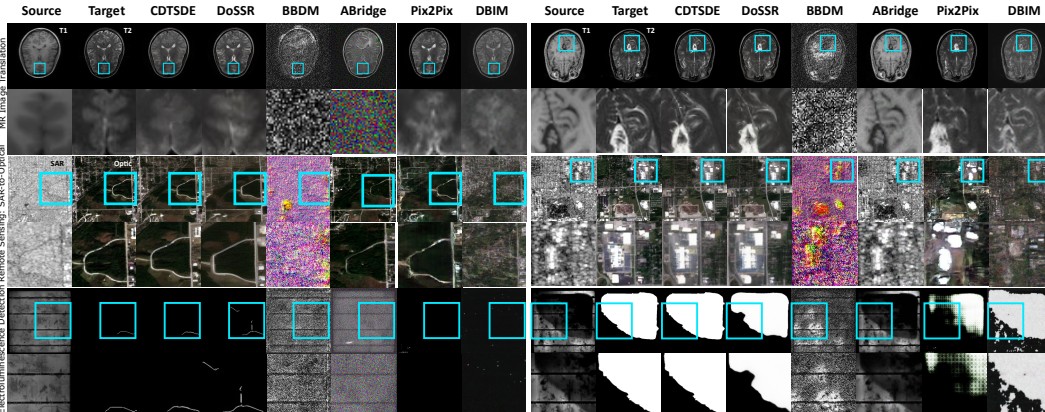

Figure 2: Visual comparison across three cross-modal tasks: IXI medical conversion, Sentinel SAR-to-optical, and PSCDE semantic mask. (See more in Appendix 6.15).

### 4.3 COMPUTATIONAL EFFICIENCY

To ensure a fair comparison, we evaluate methods at a matched-fidelity regime of PSNR ≈ 15 dB on Sentinel. For models capable of reaching this regime, we report the configuration that first attains PSNR ≈ 15 dB: DOSSR and CDTSDE at 10K training steps, and Pix2Pix at 125K training steps. Diffusion bridge-based methods: DBIM, ABridge and BBDM were trained at 20K; yet DBIM and BBDM did not reach PSNR ≈ 15 dB under comparable training budgets; we report their best checkpoints.

A key advantage of our CDTSDE framework is its efficiency at matched fidelity. As shown in Fig 3, CDTSDE reaches PSNR≈15 with only 5 sampling steps and 1.8s per image, whereas DOSSR needs 10 steps and 3.6s, giving a ∼2× speedup at similar memory (6.3 vs. 6.2 GB). Although Pix2Pix (GAN based) is fastest at test time (single forward pass), it requires **125K** training steps to reach comparable fidelity, while CDTSDE and DOSSR do so in **10K** steps, reflecting much higher training

efficiency. The diffusion bridge baselines BBDM, ABridge, and DBIM still fall well short of the PSNR≈15 band, highlighting the benefit of our adaptive domain-shift schedule.

**IXI & Sentinel Datasets:** Fig. 2 shows that on IXI, CDTSDE reconstructs anatomically coherent images with crisp cortical and ventricular boundaries. It improves over DOSSR, which slightly smooths details, and clearly surpasses Pix2Pix and the bridge-based methods (DBIM, ABridge, BBDM), which exhibit blur, noise, or partial structural collapse. On Sentinel (SAR→OPT), CDTSDE produces semantically aligned, natural optical images with stable geometry, while DOSSR loses fine detail, Pix2Pix exaggerates colors and textures, and the bridge-based methods leave fragmented or inconsistent patterns even when DBIM locally improves contrast.

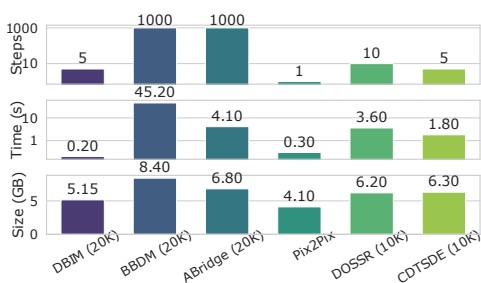

Figure 3: Computational efficiency.

**PSCDE (Semantic Translation):** PSCDE requires semantic recovery far beyond the other tasks, and CDTSDE preserves thin and fragmented structures under noise, maintaining mask continuity and topology (clearly see from the PSCDE columns: 2). DOSSR approximates shapes but slightly oversmooths edges, whereas Pix2Pix introduces spatial inconsistencies and over-segmentation. The bridge-based methods DBIM, ABridge, and BBDM struggle to recover fine structures, yielding jagged, disconnected, or artifact-heavy masks, so CDTSDE remains strongest on this dataset.

Overall, CDTSDE exhibits superior visual performance. More examples are in the Appendix 6.15.

### 4.4    ABLATION

We assess the contribution of the learnable dynamic domain-shift schedule against a fixed linear schedule. To isolate the scheduling effect, we compare three configurations. **Linear**: a fixed linear schedule $\eta_t$. **Channel Non-linear**: a per-channel but spatially uniform schedule that learns a non-linear transformation for each channel. **Dynamic**: the full non-linear with the dynamic spatial modulation domain-shift model that predicts $\Lambda_t$ at each reverse step. Additional implementation details and metrics for the intermediate Channel Non-linear variant are provided in Appendix 6.13.

On the PSCDE semantic mask generation task, where structural fidelity is paramount, the progression across the three schedules is informative. The Linear schedule attains a Dice score of $0.46$ and a Hausdorff distance of $59.5$ pixels. The Channel Non-linear variant keeps Dice essentially unchanged $(0.46)$ but reduces Hausdorff to $43.0$, indicating that per-channel adaptation already sharpens boundaries without improving region overlap. The fully spatially varying Dynamic schedule further raises Dice to $0.49$ and lowers Hausdorff to $39.8$, a relative gain of $+\mathbf{6.1}\%$ in Dice and a $\mathbf{33}\%$ reduction in boundary error compared to Linear. This trend suggests that channel-wise modulation explains part of the geometric improvement, while spatial adaptivity provides an additional boost in overlap accuracy and overall mask quality. On the remote-sensing benchmark, the Linear setting achieves an SSIM of $0.764$ and a PSNR of $23.28\,\mathrm{dB}$, whereas the Dynamic configuration improves these to $0.774$ and $23.37\,\mathrm{dB}$, corresponding to relative gains of about $1.3\%$ in SSIM and $0.4\%$ in PSNR. In cross-modality translation tasks, where SSIM values above $0.75$ already indicate near-saturation (Gu et al., 2023), the additional $\sim 1\%$ increase reflects a meaningful advance in fidelity.

## 5    DISCUSSION AND CONCLUSION

Our framework yields the largest gains on PSCDE, followed by Sentinel and IXI, reflecting increasing cross-domain complexity. In low-gap settings such as IXI, improvements are modest, suggesting that the added flexibility of CDTSDE is not always required. The formulation remains fully compatible with pretrained VP-based diffusion priors, making the approach transferable to broader image-to-image pipelines. A useful next step is to develop criteria that predict when adaptive, geometry-aware scheduling is beneficial, enabling simpler regressors in easy regimes. While CDTSDE leads on SSIM/PSNR/MSE, GAN baselines can occasionally achieve higher perceptual scores due to the

sharpness–faithfulness trade-off; lightweight perceptual or adversarial terms may help close this gap. Additional discussion, limit cases, and future extensions appear in Appendix 6.17.

In this work, we introduce a structure-adaptive domain-mixture schedule for diffusion models that follows a low-energy transition between domains. Theoretically, under mild heterogeneity and feasibility conditions, the proposed dynamic schedule strictly dominates any fixed global schedule in path energy, providing a principled explanation for its efficiency and fidelity gains. Experiments across MRI translation, SAR-to-optical conversion, and electroluminescence-to-semantic mapping highlight consistent improvements with fewer denoising steps, underscoring its readiness for real-world deployment in domains where both accuracy and efficiency are critical.

## ACKNOWLEDGMENTS

This research is supported by the Ruth S. Holmberg Grant for Faculty Excellence, and is supported in part by the Provost's Office of University of Tennessee at Chattanooga. We thank the anonymous reviewers of ICLR 2026 for their constructive feedback.

## ETHICS STATEMENT

This work focuses on image generation and cross-modal translation in medical imaging, remote sensing, and industrial inspection. Medical dataset (IXI) is involved but all datasets used (including Sentinel-1/2, PSCDE data) are publicly available and de-identified, and no personally identifiable or sensitive human subject data are included. The methods developed do not enable recovery of private or individual-level information, and all experiments comply with relevant data usage licenses. While cross-modal image generation could potentially be misused for falsification or disinformation in sensitive domains, our work is strictly intended for scientific and clinical research, industrial quality control, and remote sensing analysis. No conflicts of interest or external sponsorships influenced this study.

## REPRODUCIBILITY STATEMENT

The source code of the proposed method is available in `https://github.com/LaplaceLab/CDTSDE`.

The Appendix provides complete, step-by-step pseudocode (Algorithms 1–3) and key implementation details (model architectures, hyperparameters, training schedules, and data preprocessing in 6.11) for all procedures, enabling independent re-implementation without access to our codebase.

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

# 6 APPENDIX

## THE USE OF LARGE LANGUAGE MODELS (LLMS)

In this work, we employed large language models (LLMs) to assist with auxiliary programming tasks that are not directly related to the research methodology. Specifically, LLMs were used to polish the grammar and formatting of LaTeX tables, ensuring clarity and consistency in presentation. Additionally, portions of the source code were developed with the assistance of the GitHub Copilot plugin in Visual Studio Code, primarily for routine tasks such as data preprocessing. All code generated with the assistance of LLMs was carefully reviewed by human researchers, and input/output correctness was verified through visual inspection and testing to ensure reliability and reproducibility.

### 6.1 POSITION ENCODING FOR SPATIAL MODULATION

To provide spatial variation for the modulation network, we employ a fixed position encoding that maps each pixel coordinate $(x, y)$, normalized to $[-1, 1]^2$, into a four-dimensional embedding:

$$\boldsymbol{\pi}(x, y) = \begin{bmatrix} \sin(\pi y) \\ \cos(\pi y) \\ \sin(\pi x) \\ \cos(\pi x) \end{bmatrix}, \qquad (x, y) \in [-1, 1]^2. \tag{14}$$

Here $x = \frac{2i}{W-1} - 1$ and $y = \frac{2j}{H-1} - 1$ denote the normalized horizontal and vertical coordinates for pixel indices $i = 0, \ldots, W-1$ and $j = 0, \ldots, H-1$, respectively. The resulting tensor $\boldsymbol{\pi} \in \mathbb{R}^{4 \times H \times W}$ is broadcast along the batch dimension and concatenated with the stepwise coefficient $\lambda_t$ to serve as input to the spatial modulation network. This encoding introduces deterministic sinusoidal variations across spatial positions, enabling the network to learn adaptive mixture weights with consistent geometric priors.

### 6.2 PROOF OF THEOREM 1

**Preliminaries** Let $\bar{\Lambda}(t) = \eta(t)\mathbf{1} \in \mathcal{C}_{\text{glob}}$ be any admissible fixed-domain path and $\bar{d}(t) = \bar{\Lambda}(t) \odot \hat{x}_0^{\text{src}} + (\mathbf{1} - \bar{\Lambda}(t)) \odot x_0$ its trajectory. Denote the Gateaux variation of $\mathcal{E}$ at $\bar{d}$ along a perturbation $\delta\Lambda(t)$ by

$$\delta\mathcal{E}[\bar{d}; \delta\Lambda] = \int_0^1 \sum_{c,\mathbf{p}} \left( 2\, a_{c,\mathbf{p}}(t)\, \dot{\bar{d}}_{c,\mathbf{p}}(t)\, \delta\dot{d}_{c,\mathbf{p}}(t) + \partial_d U_{c,\mathbf{p}}\big(\bar{d}_{c,\mathbf{p}}(t), t\big)\, \delta d_{c,\mathbf{p}}(t) \right) dt,$$

where $\delta d_{c,\mathbf{p}}(t) = \Delta_{c,\mathbf{p}}\, \delta\lambda_{c,\mathbf{p}}(t)$ and $\Delta_{c,\mathbf{p}} = \hat{x}_c^{\text{src}}(\mathbf{p}) - x_{0,c}(\mathbf{p})$.

Without loss of generality we replace $\eta$ by $\eta_\varepsilon(t) = (1-\varepsilon)\eta(t) + \varepsilon t$ with fixed but arbitrarily small $\varepsilon > 0$, which makes $\eta_\varepsilon' \geq \varepsilon$ a.e. while $\mathcal{E}[\bar{d}_\varepsilon] \to \mathcal{E}[\bar{d}]$ as $\varepsilon \downarrow 0$; we thus drop the subscript in what follows.

**Step 1: Existence of a descent direction in $\mathcal{C}_{\text{pix}}$** By assumption (i)–(ii), there is a subset $\mathcal{S} \times I$ with positive measure where the local geometry is heterogeneous and $\Delta_{c,\mathbf{p}} \neq 0$. For $(c, \mathbf{p}) \in \mathcal{S}$, define

$$G_{c,\mathbf{p}}(t) := \partial_d U_{c,\mathbf{p}}\big(\bar{d}_{c,\mathbf{p}}(t), t\big)\, \Delta_{c,\mathbf{p}}.$$

If $G_{c,\mathbf{p}}(t) \not\equiv 0$ on $I$, choose $\delta\lambda_{c,\mathbf{p}}(t) = -\varepsilon\, \text{sgn}\big(G_{c,\mathbf{p}}(t)\big)\, \psi(t)$, where $\psi \in C_0^1(I)$ is a nonnegative bump (compactly supported in $I$), and $\varepsilon > 0$. For $(c, \mathbf{p}) \notin \mathcal{S}$, set $\delta\lambda_{c,\mathbf{p}} \equiv 0$.

Then the first-order variation satisfies

$$\delta\mathcal{E}_{\text{pot}} := \int_0^1 \sum_{c,\mathbf{p}} \partial_d U_{c,\mathbf{p}}\big(\bar{d}_{c,\mathbf{p}}, t\big)\, \delta d_{c,\mathbf{p}}(t)\, dt = -\varepsilon \int_I \sum_{(c,\mathbf{p}) \in \mathcal{S}} |G_{c,\mathbf{p}}(t)|\, \psi(t)\, dt \; < \; 0.$$

The kinetic part is

$$\delta\mathcal{E}_{\text{kin}} := \int_0^1 \sum_{c,\mathbf{p}} 2\, a_{c,\mathbf{p}}(t)\, \dot{\bar{d}}_{c,\mathbf{p}}(t)\, \delta\dot{d}_{c,\mathbf{p}}(t)\, dt = \int_I \sum_{(c,\mathbf{p}) \in \mathcal{S}} 2\, a_{c,\mathbf{p}}(t)\, \dot{\bar{d}}_{c,\mathbf{p}}(t)\, \Delta_{c,\mathbf{p}}\, \delta\dot{\lambda}_{c,\mathbf{p}}(t)\, dt.$$

By choosing $\psi$ piecewise-flat on most of $I$ and with very gentle ramps near its endpoints, we can make $\|\delta\dot{\lambda}\|_{L^1(I)}$ arbitrarily small while keeping $\int_I \psi(t)\, dt$ fixed. With $a_{c,\mathbf{p}}$ and $\dot{\bar{d}}_{c,\mathbf{p}}$ bounded (assumption (i)–(iii) imply admissibility and boundedness), we obtain

$$|\delta\mathcal{E}_{\text{kin}}| \; \leq \; C\,\varepsilon\, \|\delta\dot{\lambda}\|_{L^1(I)} \; \to \; 0 \quad \text{as the ramps are flattened.}$$

Hence for sufficiently small $\varepsilon$ and sufficiently flat ramps, the total first-order variation $\delta\mathcal{E}[\bar{d}; \delta\Lambda] = \delta\mathcal{E}_{\text{pot}} + \delta\mathcal{E}_{\text{kin}} < 0$, i.e., there exists an admissible perturbation that strictly reduces the energy.

**Step 2: Preservation of endpoint and monotone constraints** Let $\eta$ be monotone and satisfy the endpoint constraints. Because our perturbation is compactly supported in $I \subset (0, 1)$ and can be chosen with arbitrarily small $L^\infty$-norm, there exists $\varepsilon_0 > 0$ such that for all $0 < \varepsilon < \varepsilon_0$ the perturbed schedule $\tilde{\Lambda}(t) = \bar{\Lambda}(t) + \delta\Lambda(t)$ remains within $(0, 1)$, preserves endpoints, and is monotone. Therefore $\tilde{\Lambda} \in \mathcal{C}_{\text{pix}}$.

**Step 3: Strict domination** For any $\bar{\Lambda} \in \mathcal{C}_{\text{glob}}$, Steps 1–2 yield an admissible $\tilde{\Lambda} \in \mathcal{C}_{\text{pix}}$ with $\mathcal{E}[\tilde{d}] < \mathcal{E}[\bar{d}]$. Taking the infimum over all $\bar{\Lambda} \in \mathcal{C}_{\text{glob}}$ proves

$$\inf_{\Lambda \in \mathcal{C}_{\text{pix}}} \mathcal{E}[d] \; \leq \; \inf_{\Lambda \in \mathcal{C}_{\text{glob}}} \mathcal{E}[d] \quad \text{and in fact} \quad \inf_{\Lambda \in \mathcal{C}_{\text{pix}}} \mathcal{E}[d] \; < \; \inf_{\Lambda \in \mathcal{C}_{\text{glob}}} \mathcal{E}[d],$$

since heterogeneity (assumption (i)) and nondegenerate contrast (assumption (ii)) ensure that $G_{c,\mathbf{p}}$ is not identically zero on a set of positive measure and thus the descent is strict. $\qquad\square$

**Remark** If, additionally, $U_{c,\mathbf{p}}(\cdot, t)$ is $m_{c,\mathbf{p}}(t)$-strongly convex (Hessian $\succeq m_{c,\mathbf{p}}(t)\mathbf{I}$ on $I$), then for sufficiently smooth perturbations one obtains the second-order estimate

$$\mathcal{E}[\tilde{d}] - \mathcal{E}[\bar{d}] \; \leq \; -\varepsilon \int_I \sum_{(c,\mathbf{p}) \in \mathcal{S}} |G_{c,\mathbf{p}}(t)|\, \psi(t)\, dt \; + \; C \int_I \sum_{(c,\mathbf{p}) \in \mathcal{S}} a_{c,\mathbf{p}}(t)\, \Delta_{c,\mathbf{p}}^2 \big(\delta\dot{\lambda}_{c,\mathbf{p}}(t)\big)^2 dt,$$

so by flattening ramps the kinetic term can be made arbitrarily small while the potential decrease stays linear in $\varepsilon$, yielding a quantitative energy gap.

In the language of functional analysis, $\mathcal{E}$ is an integral functional defined on a space of admissible paths, assigning to each trajectory $\boldsymbol{d} : [0,1] \to X$ a real-valued energy via

$$\mathcal{E}[\boldsymbol{d}] = \int_0^1 L(\boldsymbol{d}(t), \dot{\boldsymbol{d}}(t), t)\, dt,$$

and minimizing $\mathcal{E}$ characterizes the optimal (least-cost) transition path.

### 6.3 DERIVATION OF THE ONE–STEP TRANSITION

If the forward marginals satisfy equation 6 for $t-1$ and $t$, then the conditional in equation 7 holds with $\rho_t = \sqrt{\bar{\alpha}_t / \bar{\alpha}_{t-1}}$.

**Proof** Assume $q(\boldsymbol{x}_{t-1} \mid \boldsymbol{x}_0, \hat{\boldsymbol{x}}_0) = \mathcal{N}\big(\sqrt{\bar{\alpha}_{t-1}}\, \boldsymbol{d}_{t-1},\, \sigma_{t-1}^2 \boldsymbol{I}\big)$ with $\sigma_{t-1}^2 = 1 - \bar{\alpha}_{t-1}$. Define the affine–Gaussian update

$$\boldsymbol{x}_t = \rho_t\, \boldsymbol{x}_{t-1} + \sqrt{\bar{\alpha}_t}\,\big(\boldsymbol{d}_t - \boldsymbol{d}_{t-1}\big) + \boldsymbol{\epsilon}_t, \qquad \boldsymbol{\epsilon}_t \sim \mathcal{N}\big(\boldsymbol{0},\, (1-\rho_t^2)\boldsymbol{I}\big), \tag{15}$$

with $\boldsymbol{\epsilon}_t$ independent of $\boldsymbol{x}_{t-1}$. Taking conditional expectation with respect to $(\boldsymbol{x}_0, \hat{\boldsymbol{x}}_0)$ yields

$$\mathbb{E}[\boldsymbol{x}_t \mid \boldsymbol{x}_0, \hat{\boldsymbol{x}}_0] = \rho_t\, \mathbb{E}[\boldsymbol{x}_{t-1} \mid \boldsymbol{x}_0, \hat{\boldsymbol{x}}_0] + \sqrt{\bar{\alpha}_t}\,(\boldsymbol{d}_t - \boldsymbol{d}_{t-1}) = \rho_t\, \sqrt{\bar{\alpha}_{t-1}}\, \boldsymbol{d}_{t-1} + \sqrt{\bar{\alpha}_t}\,(\boldsymbol{d}_t - \boldsymbol{d}_{t-1}).$$

Using $\rho_t = \sqrt{\bar{\alpha}_t / \bar{\alpha}_{t-1}}$ gives $\rho_t \sqrt{\bar{\alpha}_{t-1}} = \sqrt{\bar{\alpha}_t}$, hence

$$\mathbb{E}[\boldsymbol{x}_t \mid \boldsymbol{x}_0, \hat{\boldsymbol{x}}_0] = \sqrt{\bar{\alpha}_t}\, \boldsymbol{d}_{t-1} + \sqrt{\bar{\alpha}_t}\,(\boldsymbol{d}_t - \boldsymbol{d}_{t-1}) = \sqrt{\bar{\alpha}_t}\, \boldsymbol{d}_t,$$

which matches the mean in equation 6. For the covariance $\mathrm{Var}[\boldsymbol{x}_t \mid \boldsymbol{x}_0, \hat{\boldsymbol{x}}_0] = \rho_t^2$,

$$\mathrm{Var}[\boldsymbol{x}_{t-1} \mid \boldsymbol{x}_0, \hat{\boldsymbol{x}}_0] + (1-\rho_t^2)\boldsymbol{I} = \rho_t^2(1-\bar{\alpha}_{t-1})\boldsymbol{I} + (1-\rho_t^2)\boldsymbol{I} = \big(1 - \rho_t^2 \bar{\alpha}_{t-1}\big)\boldsymbol{I}.$$

Since $\rho_t^2 \bar{\alpha}_{t-1} = \bar{\alpha}_t$, the variance equals $(1 - \bar{\alpha}_t)\boldsymbol{I} = \sigma_t^2 \boldsymbol{I}$, which matches equation 6. Therefore the conditional distribution of $\boldsymbol{x}_t$ given $\boldsymbol{x}_{t-1}$ and $(\boldsymbol{x}_0, \hat{\boldsymbol{x}}_0)$ is Gaussian with mean and covariance in equation 7, completing the derivation. $\square$

### 6.4 DERIVATION OF EQ. 8

Discretized equation 8 with step $\Delta t$:

$$\boldsymbol{x}_{t+\Delta t} = \big(1 + f(t)\Delta t\big)\boldsymbol{x}_t + \sqrt{\bar{\alpha}_t}\, \dot{\Lambda}(t) \odot \big(\hat{\boldsymbol{x}}_0^{\mathrm{src}} - \boldsymbol{x}_0\big)\Delta t + g(t)\sqrt{\Delta t}\, \boldsymbol{z}_1, \quad \boldsymbol{z}_1 \sim \mathcal{N}(\boldsymbol{0}, \boldsymbol{I}). \tag{16}$$

Assume the inductive marginal at time $t$ equals $q(\boldsymbol{x}_t \mid \boldsymbol{x}_0, \hat{\boldsymbol{x}}_0^{\mathrm{src}}) = \mathcal{N}\big(\boldsymbol{x}_t;\, \sqrt{\bar{\alpha}_t}\, \boldsymbol{d}(t),\, \sigma_t^2 \boldsymbol{I}\big)$, $\quad t \in [0, T]$, i.e.,

$$\boldsymbol{x}_t = \sqrt{\bar{\alpha}_t}\, \boldsymbol{d}(t) + \sigma_t \boldsymbol{z}_2, \qquad \boldsymbol{z}_2 \sim \mathcal{N}(\boldsymbol{0}, \boldsymbol{I}). \tag{17}$$

Substitution yields

$$\boldsymbol{x}_{t+\Delta t} = \sqrt{\bar{\alpha}_t}\big(1 + f(t)\Delta t\big)\boldsymbol{d}(t) + \sqrt{\bar{\alpha}_t}\, \dot{\Lambda}(t) \odot \big(\hat{\boldsymbol{x}}_0^{\mathrm{src}} - \boldsymbol{x}_0\big)\Delta t + \sqrt{(1 + f(t)\Delta t)^2 \sigma_t^2 + g(t)^2 \Delta t}\, \tilde{\boldsymbol{z}}, \tag{18}$$

with $\tilde{\boldsymbol{z}} \sim \mathcal{N}(\boldsymbol{0}, \boldsymbol{I})$. On the other hand, the target marginal at $t + \Delta t$ is

$$\boldsymbol{x}_{t+\Delta t} = \sqrt{\bar{\alpha}_{t+\Delta t}}\, \boldsymbol{d}(t+\Delta t) + \sigma_{t+\Delta t}\, \boldsymbol{\epsilon}, \qquad \boldsymbol{\epsilon} \sim \mathcal{N}(\boldsymbol{0}, \boldsymbol{I}). \tag{19}$$

Using first–order expansions $\sqrt{\bar{\alpha}_{t+\Delta t}} = \sqrt{\bar{\alpha}_t} + \frac{d}{dt}\sqrt{\bar{\alpha}_t}\,\Delta t$ and $\boldsymbol{d}(t+\Delta t) = \boldsymbol{d}(t) + \dot{\Lambda}(t) \odot (\hat{\boldsymbol{x}}_0^{\mathrm{src}} - \boldsymbol{x}_0)\Delta t$, equating the mean terms gives

$$\sqrt{\bar{\alpha}_t}\, f(t)\, \boldsymbol{d}(t) = \frac{d}{dt}\sqrt{\bar{\alpha}_t}\, \boldsymbol{d}(t), \quad \Rightarrow \quad f(t) = \frac{d}{dt} \ln \sqrt{\bar{\alpha}_t}.$$

Equating variances yields

$$\sigma_{t+\Delta t}^2 = (1 + f(t)\Delta t)^2 \sigma_t^2 + g(t)^2 \Delta t = \sigma_t^2 + \big(2f(t)\sigma_t^2 + g(t)^2\big)\Delta t + o(\Delta t),$$

hence, in the limit $\Delta t \to 0$,

$$\frac{d\sigma_t^2}{dt} = 2f(t)\sigma_t^2 + g(t)^2 \quad \Rightarrow \quad g(t) = \sqrt{\frac{d\sigma_t^2}{dt} - 2\sigma_t^2 f(t)}.$$

## 6.5 PROOF OF THE FORWARD SDE

**Lemma 1.** *Let $d_t$ be defined by the domain mixture equation 1: $d_t = \Lambda_t \odot \hat{x}_0^{\mathrm{src}} + (1 - \Lambda_t) \odot x_0$. Then, for all $t \geq 1$,*

$$d_t - d_{t-1} = (\Lambda_t - \Lambda_{t-1}) \odot (\hat{x}_0^{\mathrm{src}} - x_0).$$

*Proof. Expand both $d_t$ and $d_{t-1}$ from equation 1 and collect terms.* $\square$

**Proposition 2** (Consistency of the forward marginal). *Fix a variance–preserving schedule $\{\alpha_t\}_{t=1}^T \subset (0,1)$ with $\bar{\alpha}_t = \prod_{s=1}^t \alpha_s$ and $\rho_t = \sqrt{\bar{\alpha}_t / \bar{\alpha}_{t-1}} = \sqrt{\alpha_t}$. Assume $\Lambda_0 \equiv 0$ (hence $d_0 = x_0$). Consider the one–step Markov transition equation 7:*

$$q(x_t \mid x_{t-1}, x_0, \hat{x}_0^{\mathrm{src}}) = \mathcal{N}\Big(x_t; \rho_t x_{t-1} + \sqrt{\bar{\alpha}_t}(d_t - d_{t-1}), (1 - \rho_t^2)I\Big), \qquad t = 1, \dots, T,$$

*where $d_t$ is given by equation 1. Then for every $t \in \{1, \dots, T\}$ the marginal distribution conditional on $(x_0, \hat{x}_0^{\mathrm{src}})$ equals*

$$q(x_t \mid x_0, \hat{x}_0^{\mathrm{src}}) = \mathcal{N}(x_t; \sqrt{\bar{\alpha}_t} d_t, (1 - \bar{\alpha}_t)I), \tag{20}$$

*i.e., the mean is $\sqrt{\bar{\alpha}_t} d_t$ and the covariance is $(1 - \bar{\alpha}_t)I$.*

*Proof.* We proceed by induction on $t$.

**Base case** ($t$=1)**.** Using $\Lambda_0 \equiv 0$ gives $d_0 = x_0$ and, by equation 7,

$$\mathbb{E}[x_1 \mid x_0, \hat{x}_0^{\mathrm{src}}] = \rho_1 x_0 + \sqrt{\bar{\alpha}_1}(d_1 - d_0) = \sqrt{\bar{\alpha}_1} d_1,$$

since $\rho_1 = \sqrt{\alpha_1} = \sqrt{\bar{\alpha}_1}$. Moreover, $\mathrm{Var}[x_1 \mid x_0, \hat{x}_0^{\mathrm{src}}] = (1 - \rho_1^2)I = (1 - \bar{\alpha}_1)I$. Thus equation 20 holds for $t = 1$.

**Inductive step.** Assume that for some $t \geq 2$ the claim holds at $t - 1$:

$$x_{t-1} \mid (x_0, \hat{x}_0^{\mathrm{src}}) \sim \mathcal{N}(\sqrt{\bar{\alpha}_{t-1}} d_{t-1}, (1 - \bar{\alpha}_{t-1})I).$$

Taking conditional expectation of equation 7 and using independence of the Gaussian noise, we get

$$\mathbb{E}[x_t \mid x_0, \hat{x}_0^{\mathrm{src}}] = \rho_t \mathbb{E}[x_{t-1} \mid x_0, \hat{x}_0^{\mathrm{src}}] + \sqrt{\bar{\alpha}_t}(d_t - d_{t-1}) = \rho_t \sqrt{\bar{\alpha}_{t-1}} d_{t-1} + \sqrt{\bar{\alpha}_t}(d_t - d_{t-1}).$$

Since $\rho_t \sqrt{\bar{\alpha}_{t-1}} = \sqrt{\alpha_t} \sqrt{\bar{\alpha}_{t-1}} = \sqrt{\bar{\alpha}_t}$, the two terms telescope:

$$\mathbb{E}[x_t \mid x_0, \hat{x}_0^{\mathrm{src}}] = \sqrt{\bar{\alpha}_t} d_{t-1} + \sqrt{\bar{\alpha}_t}(d_t - d_{t-1}) = \sqrt{\bar{\alpha}_t} d_t.$$

For the covariance,

$$\mathrm{Var}[x_t \mid x_0, \hat{x}_0^{\mathrm{src}}] = \rho_t^2 \mathrm{Var}[x_{t-1} \mid x_0, \hat{x}_0^{\mathrm{src}}] + (1 - \rho_t^2)I = \rho_t^2(1 - \bar{\alpha}_{t-1})I + (1 - \rho_t^2)I = (1 - \bar{\alpha}_t)I,$$

again using $\rho_t^2 \bar{\alpha}_{t-1} = \bar{\alpha}_t$. Thus equation 20 holds at time $t$, closing the induction. $\square$

**Corollary 1.** *If $\Lambda_t \equiv 1$ for all $t \geq t_1$, then for every $t \geq t_1$ one has $d_t = \hat{x}_0^{\mathrm{src}}$ and hence*

$$q(x_t \mid x_0, \hat{x}_0^{\mathrm{src}}) = \mathcal{N}(x_t; \sqrt{\bar{\alpha}_t} \hat{x}_0^{\mathrm{src}}, (1 - \bar{\alpha}_t)I).$$

*In particular, sampling can be initialized at $t_1$ with $x_{t_1} \sim \mathcal{N}(\sqrt{\bar{\alpha}_{t_1}} \hat{x}_0^{\mathrm{src}}, (1 - \bar{\alpha}_{t_1})I)$.*

**Remark** Let $\alpha_t = 1 - \beta(t)\Delta t$ with $\Delta t \to 0$. Writing $\bar{\alpha}(t) = \exp(-\int_0^t \beta(s) \, ds)$ and $d(t)$ as the continuous interpolation of $d_t$, the discrete kernel equation 7 converges to the VP forward SDE with a moving mean,

$$dx_t = \underbrace{-\tfrac{1}{2}\beta(t) x_t}_{f(t)x_t} dt + \underbrace{\sqrt{\bar{\alpha}(t)} \dot{d}(t)}_{\text{mean drift due to mixer}} dt + \underbrace{\sqrt{\beta(t)}}_{g(t)} dw_t,$$

which is the continuous counterpart used to derive the reverse-time SDE in the main text.

## 6.6 DERIVATION OF EQ. (10)

The data prediction model $\boldsymbol{\Phi}_\theta(\boldsymbol{x}_t, \hat{\boldsymbol{x}}_0^{\mathrm{src}}, t)$ approximates the clean target image, $\boldsymbol{\Phi}_\theta \approx \boldsymbol{x}_0$. From the forward marginal (Eq. 6) under the dynamic mixture,

$$q_t\big(\boldsymbol{x}_t \mid \boldsymbol{x}_0, \hat{\boldsymbol{x}}_0^{\mathrm{src}}\big) = \mathcal{N}\big(\boldsymbol{x}_t; \sqrt{\bar{\alpha}_t}\, \boldsymbol{d}_t,\ \sigma_t^2 \boldsymbol{I}\big), \quad \boldsymbol{d}_t = \Lambda_t \odot \hat{\boldsymbol{x}}_0^{\mathrm{src}} + (\boldsymbol{1} - \Lambda_t) \odot \boldsymbol{x}_0,$$

so the score w.r.t. $\boldsymbol{x}_t$ is

$$\nabla_{\boldsymbol{x}} \log q_t\big(\boldsymbol{x}_t \mid \boldsymbol{x}_0, \hat{\boldsymbol{x}}_0^{\mathrm{src}}\big) = -\frac{\boldsymbol{x}_t - \sqrt{\bar{\alpha}_t}\boldsymbol{d}_t}{\sigma_t^2}.$$

Replacing $\boldsymbol{x}_0$ by $\boldsymbol{\Phi}_\theta(\boldsymbol{x}_t, \hat{\boldsymbol{x}}_0^{\mathrm{src}}, t)$ yields the plug-in estimator

$$\nabla_{\boldsymbol{x}} \log q_t\big(\boldsymbol{x}_t \mid \hat{\boldsymbol{x}}_0^{\mathrm{src}}\big) \approx -\frac{\boldsymbol{x}_t - \sqrt{\bar{\alpha}_t}\boldsymbol{d}_t^\theta}{\sigma_t^2}, \quad \boldsymbol{d}_t^\theta := \Lambda_t \odot \hat{\boldsymbol{x}}_0^{\mathrm{src}} + \big(\boldsymbol{1} - \Lambda_t\big) \odot \boldsymbol{\Phi}_\theta(\boldsymbol{x}_t, \hat{\boldsymbol{x}}_0^{\mathrm{src}}, t),$$

which is Eq. (10). Moreover, $\varepsilon_\theta = -\sigma_t \nabla_{\boldsymbol{x}} \log q_t(\boldsymbol{x}_t \mid \hat{\boldsymbol{x}}_0^{\mathrm{src}}) = \frac{\boldsymbol{x}_t - \sqrt{\bar{\alpha}_t}\boldsymbol{d}_t^\theta}{\sigma_t}$, giving Eq. 11. $\qquad\square$

## 6.7 PROOF OF PROPOSITION 1

Recall the reparameterization $\boldsymbol{\Upsilon}_t = \alpha_t(\boldsymbol{1} - \Lambda_t)$, $\boldsymbol{\lambda}_t = \sigma_t \oslash \boldsymbol{\Upsilon}_t$, and define $\boldsymbol{y}_t := \boldsymbol{x}_t \oslash \boldsymbol{\Upsilon}_t$. From Eq. 10 and the derivation of Eq. 12, the reverse dynamics with respect to $\boldsymbol{\lambda}$ takes the diagonal linear SDE form

$$d\boldsymbol{y}_{\boldsymbol{\lambda}} = \Big(\tfrac{2}{\boldsymbol{\lambda}}\Big) \odot \boldsymbol{y}_{\boldsymbol{\lambda}}\, d\boldsymbol{\lambda} \ + \ \Big[\big(\boldsymbol{1} \oslash (\boldsymbol{1} - \Lambda_{\boldsymbol{\lambda}})^2\big) \odot d\Lambda_{\boldsymbol{\lambda}} \ - \ \big(\Lambda_{\boldsymbol{\lambda}} \oslash (\boldsymbol{1} - \Lambda_{\boldsymbol{\lambda}})\big) \odot \Big(\tfrac{2}{\boldsymbol{\lambda}}\Big) d\boldsymbol{\lambda}\Big] \odot \hat{\boldsymbol{x}}_0^{\mathrm{src}} \tag{21}$$

$$- \ \Big(\tfrac{2}{\boldsymbol{\lambda}}\Big) \odot \boldsymbol{\Phi}_\theta(\boldsymbol{x}_{\boldsymbol{\lambda}}, \hat{\boldsymbol{x}}_0^{\mathrm{src}}, \boldsymbol{\lambda})\, d\boldsymbol{\lambda} \ + \ \sqrt{2\boldsymbol{\lambda}} \odot d\boldsymbol{w}_{\boldsymbol{\lambda}}, \tag{22}$$

.

Fix any index $i = (c, \mathbf{p})$. The $i$-th component of equation 21 is a scalar linear SDE

$$dy_i = \frac{2}{\lambda_i} y_i\, d\lambda_i + \Big(\frac{1}{(1 - \Lambda_i)^2} d\Lambda_i - \frac{\Lambda_i}{1 - \Lambda_i} \frac{2}{\lambda_i} d\lambda_i\Big) \hat{x}_{0,i}^{\mathrm{src}} - \frac{2}{\lambda_i} x_{\theta,i}(\cdot)\, d\lambda_i + \sqrt{2\lambda_i}\, dw_{\lambda_i}.$$

By the variation-of-constants formula with integrating factor $\exp\big(\int (2/\lambda_i)\, d\lambda_i\big) = \lambda_i^2$, we obtain for any $t \in [0, s]$:

$$y_i(t) = \frac{\lambda_i(t)^2}{\lambda_i(s)^2} y_i(s) + \int_{\lambda_i(s)}^{\lambda_i(t)} \frac{\lambda_i(t)^2}{\tau^2} \Big(\frac{1}{(1 - \Lambda_i)^2} d\Lambda_i - \frac{\Lambda_i}{1 - \Lambda_i} \frac{2}{\tau} d\tau\Big) \hat{x}_{0,i}^{\mathrm{src}}$$

$$- \int_{\lambda_i(s)}^{\lambda_i(t)} \frac{\lambda_i(t)^2}{\tau^3} 2\, x_{\theta,i}(\cdot)\, d\tau + \int_{\lambda_i(s)}^{\lambda_i(t)} \frac{\lambda_i(t)^2}{\tau^2} \sqrt{2\tau}\, dw_\tau.$$

The last Itô integral is Gaussian with zero mean and variance $\lambda_i(t)^2 - \lambda_i(t)^4/\lambda_i(s)^2$. Stacking all components and returning to $\boldsymbol{x}_t = \boldsymbol{\Upsilon}_t \odot \boldsymbol{y}_t$ yields Eq. 13, where all operations are Hadamard and $\boldsymbol{z}_s \sim \mathcal{N}(\boldsymbol{0}, \boldsymbol{I})$ is independent of $\boldsymbol{x}_s$.

## 6.8 REVERSE SDE

By the time-reversal formula for Itô diffusion with state-independent diffusion (Anderson, 1982), the reverse drift is $\bar{\boldsymbol{\mu}}(\boldsymbol{x}, t) = \boldsymbol{\mu}(\boldsymbol{x}, t) - \sigma^2(t) \nabla_{\boldsymbol{x}} \log q_t(\boldsymbol{x})$ with the same diffusion coefficient $\sigma(t)$. Substituting $\boldsymbol{\mu}$ and $\sigma(t) = g(t)$ yields exactly Eq. 9.

## 6.9 FIRST-ORDER SOLVER

**Notation.** We write $\odot$ for Hadamard multiplication and $\oslash$ for Hadamard division. Recall $\alpha_t = \sqrt{\bar{\alpha}_t}$, $\sigma_t^2 = 1 - \bar{\alpha}_t$, and define

$$\boldsymbol{\Upsilon}_t := \sqrt{\bar{\alpha}_t}\big(\boldsymbol{1} - \Lambda_t\big), \qquad \boldsymbol{\lambda}_t := \sigma_t \oslash \boldsymbol{\Upsilon}_t.$$

**Proposition 3.** *Given an initial value $\boldsymbol{x}_s$ at time $s > 0$, the solution $\boldsymbol{x}_t$ of the reverse-time (cf. Eq. 12 in the main text) for $t \in [0, s]$ is*

$$
\boldsymbol{x}_t = \underbrace{\boldsymbol{\Upsilon}_t \odot \left(\boldsymbol{\lambda}_t^2 \oslash \boldsymbol{\lambda}_s^2\right) \odot \left(\boldsymbol{x}_s \oslash \boldsymbol{\Upsilon}_s\right)}_{\textit{(A) state transport}}
$$

$$
+ \underbrace{\boldsymbol{\Upsilon}_t \odot \left[\left(\Lambda_t \oslash (\mathbf{1} - \Lambda_t)\right) - \left(\Lambda_s \oslash (\mathbf{1} - \Lambda_s)\right) \odot \left(\boldsymbol{\lambda}_t^2 \oslash \boldsymbol{\lambda}_s^2\right)\right] \odot \hat{\boldsymbol{x}}_0^{\mathrm{src}}}_{\textit{(B) domain-shift drift}}
\tag{23}
$$

$$
- \underbrace{\boldsymbol{\Upsilon}_t \odot \int_{\boldsymbol{\lambda}_s}^{\boldsymbol{\lambda}_t} \left(2\,\boldsymbol{\lambda}_t^2 \oslash \boldsymbol{\lambda}^3\right) \odot \boldsymbol{\Phi}_\theta(\boldsymbol{x}_{\boldsymbol{\lambda}}, \hat{\boldsymbol{x}}_0^{\mathrm{src}}, \boldsymbol{\lambda})\, d\boldsymbol{\lambda}}_{\textit{(C) network-guided denoising}}
$$

$$
+ \boldsymbol{\Upsilon}_t \odot \int_{\boldsymbol{\lambda}_s}^{\boldsymbol{\lambda}_t} \left(\boldsymbol{\lambda}_t^2 \oslash \boldsymbol{\lambda}^2\right) \odot \left(\tfrac{g}{\boldsymbol{\Upsilon}}\right) \oslash \sqrt{\tfrac{d\boldsymbol{\lambda}}{dt}}\, d\boldsymbol{w}_{\boldsymbol{\lambda}}.
$$

*Proof.* Starting from the time-changed reverse dynamics (main-text Eq. 12), introduce $\boldsymbol{y}_t := \boldsymbol{x}_t \oslash \boldsymbol{\Upsilon}_t$ so that each component $y_i$ satisfies a linear SDE

$$
dy_i = \frac{2}{\lambda_i} y_i\, d\lambda_i + \left(\frac{1}{(1 - \Lambda_i)^2} d\Lambda_i - \frac{\Lambda_i}{1 - \Lambda_i} \frac{2}{\lambda_i} d\lambda_i\right)\hat{x}_{0,i}^{\mathrm{src}} - \frac{2}{\lambda_i} x_{\theta,i}(\cdot)\, d\lambda_i + \frac{g_i}{\Upsilon_i} \frac{1}{\sqrt{d\lambda_i/dt}}\, dw_{\lambda_i}.
$$

Applying the variation-of-constants formula with integrating factor $\exp\int (2/\lambda_i)\, d\lambda_i = \lambda_i^2$, integrating from $s$ to $t$, and stacking all components, yields Eq. 23 after multiplying back by $\boldsymbol{\Upsilon}_t$. The Itô integral has zero mean and covariance

$$
\mathrm{Cov}\left[\int_{\lambda_i(s)}^{\lambda_i(t)} \frac{\lambda_i(t)^2}{\tau^2} \frac{g_i}{\Upsilon_i} \frac{1}{\sqrt{d\lambda_i/dt}}\, dw_\tau\right] = \int_{\lambda_i(s)}^{\lambda_i(t)} \frac{\lambda_i(t)^4}{\tau^4} \frac{g_i^2}{\Upsilon_i^2} \frac{1}{d\lambda_i/dt}\, d\tau,
$$

so the stochastic term remains in Itô integral form. $\square$

The only non-analytic piece in Eq. 23 is the integral in (C). We approximate it by Itô–Taylor expansion of order 0 at $\boldsymbol{\lambda}_s$:

$$
\boldsymbol{\Phi}_\theta\big(\boldsymbol{x}_{\boldsymbol{\lambda}}, \hat{\boldsymbol{x}}_0^{\mathrm{src}}, \boldsymbol{\lambda}\big) \approx \boldsymbol{\Phi}_\theta(\boldsymbol{x}_s, \hat{\boldsymbol{x}}_0^{\mathrm{src}}, s).
$$

Thus, componentwise,

$$
-\int_{\lambda_s}^{\lambda_t} \frac{2\lambda_t^2}{\lambda^3} x_\theta(\cdot)\, d\lambda \approx x_\theta(\cdot)\left(1 - \frac{\lambda_t^2}{\lambda_s^2}\right),
$$

which gives

$$
-\int_{\boldsymbol{\lambda}_s}^{\boldsymbol{\lambda}_t} \left(2\,\boldsymbol{\lambda}_t^2 \oslash \boldsymbol{\lambda}^3\right) \odot \boldsymbol{\Phi}_\theta(\cdot)\, d\boldsymbol{\lambda} \approx \left(\mathbf{1} - \boldsymbol{\lambda}_t^2 \oslash \boldsymbol{\lambda}_s^2\right) \odot \boldsymbol{\Phi}_\theta(\boldsymbol{x}_s, \hat{\boldsymbol{x}}_0^{\mathrm{src}}, s).
$$

Substituting into Eq. 23, we obtain the first-order solver:

$$
\tilde{\boldsymbol{x}}_t = \boldsymbol{\Upsilon}_t \odot \left(\boldsymbol{\lambda}_t^2 \oslash \boldsymbol{\lambda}_s^2\right) \odot \left(\boldsymbol{x}_s \oslash \boldsymbol{\Upsilon}_s\right)
$$

$$
+ \boldsymbol{\Upsilon}_t \odot \left[\left(\Lambda_t \oslash (\mathbf{1} - \Lambda_t)\right) - \left(\Lambda_s \oslash (\mathbf{1} - \Lambda_s)\right) \odot \left(\boldsymbol{\lambda}_t^2 \oslash \boldsymbol{\lambda}_s^2\right)\right] \odot \hat{\boldsymbol{x}}_0^{\mathrm{src}}
$$

$$
+ \boldsymbol{\Upsilon}_t \odot \left(\mathbf{1} - \boldsymbol{\lambda}_t^2 \oslash \boldsymbol{\lambda}_s^2\right) \odot \boldsymbol{\Phi}_\theta(\boldsymbol{x}_s, \hat{\boldsymbol{x}}_0^{\mathrm{src}}, s)
\tag{24}
$$

$$
+ \boldsymbol{\Upsilon}_t \odot \int_{\boldsymbol{\lambda}_s}^{\boldsymbol{\lambda}_t} \left(\boldsymbol{\lambda}_t^2 \oslash \boldsymbol{\lambda}^2\right) \odot \left(\tfrac{g}{\boldsymbol{\Upsilon}}\right) \oslash \sqrt{\tfrac{d\boldsymbol{\lambda}}{dt}}\, d\boldsymbol{w}_{\boldsymbol{\lambda}}
$$

## 6.10 DATASET DETAILS

**IXI (MRI T1→T2)** We use the multi-center IXI dataset (London et al., 2025), comprising scans from three hospitals and mixed vendors/field strengths (Philips 3T, Philips 1.5T, GE 1.5T). The task is directional T1→T2 mapping. We extract 2D axial slices from NIFTI volumes and form 500 training pairs and 77 test pairs. This setting primarily reflects minor contrast changes under strong structural alignment and is clinically relevant when partial protocols are acquired due to time constraints or motion artifacts.

**Sentinel-1/2 (SAR→Optical)** We use co-registered Sentinel-1 (SAR) and Sentinel-2 (optical) imagery (Schmitt et al., 2018). SAR provides structure that is robust to weather; optical captures color and texture. We curate temporally aligned 256×256 image pairs covering urban, agricultural, forest, and coastal scenes. The final split includes 1,200 training pairs and 300 test pairs. This task introduces larger appearance gaps and stronger cross-modal difficulty relative to IXI.

**PSCDE (EL-to-semantic defects)** PSCDE (Wang et al., 2024a) contains 700 high-resolution electroluminescence images of polycrystalline solar cells with expert annotations for seven defect types (e.g., linear cracks, finger interruptions, black cores). We use 420 images for training and 280 for testing with balanced category representation. This setting demands fine detail synthesis under significant acquisition variability, manufacturing differences, and heterogeneous defect morphology.

## 6.11 IMPLEMENTATION DETAILS

Experiments ran on a single NVIDIA A100 80 GB GPU on the cluster's AMD EPYC 7662 node (128 CPU cores, 512 GB RAM, InfiniBand EDR 100 Gb/s).

We train the model in PyTorch Lightning using single-node distributed data parallelism with mixed precision and a fixed random seed. The network includes a backbone (Lin et al., 2025) and a Stable-Diffusion UNet (Rombach et al., 2022) with base channel width 320, four resolution scales. Visual inputs are encoded into a four-channel latent space by a KL autoencoder with a $32\times$ spatial reduction.

The adaptive domain-mixing module $\mathcal{N}_\phi$ expands $\lambda_t$ into a spatial field via a lightweight convolutional network (channel progression $1 \to 8 \to 16 \to C$) augmented with sinusoidal position encodings. Its output is squashed by a calibrated logistic function so that every entry lies in $(0, 1)$.

For the reverse process we employ the first-order solver derived in Algorithm 3. At every step the sampler reconstructs the quantities $\Upsilon_t = \sqrt{\overline{\alpha}_t}(1 - \Lambda_t)$ and $\lambda_t = \sigma_t/\Upsilon_t$ and applies the analytic transport, domain-drift, denoising, and stochastic components to advance the state. Training minimizes the mean-squared error between the predicted and ground-truth latents.

Optimization uses AdamW with a base learning rate of $5 \times 10^{-5}$ for the latent model parameters, while the domain-mixing module receives a tenfold higher rate. Weight decay is set to 0.01, gradient accumulation is disabled, and exponential moving averages are not maintained. The diffusion horizon is fixed at $T = 1000$; for the PSCDE experiments the trainer executes 5,001 optimization steps, evaluates every 70 iterations, logs images every 500 steps, and saves checkpoints every 2,000 steps. Dataloader settings include batch size four, $256 \times 256$ center crops, four worker threads, and a prefetch factor of two. Before optimization begins we warm-start from a pre-trained checkpoint (Stability-AI, 2022) so that the dynamic domain mixer can fine-tune from a meaningful initialization.

For fair comparison across baselines (including BBDM and ABRIDGE), we adopt a consistent training budget within each dataset: all methods are trained for 20,000 steps on **Sentinel**, 10,000 steps on **IXI**, and 5,000 steps on **PSCDE**. Regarding sampling, because BBDM does not provide a fast-sampling variant, we evaluate it using its recommended default of **1,000** sampling steps.

## 6.12 BASELINES AND METRICS

We compare against Pix2Pix (Isola et al., 2017) (supervised cGAN with L1 reconstruction; effective on well-aligned pairs but prone to structural issues under complex gaps); BBDM (Li et al., 2023) (Brownian-bridge diffusion connecting source and target via stochastic paths; strong semantic alignment but high sampling cost and potential instability under large modality gaps); ABridge (Xiao

Table 3: PSCDE dataset: semantic translation metrics. Higher is better except Hausdorff.

| Method | Dice ↑ | IoU ↑ | Precision ↑ | Recall ↑ | Hausdorff ↓ | Skeleton F1 ↑ |
|---|---|---|---|---|---|---|
| Linear | $0.46 \pm 0.33$ | $0.37 \pm 0.34$ | $0.66 \pm 0.32$ | $0.41 \pm 0.34$ | $59.53 \pm 62.50$ | $0.59 \pm 0.33$ |
| Channel Non-linear | $0.46 \pm 0.36$ | $0.38 \pm 0.36$ | $0.78 \pm 0.25$ | $0.42 \pm 0.36$ | $43.03 \pm 53.31$ | $0.57 \pm 0.36$ |
| Fully Non-linear | $\mathbf{0.49 \pm 0.35}$ | $\mathbf{0.40 \pm 0.35}$ | $\mathbf{0.78 \pm 0.26}$ | $\mathbf{0.43 \pm 0.36}$ | $\mathbf{39.87 \pm 50.76}$ | $\mathbf{0.61 \pm 0.34}$ |

et al., 2025) (explicit semantic bridging constraints within diffusion; improved modality consistency but reduced performance on fine-scale structures and low-contrast data); and a fixed-schedule diffusion baseline aligned with DOSSR (Cui et al., 2025), adapted to cross-modal translation with the same UNet backbone and a linear domain interpolation schedule.

Metrics include SSIM and PSNR for structural similarity and signal quality, MSE and MAE for pixel-wise accuracy.

### 6.13 ADDITIONAL ABLATION DETAILS

To further study the effect of spatial adaptivity, we introduce a *per-channel, spatially uniform* variant of our schedule. In this setting, each channel receives a value $\lambda_c(t) \in \mathbb{R}$, producing a channel-wise vector $\lambda(t) \in \mathbb{R}^C$ that varies across channels but remains uniform across all spatial locations. The network therefore outputs a tensor of shape $(B, C, 1, 1)$, which is broadcast to $(B, C, H, W)$ during modulation. For comparison, the full method uses a *spatially varying* schedule in which the modulation depends on both channel and spatial position, yielding $\lambda(t) \in \mathbb{R}^{C \times H \times W}$ with entries $\lambda_c(t, p)$ at location $p$. The lambda network applies a per-channel polynomial for this channel-wise setting: $\lambda_c(t) = f_c(\eta(t)) = \sum_i a_{c,i} \, \eta(t)^i$, where $a_{c,i}$ are the learnable coefficients and $\eta(t)$ is the base linear schedule. The resulting tensor has shape $(B, C, 1, 1)$ and is broadcast to $(B, C, H, W)$.

As presented in Tab. 3, this per-channel uniform schedule attains a Dice score of $0.46$ and a Hausdorff distance of $43.0$ pixels. Compared to the fixed linear schedule (Dice $0.46$, Hausdorff $59.5$), we observe a substantial reduction in boundary error but no improvement in Dice, consistent with the high precision ($0.78$) and relatively low recall ($0.42$) of this variant. The skeleton- and connectivity-based metrics (skeleton F1 $0.57$, connectivity IoU $0.31$) indicate that channel-wise adaptation improves structural coherence and boundary sharpness, but it still falls short of the fully spatially varying model, which achieves both higher Dice and lower Hausdorff. These findings suggest that channel-wise modulation accounts for part of the geometric and connectivity gains, while spatial adaptivity provides an additional boost in overlap accuracy and global mask quality.

### 6.14 ROBUSTNESS ANALYSIS

To assess robustness to registration noise, we systematically misalign the PSCDE inputs by 0–8 px and re-evaluate the segmentation metrics (Fig. 4). Performance is essentially stable under small perturbations: at 0 px and 1 px shifts, Dice remains around $0.49$–$0.50$ and even slightly improves, while Hausdorff decreases from $51.7$ to $46.9$ pixels and skeleton F1 increases from $0.61$ to $0.63$. For larger misalignments (4–8 px), quality degrades gradually rather than catastrophically (Dice drops from $0.46$ to $0.43$, Hausdorff increases to $63.1$ pixels, and skeleton F1 remains above $0.55$), indicating that the proposed model is moderately robust to realistic registration errors and maintains coherent structural predictions even under non-trivial spatial perturbations.

### 6.15 ADDITIONAL RESULT VISUALIZATION

We present additional qualitative visualization on the testing dataset for the four different image translation tasks: Fig. 5 for Sentinel-1/2 (SAR→Optical) remote sensing application; Figs. 6 and 7 are for the MR T1 to T2 and T2 to T1 imaging application, respectively; Fig. 8 for Solar Panel Electroluminescence semantic masking application.

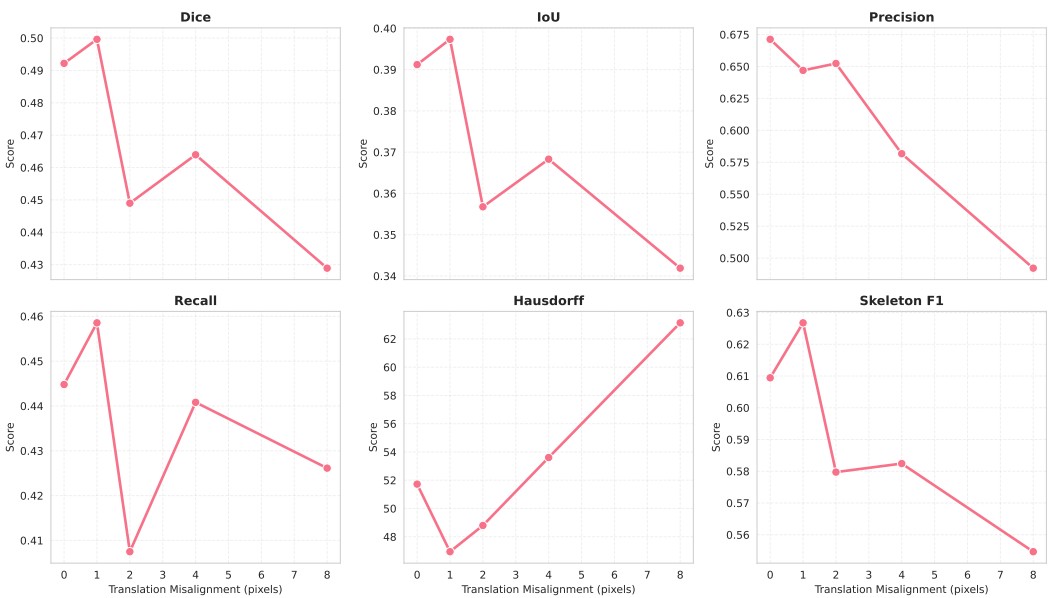

Figure 4: Additional assessment regarding the robustness to the registration error.

## 6.16 ALGORITHMS

We provide the algorithms for the core implementations of the proposed method. Algorithm 1 builds a *domain-shift guidance field* by spatially modulating a linear step schedule with a learned mixer $\mathcal{S}_\theta$. The result is a per-time, per-location mixing map $\Lambda_t(\mathbf{p})$ that blends the source latent $\hat{x}_0^{\mathrm{src}}$ and the target latent $x_0$ into a time-varying guidance signal $d_t$. Algorithm 2 then trains the diffusion network $\Phi_\theta$ via score matching.

Algorithm 3 performs reverse-time generation on the spaced grid: at each step it combines state transport, domain-shift drift, network-guided denoising, and stochastic noise—weighted by the schedule ratios and decode the final image.

---

**Algorithm 1** Domain-Shift Field Construction

---

**Require:** linear step schedule $\{\lambda_t^{\mathrm{lin}}\}_{t=0}^T$; source latent $\hat{x}_0^{\mathrm{src}}$; target latent $x_0$; spatial mixer $\mathcal{S}_\theta$; position encoding map $\pi$; clamp $\varepsilon = 10^{-4}$

1: **Pre-compute:** $\beta \leftarrow -\log\frac{\varepsilon}{1-\varepsilon}, \quad \alpha \leftarrow -2\beta$
2: **for** $t = 0$ **to** $T$ **do**
3:     **Broadcast base scalar:** $\lambda_t^{\mathrm{lin}}(\mathbf{p}) \leftarrow \lambda_t^{\mathrm{lin}}$ for all spatial $\mathbf{p}$
4:     **Spatial modulation:** $h_{t,c}(\mathbf{p}) \leftarrow \mathcal{S}_\theta\big(\lambda_t^{\mathrm{lin}}, \pi(\mathbf{p})\big)$
5:     **Zero-centered modulation:** $g_{t,c}(\mathbf{p}) \leftarrow 2\, h_{t,c}(\mathbf{p}) - 1$
6:     **Boundary-preserving interpolation:** $f_{t,c}(\mathbf{p}) \leftarrow \lambda_t^{\mathrm{lin}}\big[1 + g_{t,c}(\mathbf{p})\,(1 - \lambda_t^{\mathrm{lin}})\big]$
7:     **Logistic squashing:** $\Lambda_{t,c}(\mathbf{p}) \leftarrow \sigma\big(\alpha\,(f_{t,c}(\mathbf{p}) - \beta)\big)$, where $\sigma(z) = \dfrac{1}{1 + e^{-z}}$
8: **end for**
9: **Clamp endpoints:** $\Lambda_{0,c}(\mathbf{p}) \leftarrow 0, \quad \Lambda_{T,c}(\mathbf{p}) \leftarrow 1$
10: **Form domain mixture (Eq. (1)):** $d_t \leftarrow \Lambda_t \odot \hat{x}_0^{\mathrm{src}} + \big(\mathbf{1} - \Lambda_t\big) \odot x_0$
11: Provide $d_t$ to the diffusion sampler at step $t$ as the domain-shift guidance field.

---

## 6.17 ADDITIONAL DISCUSSION

**Positioning relative to diffusion bridge models:** The underlying generative mechanisms between the proposed method and Diffusion Bridge-based method for image translation are fundamentally different. Bridge-based methods explicitly redefine the forward process as a path conditioned on

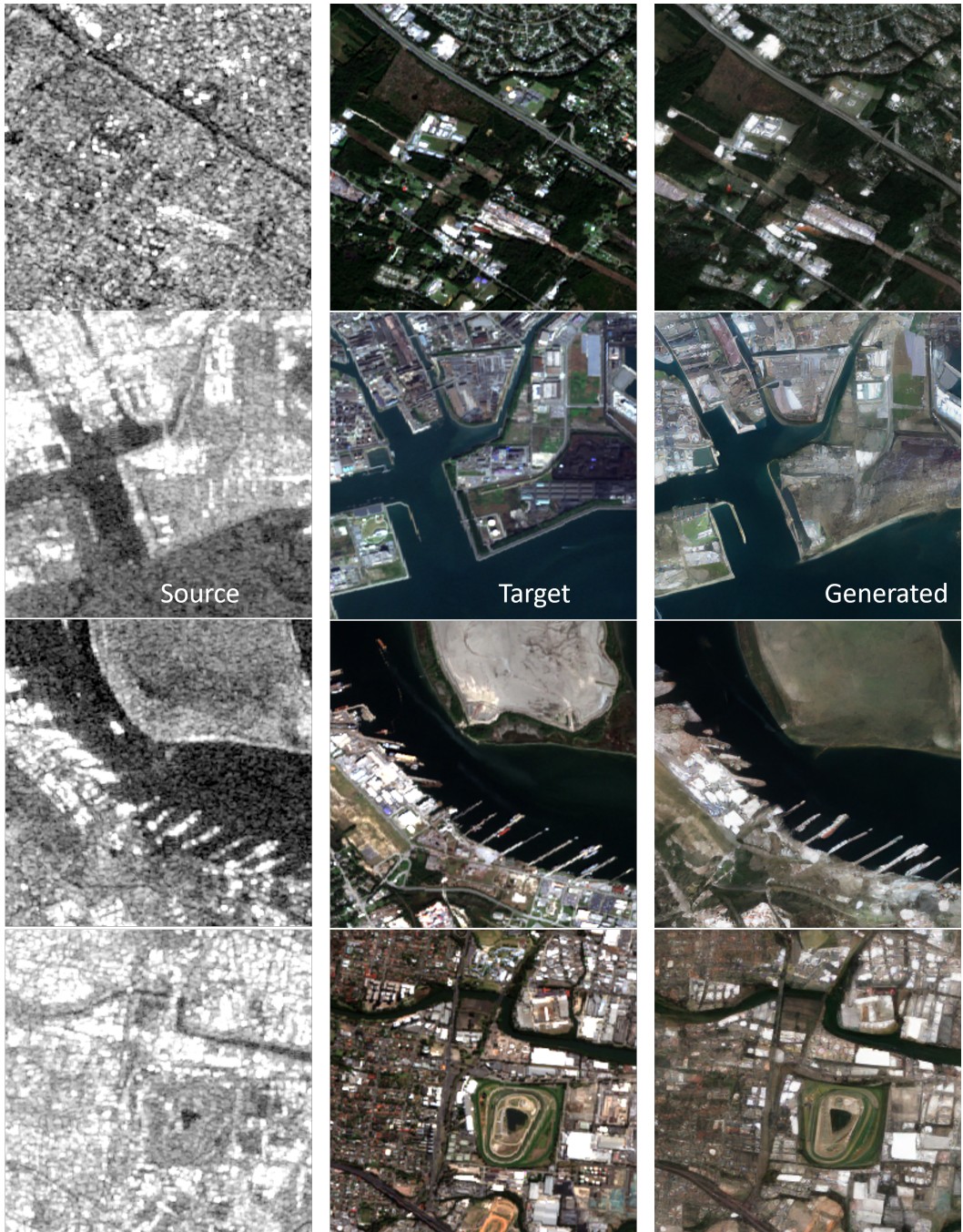

Figure 5: Visualization of the SAR to Optical image translation task; samples are randomly picked from the testing dataset.

both endpoints $(x_0, y)$, constructing a Doob-$h$-transform or related bridge kernel $q(x_t \mid x_0, y)$. Yet, our method retains the standard VP-type diffusion formulation and only modifies the drift by injecting an adaptive domain-shift term. This is very useful in practice because it operates directly on the VP-SDE backbone, they are structurally compatible with existing large-scale pretrained diffusion priors (e.g., latent diffusion models or domain-specific medical priors). In practice, this allows us to initialize from or distill into models that have already learned powerful generic image statistics, and then specialize them via a lightweight drift correction that accounts for the cross-domain gap. Bridge-based methods (He et al. (2024); Zheng et al. (2025); Zhou et al. (2023); Liu et al. (2023);

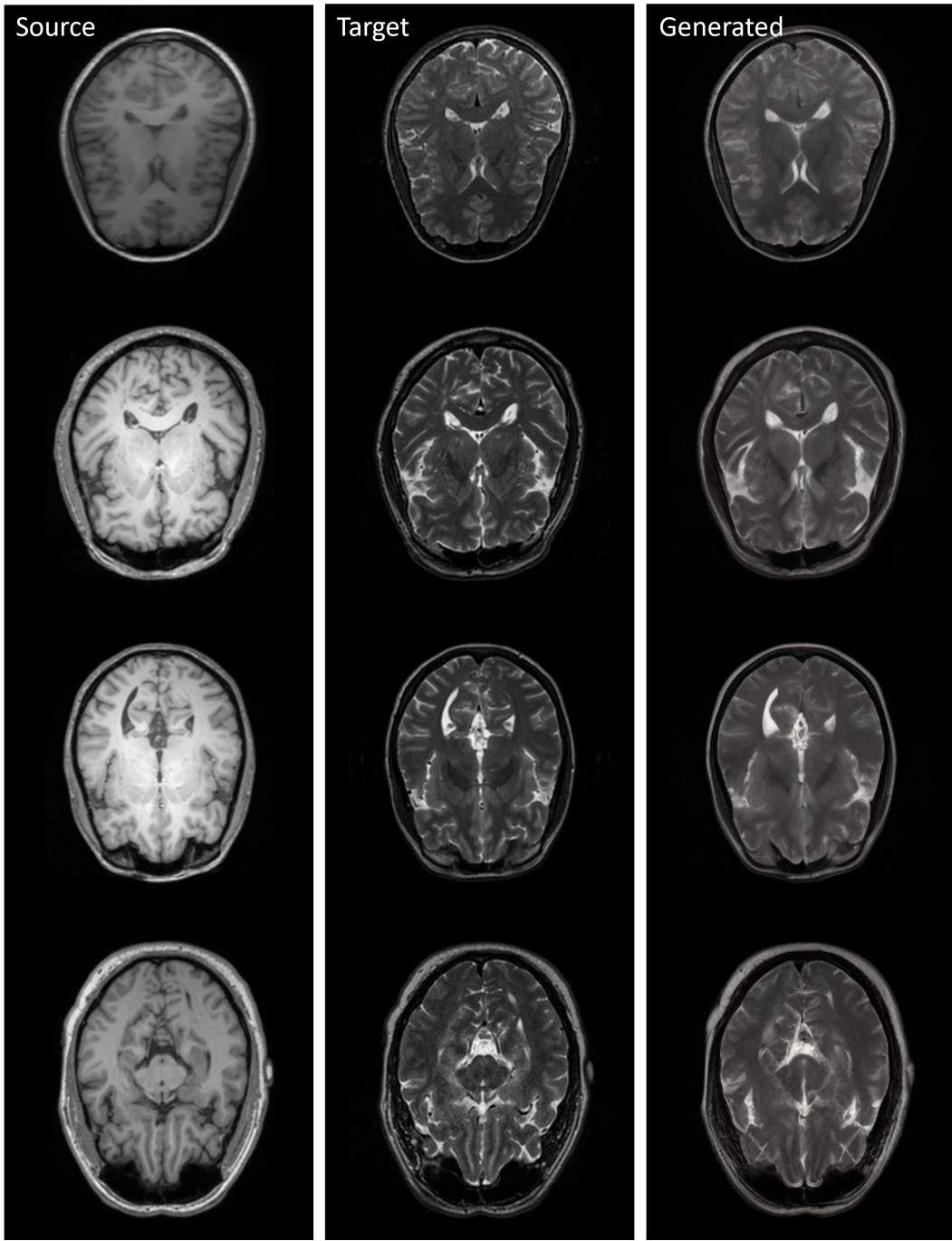

Figure 6: Visualization of the T1 to T2 MRI translation task; samples are randomly picked from the testing dataset.

Xiao et al. (2025); Li et al. (2023)) employ a different forward kernel and noise schedule tied to the bridge dynamics; this often requires training from scratch or re-deriving the entire sampling scheme, making it significantly harder to plug into the current diffusion ecosystem. And, diffusion bridge models need design solvers around a more intricate (Li et al. (2023)), endpoint-conditioned kernel; even with implicit or ODE reformulations, the resulting dynamics tend to be numerically heavier and less amenable to extreme step-size reduction under comparable quality targets.

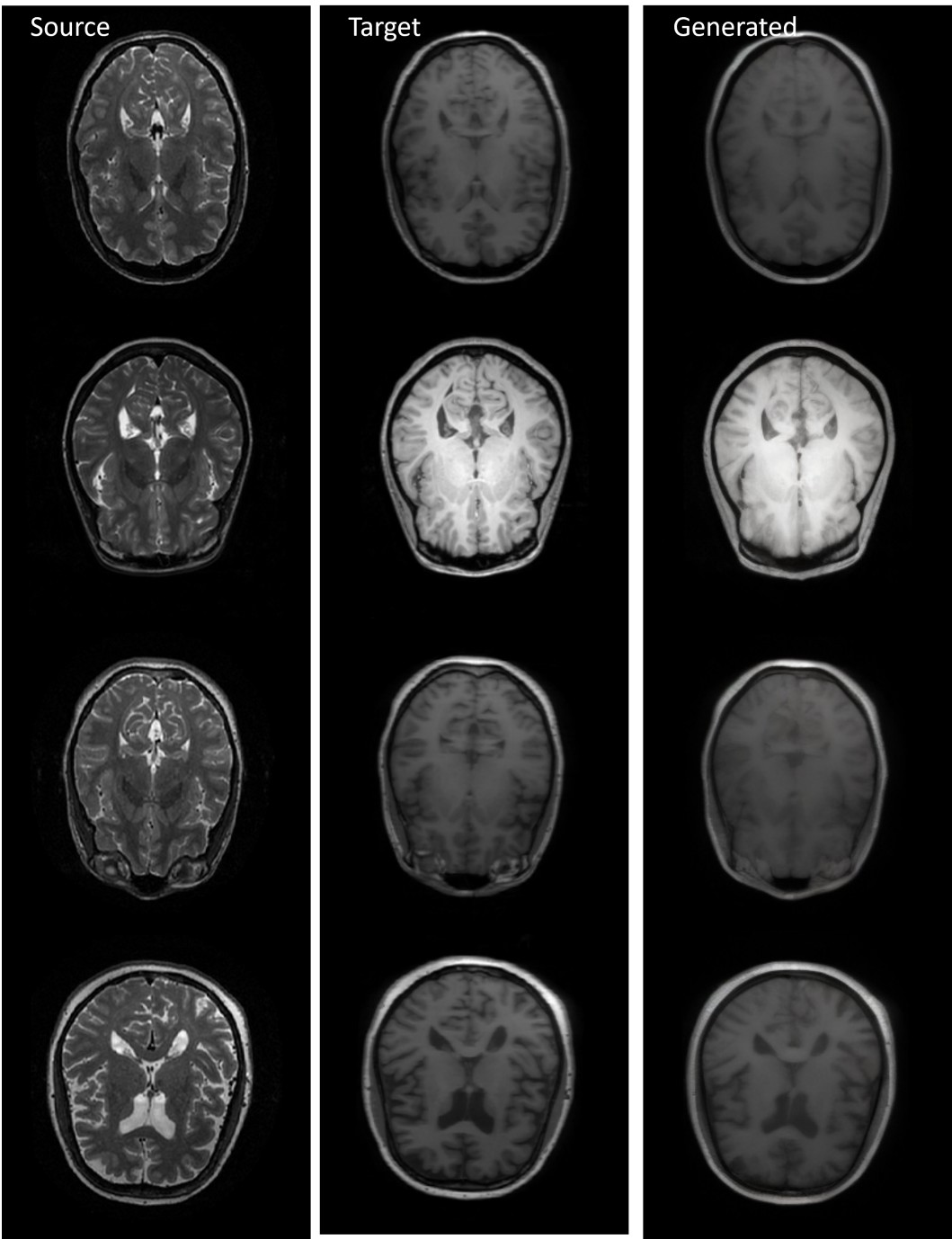

Figure 7: Visualization of the T2 to T1 MRI translation task; samples are randomly picked from the testing dataset.

**Regimes where CDTSDE provides marginal gains:**    As discussed in the main text, the three evaluated tasks (IXI (MRI contrast translation), Sentinel (SAR→optical), and PSCDE (EL→semantic defects)) form an approximate spectrum of increasing modality discrepancy and structural complexity. On IXI, where the contrast shift between T1 and T2 is relatively mild and the anatomical structures are highly aligned, the global linear schedule already traces a reasonable transition path between source and target domains. In this low-gap regime, the dynamic domain-mixture schedule has limited room to improve the trajectory, which is reflected in the modest quantitative gains (e.g.,

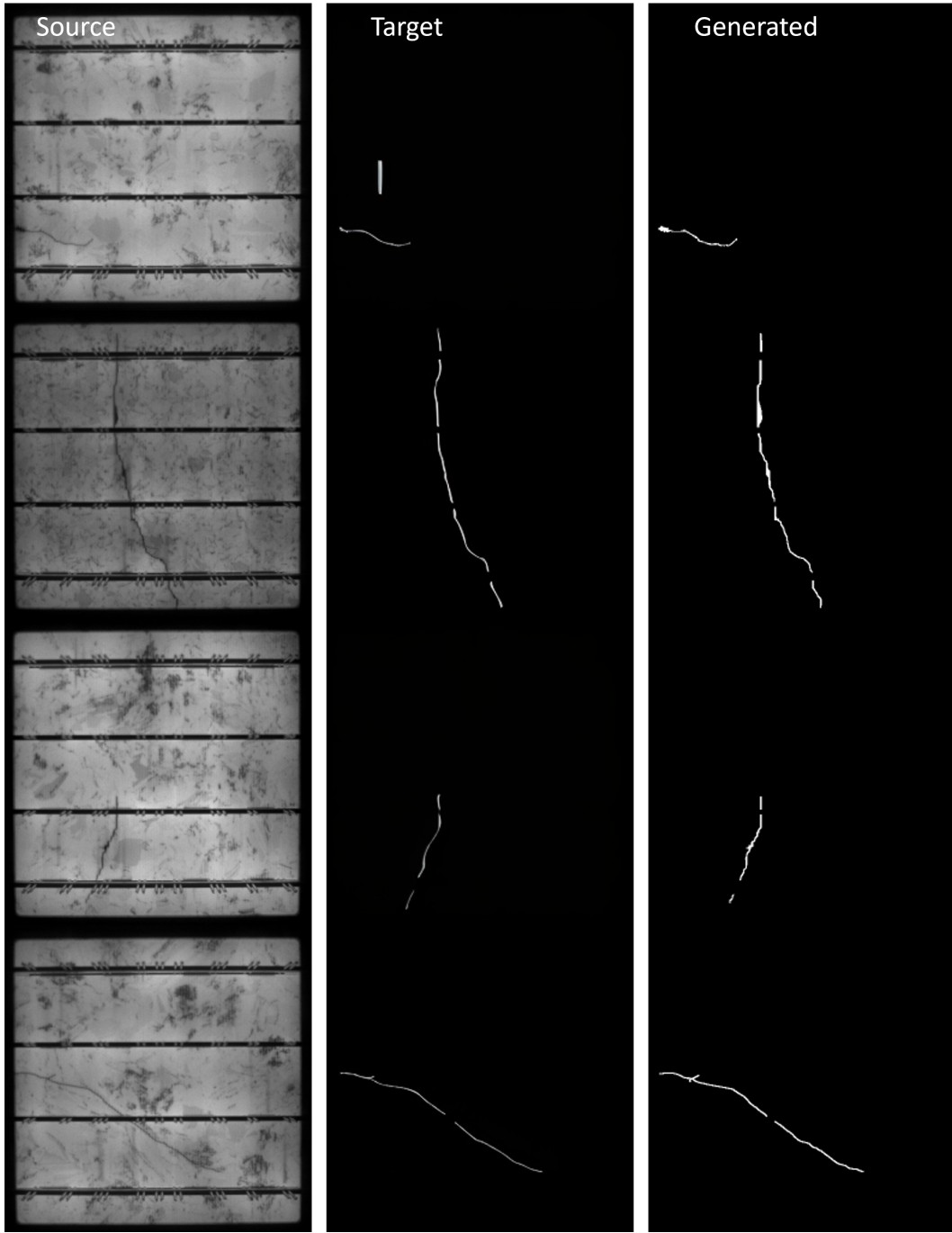

Figure 8: Visualization of the Electroluminescence to Semantic mask translation task; samples are randomly picked from the testing dataset.

small SSIM/PSNR improvements) and visually near-identical reconstructions for many slices. Empirically, we observe that the learned $\Lambda_t$ fields on IXI often stay close to the baseline linear schedule, effectively behaving as a gentle refinement rather than a qualitatively different path.

**Failure and limit cases:** We emphasize that CDTSDE is not a universally necessary replacement for fixed schedules. In particular, when the underlying task is too easy (low modality gap, clean acquisition, strong spatial alignment), the additional flexibility of a pixelwise, time-dependent sched-

ule may not translate into visible benefits. In our experiments on IXI, we do not observe systematic hallucination of new structures or topology breaks introduced by CDTSDE compared to the linear baseline. Instead, the typical limit case is that both methods already produce anatomically plausible reconstructions and CDTSDE only changes local contrast or edge sharpness slightly.

**When CDTSDE is most beneficial:** In contrast, on Sentinel and especially PSCDE, the modality gap and local heterogeneity are much larger: intensity statistics, textures, and semantic structures differ substantially between domains. In these settings, the assumptions underpinning a global linear schedule are violated, and the pixelwise dynamic schedule has substantial freedom to reduce path energy and avoid high-cost, off-manifold regions. This is reflected both in the larger improvements in structural and semantic metrics (e.g., Dice and Hausdorff distance on PSCDE) and in qualitative behavior, where CDTSDE better preserves fine structures and boundaries. These observations are consistent with the theoretical result that the pixelwise schedule strictly dominates any global schedule under heterogeneous local geometry and nondegenerate contrast, and suggest that CDTSDE is most useful when cross-modal translation is genuinely hard and spatially non-uniform.

## 6.18 FUTURE WORK

**Quantifying translation difficulty and adaptive deployment:** A natural next step is to define quantitative measures of cross-modal translation difficulty and relate them to the observed benefit of CDTSDE. Simple statistics such as cross-domain histogram distances, mutual information, or structural similarity under a naive baseline could serve as coarse indicators of heterogeneity. This would support adaptive deployment, where a system uses inexpensive pre-analysis to decide whether the added complexity of CDTSDE is warranted or a simpler linear schedule suffices.

**Extensions and future work** CDTSDE can be extended in several directions. Combining the dynamic schedule with flow-matching (Lipman et al. (2023)) or probability-flow (Wang et al. (2025)) ODEs could enable distillation into very few-step or even single-step generators while retaining the geometry-aware path. Additional regularization on $\Lambda_t$ (e.g., spatial smoothness or curvature penalties) may further guard against pathological local mixing. Multi-scale or region-aware schedules could allocate more flexibility to challenging regions(Xu et al. (2025)). Finally, explicit diagnostics to detect when CDTSDE offers little advantage would help practitioners choose between dynamic and linear schedules without extensive tuning.

## 6.19 FAILURE MODE ANALYSIS

Fig. 10 presents several representative counterexamples highlighting regimes where CDTSDE does not yield clear benefits and may introduce minor artifacts. We collect those from the PSCDE benchmark. The PSCDE exhibits strong nonlinear modality gaps and complex defect topology. As circled (in yellow) in the figure, CDTSDE occasionally produces small hallucinated streaks or faint boundary irregularities when the local cross-domain geometry is highly ambiguous. These issues typically arise near thin cracks, low-contrast defect boundaries, or regions with strong texture–structure interference. Similar failure cases are difficult to obtain for the IXI dataset because the T1/T2 contrast shift is mild and structurally well-aligned, leaving little room for dynamic scheduling to deviate from the global baseline. These counterexamples help clarify that CDTSDE is most beneficial in high-gap, heterogeneous translation regimes, and its advantage diminishes as the domain shift becomes simpler.

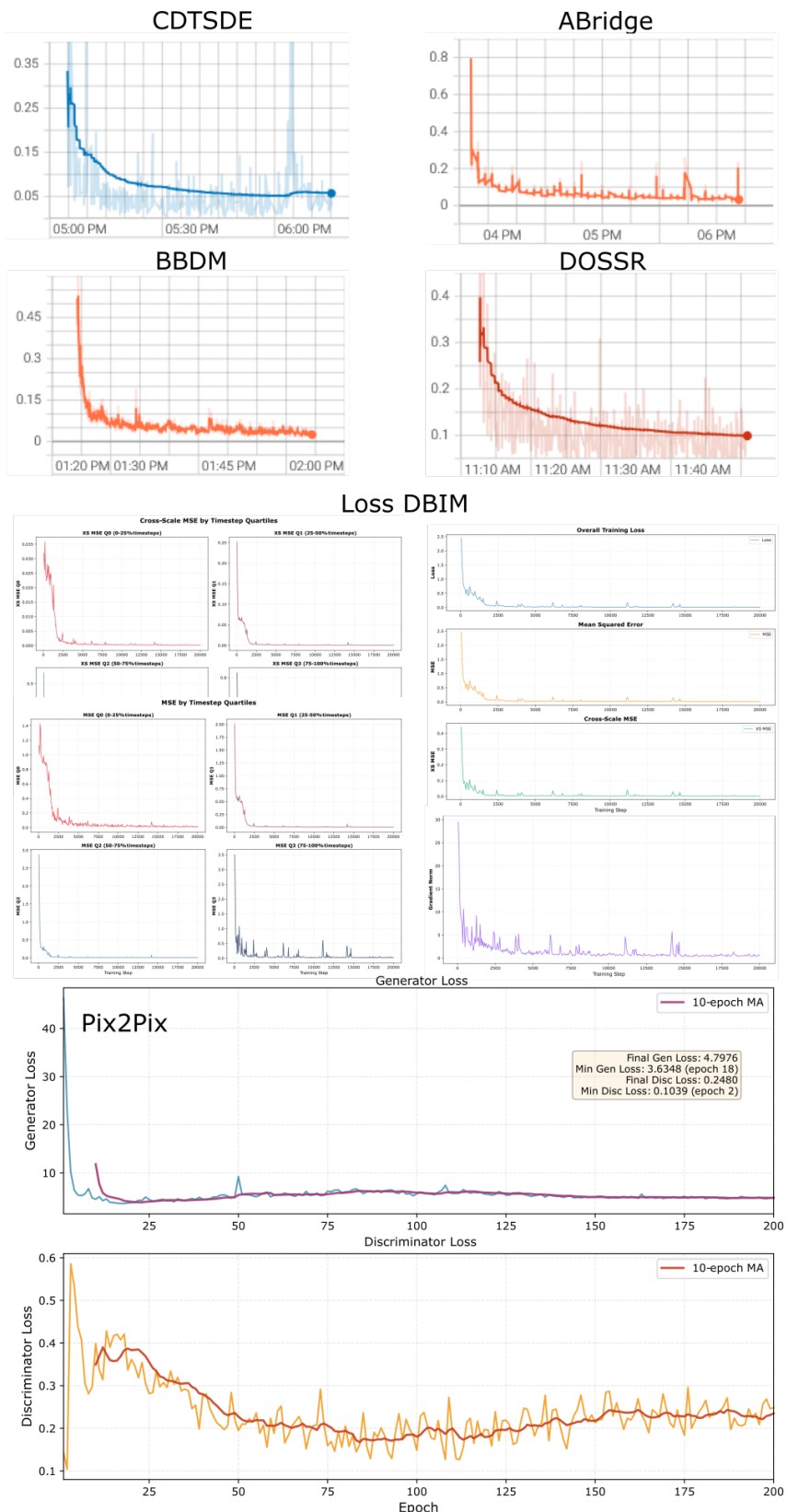

Figure 9: Loss curves for the CDTSDE and baselines: ABridge, BBDM, DoSSR, DBIM, Pix2Pix.

Table 4: Summary of Notation.

| Symbol | Description |
|--------|-------------|
| $\mathcal{X}_{\mathrm{src}}, \mathcal{X}_{\mathrm{tar}}$ | Source and target image spaces (e.g., SAR vs. OPT, or different MR/CT contrasts). |
| $(\hat{\boldsymbol{x}}_0^{\mathrm{src}}, \boldsymbol{x}_0)$ | Paired training sample, with $\hat{\boldsymbol{x}}_0^{\mathrm{src}} \in \mathcal{X}_{\mathrm{src}}$ the observed source image and $\boldsymbol{x}_0 \in \mathcal{X}_{\mathrm{tar}}$ the ground-truth target image. |
| $\hat{\boldsymbol{x}}_0^{\mathrm{gen}}$ | Generated target-domain image produced by the model at $t = 0$. |
| $\boldsymbol{x}_t$ | Diffusion state at time $t$ in the target domain; $t$ can be continuous ($t \in [0,1]$) or discrete ($t \in \{0, \ldots, T\}$). |
| $t, s; T$ | Diffusion time indices and the total number of diffusion steps $T$. |
| $H, W, C$ | Image height, width, and number of channels, respectively. |
| $c \in \{1, \ldots, C\}, \mathbf{p} \in \{1, \ldots, H\} \times \{1, \ldots, W\}$ | Channel and spatial (pixel) indices. |
| $\beta_t$ | Variance schedule at step $t$ in the variance-preserving (VP) diffusion process. |
| $\alpha_t, \bar{\alpha}_t$ | Per-step and cumulative signal coefficients in the forward diffusion; typically $\bar{\alpha}_t = \prod_{i=1}^t \alpha_i$. |
| $\sigma_t^2 = 1 - \bar{\alpha}_t$ | Noise variance at time $t$ in the VP diffusion. |
| $\Lambda_t \in (0,1)^{C \times H \times W}$ | Domain-mixing field at time $t$. |
| $\Lambda_{t,c}(\mathbf{p})$ | Scalar mixing weight at channel $c$ and spatial location $\mathbf{p}$ (element of $\Lambda_t$). |
| $\boldsymbol{d}_t$ | Domain mixture $\boldsymbol{d}_t = \Lambda_t \odot \hat{\boldsymbol{x}}_0^{\mathrm{src}} + (\mathbf{1} - \Lambda_t) \odot \boldsymbol{x}_0$, used as guidance for cross-domain translation. |
| $\odot, \oslash$ | Hadamard product and division, respectively. |
| $\boldsymbol{\Upsilon}_t = \sqrt{\bar{\alpha}_t}(\mathbf{1} - \Lambda_t)$ | Effective signal scale. |
| $\boldsymbol{\lambda}_t = \sigma_t \oslash \boldsymbol{\Upsilon}_t$ | Local mixing–to–noise ratio. |
| $\boldsymbol{y}_t = \boldsymbol{x}_t \oslash \boldsymbol{\Upsilon}_t$ | Rescaled diffusion state expressed in $\boldsymbol{\lambda}$-coordinates. |
| $d\boldsymbol{w}_{\boldsymbol{\lambda}}$ | Wiener increment. |
| $\boldsymbol{\varepsilon}_\theta(\boldsymbol{x}_t, \hat{\boldsymbol{x}}_0^{\mathrm{src}}, t)$ | Conditional noise-prediction network (score model) parameterized by $\theta$. |
| $\boldsymbol{h}_t(\mathbf{p})$ | Learned spatial gating map produced by the schedule network at time $t$. |
| $\boldsymbol{\pi}(\mathbf{p}) \in \mathbb{R}^4$ | Positional encoding of spatial index $\mathbf{p}$. |
| $a_{c,\mathbf{p}}(t) > 0$ | Local metric on the trajectory velocity at channel $c$, location $\mathbf{p}$, and time $t$. |
| $U_{c,\mathbf{p}}(d_{c,\mathbf{p}}(t), t)$ | Strictly convex local potential acting on the mixed-domain trajectory value $d_{c,\mathbf{p}}(t)$ at time $t$. |
| $\mathcal{E}[\boldsymbol{d}]$ | Path energy functional of the domain-mixing trajectory $\boldsymbol{d}(t)$, integrating local kinetic and potential terms over $t \in [0,1]$. |

---

**Algorithm 2** Score-Matching

---

**Require:** paired dataset $\mathcal{D}$ with targets $x_0$ and source observations $\hat{x}_0^{\text{src}}$; variance-preserving schedule $\{\alpha_t\}_{t=0}^{T-1}$; mixer $\mathcal{N}_\phi$; score network $\Phi_\theta$

1: **repeat**
2:     **Mini-batch draw:** $\mathcal{B} = \{(x_0^{(i)}, \hat{x}_{0,\text{src}}^{(i)}, y^{(i)})\}_{i=1}^B \subset \mathcal{D}$
3:     **Random timestep:** $t_i \sim \text{Uniform}\{0, \ldots, T-1\}$ independently for each $i$
4:     **Encode conditions:** $c_{\text{latent}}^{(i)} \leftarrow \text{Enc}_{\text{lat}}(\hat{x}_{0,\text{src}}^{(i)})$
5:     **VP coefficients:** $\bar{\alpha}_{t_i} \leftarrow \prod_{s=0}^{t_i} \alpha_s, \quad \sigma_{t_i} \leftarrow \sqrt{1 - \bar{\alpha}_{t_i}}, \quad \Upsilon_{t_i} \leftarrow \sqrt{\bar{\alpha}_{t_i}}(1 - \Lambda_{t_i})$
6:     **Mixing field:** $\Lambda_{t_i}^{(i)} \leftarrow \mathcal{N}_\phi(\Lambda_{t_i}^{\text{base}}, C, H, W)$
7:     **Domain mixture:** $d_{t_i}^{(i)} \leftarrow \Lambda_{t_i}^{(i)} \odot \hat{x}_{0,\text{src}}^{(i)} + (1 - \Lambda_{t_i}^{(i)}) \odot x_0^{(i)}$
8:     **Forward diffusion:** draw $\epsilon^{(i)} \sim \mathcal{N}(0, I)$ and set $x_{t_i}^{(i)} \leftarrow \sqrt{\bar{\alpha}_{t_i}}\, d_{t_i}^{(i)} + \sigma_{t_i}\, \epsilon^{(i)}$
9:     **Score prediction:** $\hat{\epsilon}_\theta^{(i)} \leftarrow \Phi_\theta(x_{t_i}^{(i)}, t_i, c_{\text{latent}}^{(i)})$
10:    **Score-matching loss:** $L \leftarrow \frac{1}{B} \sum_{i=1}^B \|\hat{\epsilon}_\theta^{(i)} - \epsilon^{(i)}\|_2^2$
11:    **Parameter update:** $\theta \leftarrow \theta - \eta \nabla_\theta L, \quad \phi \leftarrow \phi - \eta_\phi \nabla_\phi L$
12: **until** convergence

---

**Algorithm 3** First-Order Sampler

---

**Require:** steps $N$, latent shape $s$, condition image $x_{\text{cond}}$

1: **Build spaced schedule:** $\{(\beta_t, \Lambda_t, \Upsilon_t, \lambda_t)\}_{t \in \mathcal{I}}, \; \mathcal{I} \leftarrow \text{MAKESCHEDULE}(N)$
2: **Encode source latent:** $x_0^{\text{src}} \leftarrow \text{ENCODEFIRSTSTAGE}(x_{\text{cond}})$
3: **Sample initial state:** Draw $\epsilon \sim \mathcal{N}(0, I)$ and set $x_T \leftarrow \Upsilon_{t_{\max}} x_0^{\text{src}} + \sigma_{t_{\max}} \epsilon$
4: **Reverse loop over indices:**
5: **for** successive $(s, t)$ in $\mathcal{I}$ (descending) **do**
6:     **Noise prediction:** $e_s \leftarrow \text{PREDICTNOISE}(x_s, s)$
7:     **Predicted clean sample:** $\hat{x}_0^{(s)} \leftarrow \bar{\alpha}_s^{-1/2} x_s - \sqrt{\bar{\alpha}_s^{-1} - 1}\, e_s$
8:     **Ratios:** $r_\lambda \leftarrow \dfrac{\lambda_t}{\lambda_s}, \quad r_\Upsilon \leftarrow \dfrac{\Upsilon_t}{\Upsilon_s}$
9:     **State transport:** $x^{\text{transport}} \leftarrow r_\Upsilon\, r_\lambda^2 \cdot x_s$
10:    **Domain-shift drift:** $x^{\text{drift}} \leftarrow \Upsilon_t \cdot \left[ \dfrac{\Lambda_t}{1 - \Lambda_t} - \dfrac{\Lambda_s}{1 - \Lambda_s} \cdot r_\lambda^2 \right] \cdot x_0^{\text{src}}$
11:    **Network-guided term:** $x^{\text{denoise}} \leftarrow \Upsilon_t \cdot (1 - r_\lambda^2) \cdot \hat{x}_0^{(s)}$
12:    **Stochastic variance:** $\sigma_{\text{hat}}^2 \leftarrow \lambda_t^2 (1 - r_\lambda^2)$
13:    **Draw noise:** $z \sim \mathcal{N}(0, I), \quad x^{\text{noise}} \leftarrow \Upsilon_t \cdot \sigma_{\text{hat}} \cdot z$
14:    **State update:** $x_t \leftarrow x^{\text{transport}} + x^{\text{drift}} + x^{\text{denoise}} + x^{\text{noise}}$
15: **end for**
16: **Decode to image space: return** $(\text{DECODEFIRSTSTAGE}(x_0) + 1)/2$

---

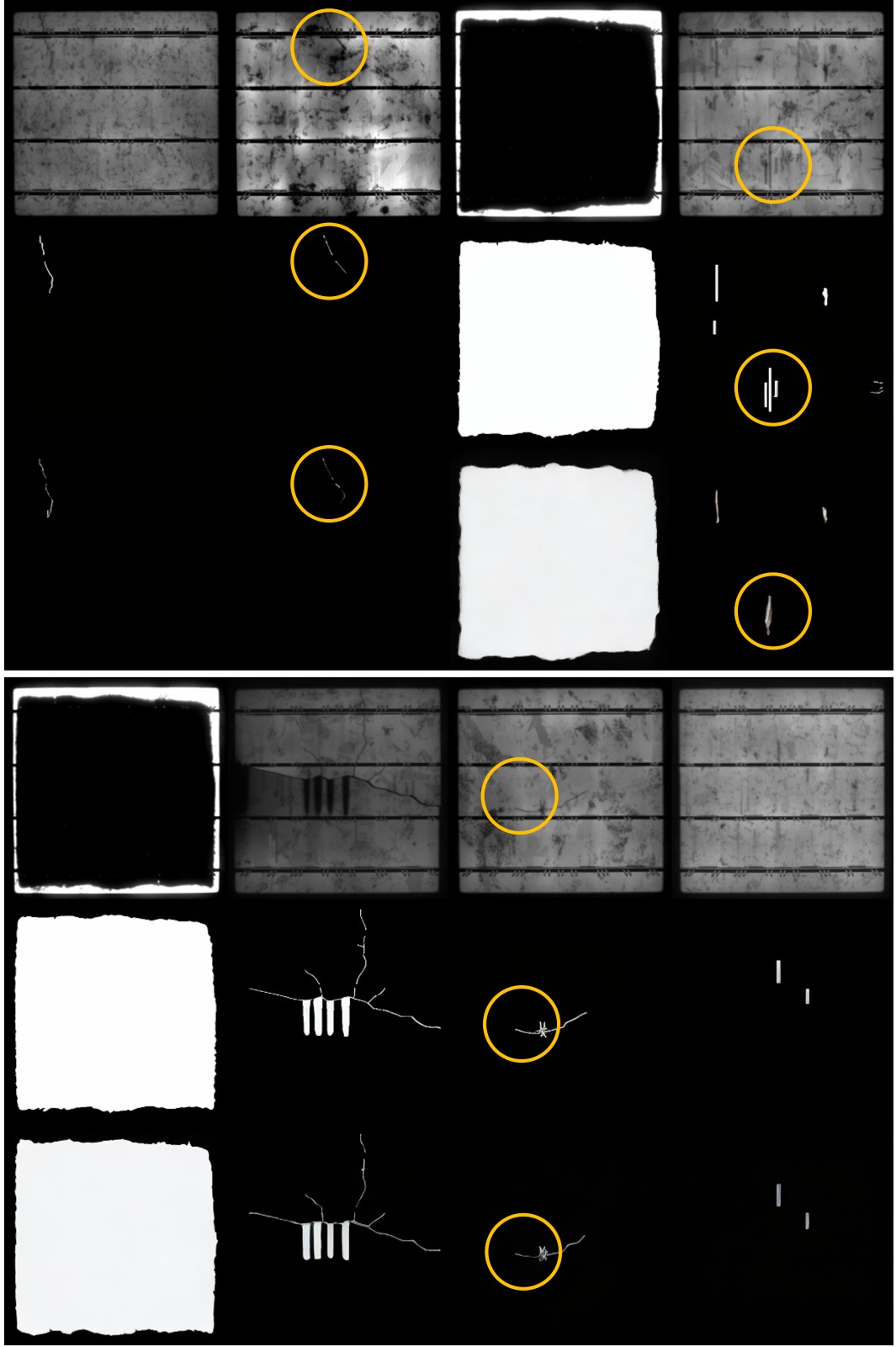

Figure 10: Counterexamples for failure mode analysis.

