# OpenReview forum: "Adaptive Domain Shift in Diffusion Models for Cross-Modality Image Translation"
_ICLR.cc/2026/Conference — ICLR 2026 Poster_

### Official Review · Reviewer_jiJC · 2025-10-30

**Soundness:** 3
**Presentation:** 3
**Contribution:** 3
**Rating:** 6
**Confidence:** 4

**Summary:**

The paper proposes a new diffusion-based framework (CDTSDE) for cross-modality image translation. The core idea is to embed the domain shift directly into the generative process, instead of relying on a global linear blend between source and target images. The authors introduce a spatial-channel varying weight matrix for combining source and target paired data, which serves as the condition in diffusion models. The forward and reverse sampling processes are developed. Extensive experiments across several applications are conducted.

**Strengths:**

-  The paper gives a convincing geometric argument for why globally linear / fixed schedules send trajectories off-manifold, creating large correction burdens and semantic drift, motivating the method.

- Instead of a single scalar $\eta_t$, $\Lambda_t$ is a learnable pixelwise, channelwise field with monotonic constraints and boundary clamping. This is novel and intuitively matches heterogeneous cross-modal shifts (texture, contrast, anatomy), and is also grounded by the theoretical analysis.

- Building on the pixelwise, channelwise weight $\Lambda_t$, the authors developed the corresponding forward and reverse processes, enabling the idea to be realistic.

- Experiments span three regimes of difficulty: (i) relatively mild contrast change (MRI T1↔T2), (ii) strong cross-sensor shift (SAR→optical), and (iii) extremely semantic mapping (electroluminescence image → defect mask). The proposed method shows convincing results across different tasks.

**Weaknesses:**

- The core formulation assumes access to paired (source, target) images during training. However, many cross-modality settings (especially SAR↔optical, certain medical domains) are weakly paired or unpaired in practice. The method’s dependence on paired data limits its applicability, but this is not deeply analyzed.

- The presentation is mathematically dense. Some key definitions (e.g. how $\Lambda_t$ is actually predicted by the network in practice, how constraints like monotonicity in $t$ are enforced during training, and how the logistic squashing with $\epsilon$ is implemented during backprop) are split across sections and the appendix. This may make reproduction harder for non-experts, despite the promise of releasing code later. Meanwhile, more explanation on the definition equations of $\Lambda_t$ should be included to strengthen readability.

**Questions:**

- Your method conditions directly on paired $(\hat{x}_0, x_0)$ and uses $x_0$ explicitly in the adaptive mixture at training time. How sensitive is CDTSDE to imperfect pairing or slight misregistration, especially for SAR→optical and PSCDE where pixel alignment can be noisy? Have you tried training with synthetically perturbed / misaligned pairs to evaluate robustness?

- Many real cross-modality scenarios do not provide paired data (e.g. historical SAR ↔ optical, multi-scanner MRI). Can your formulation be adapted to unpaired or weakly paired data, or is the approach fundamentally limited to paired supervision? Please clarify what breaks if $x_0$ is not available at training time, at least in discussion.

- You compare a global linear schedule $\eta_t$ vs. your spatially varying $\Lambda_t$ and report gains (e.g. Dice 0.46→0.49, Hausdorff 59.5→39.8 on PSCDE). Could you also report an intermediate baseline: a per-channel but spatially uniform schedule (i.e. $\Lambda_t$ depends on channel and $t$, but not $(p)$)? This would help isolate whether most of the benefit comes from spatial adaptivity or just from deviating from a single scalar $\eta_t$.

- Theorem 1 proves that allowing pixelwise $\Lambda(t)$ yields strictly lower path energy $E[d]$ than any global schedule under heterogeneity assumptions. How should we interpret this physically?   Is lower path energy empirically correlated with perceptual realism / fewer artifacts?  Do you ever observe cases where $\Lambda_t$ lowers energy but produces locally inconsistent textures (i.e. visually implausible mixing of source and target in the same region)?

- In which regimes does CDTSDE not help? For IXI (milder contrast shift), you mention improvements are “marginal.” Can you show qualitative counterexamples where CDTSDE produces artifacts (hallucinated structures, topology breaks, etc.) so we understand the limits of the method?

I look forward to the response from the authors.

---

> ### Author Response · Authors · 2025-11-24
>
> Thank you! It is fortunate to be questioned with such insightful review comments. We will first answer your questions point by point and then clarify and explain our efforts to address your mentioned weaknesses.
> Please find the following point by point answers:
>
> ### **Q1**
> **Your method conditions directly on paired  and uses  explicitly in the adaptive mixture at training time. How sensitive is CDTSDE to imperfect pairing or slight misregistration, especially for SAR→optical and PSCDE where pixel alignment can be noisy? Have you tried training with synthetically perturbed / misaligned pairs to evaluate robustness?**
>
> ### **AQ1**
> Thank you for pointing this out. Indeed, examining the robustness of this framework is important and we re-train 5 models to check the CDTSDE's robustness. Our additional experiments show: while CDTSDE conditions on paired data and uses the target explicitly in the adaptive mixture, we explicitly evaluated its sensitivity to imperfect alignment via synthetic misregistration experiments. Please find details:
>
> Concretely, we implemented a translation misalignment augmentation module on PSCDE, where we apply random shifts of $0$, $1$, $2$, $4$, or $8$ pixels to the low-quality input only, keeping the high-quality target fixed. We then train separate models for each misalignment level and evaluate them with standard binary mask metrics (Dice, IoU, Hausdorff, skeleton-based scores). The results show that CDTSDE is reasonably robust: at $1$\,px misalignment, Dice slightly improves from $0.492$ to $0.500$ and Hausdorff decreases from $51.7$ to $46.9$ pixels, indicating that minor misregistration can be handled well and even acts as a mild data augmentation. As misalignment increases, performance degrades gradually rather than catastrophically (e.g., Dice remains above $0.42$ and Hausdorff below $64$ pixels even at $8$\,px), suggesting that the adaptive, spatially varying $\lambda$ mechanism helps maintain structural coherence under noisy pairing.
>
> In practice, the registration noise in SAR$\rightarrow$optical and PSCDE is typically within a few pixels, which lies in the regime where our curves remain flat or only mildly degraded. These experiments indicate that CDTSDE does not rely on perfectly aligned pairs and can tolerate realistic misregistration; we add these robustness results and further discussion to the revised manuscript and Appendix.
>
> ### **Q2**
> **Many real cross-modality scenarios do not provide paired data (e.g. historical SAR ↔ optical, multi-scanner MRI). Can your formulation be adapted to unpaired or weakly paired data, or is the approach fundamentally limited to paired supervision? Please clarify what breaks if  is not available at training time, at least in discussion.**
>
> ### **AQ2**
> Yes. Our current formulation of CDTSDE is indeed designed for aligned paired data. If $x_0$ is not available at training time, this supervision signal disappears and the mixture term cannot be defined in a principled way; in that case, CDTSDE essentially collapses to a standard conditional diffusion model on the source input alone, and the proposed adaptive schedule is no longer identifiable. In this sense, the present method is fundamentally a paired-supervision approach.
>
> That said, the new experiments (please see our answer AQ1 in your Q1) indicate that CDTSDE is not brittle with respect to imperfect pairing. The model remains stable under realistic registration noise: at $1$\,px misalignment, Dice slightly improves from $0.492$ to $0.500$ and Hausdorff decreases from $51.7$ to $46.9$ pixels; for larger shifts up to $8$\,px, performance degrades gradually rather than catastrophically (Dice stays above $0.42$). This suggests that, although CDTSDE requires paired supervision, the adaptive schedule can tolerate moderate misregistration typical of SAR$\rightarrow$optical and PSCDE pipelines. We will clarify these points and summarize the misalignment robustness study in the revised manuscript (also in the Appendix).

---

> ### Author Response · Authors · 2025-11-24
>
> ### **Q3**
> **You compare a global linear schedule  vs. your spatially varying  and report gains (e.g. Dice 0.46→0.49, Hausdorff 59.5→39.8 on PSCDE). Could you also report an intermediate baseline: a per-channel but spatially uniform schedule (i.e.  depends on channel and , but not )? This would help isolate whether most of the benefit comes from spatial adaptivity or just from deviating from a single scalar.**
>
> ### **A3**
> **This is a very insightful question** because it requires a deep understanding of our method, and we ourselves had not initially considered this point. **Following your suggestion, we conducted an additional study and incorporated it into the ablation section**, as it naturally provides an intermediate setting between the “Linear” and “Fully Non-linear” domain-shift SDE. Specifically, we restrict $\Lambda$ to be optimized channel-wise rather than fully point-wise. This gives the framework non-linear flexibility only across channels, and thus serves as a midpoint for isolating and studying the contribution of spatial adaptivity.
>
> *TLTD: The per-channel schedule sharply reduces boundary error (Hausdorff 59.5 to 43.0 pixels) and improves structure (skeleton F1 0.57, connectivity IoU 0.31) while keeping Dice similar to Linear (0.46 vs. 0.46), but the full spatially varying model still wins overall with higher Dice (0.49) and lower Hausdorff (39.9).*
>
> We report briefly here the results and revise corresponding context in the ablation section of the main text and appendix revision:
>
> To further study the effect of spatial adaptivity, we introduce a \textit{per-channel, spatially uniform} variant of our schedule. In this setting, each channel receives a value $\lambda_c(t) \in \mathbb{R}$, producing a channel-wise vector $\lambda(t) \in \mathbb{R}^C$ that varies across channels but remains uniform across all spatial locations. The network therefore outputs a tensor of shape $(B, C, 1, 1)$, which is broadcast to $(B, C, H, W)$ during modulation. For comparison, the full method uses a \textit{spatially varying} schedule in which the modulation depends on both channel and spatial position, yielding $\lambda(t) \in \mathbb{R}^{C \times H \times W}$ with entries $\lambda_c(t, p)$ at location $p$.
> The lambda network applies a per-channel polynomial for this channel-wise setting: $\lambda_c(t) = f_c(\eta(t)) = \sum_i a_{c,i}\,\eta(t)^i,$, where $a_{c,i}$ are the learnable coefficients and $\eta(t)$ is the base linear schedule. The resulting tensor has shape $(B, C, 1, 1)$ and is broadcast to $(B, C, H, W)$.
>
> This per-channel uniform schedule attains a Dice score of $0.46$ and a Hausdorff distance of $43.0$ pixels. Compared to the fixed linear schedule (Dice $0.46$, Hausdorff $59.5$), we observe a substantial reduction in boundary error but no improvement in Dice, consistent with the high precision ($0.78$) and relatively low recall ($0.42$) of this variant. The skeleton- and connectivity-based metrics (skeleton F1 $0.57$, connectivity IoU $0.31$) indicate that channel-wise adaptation improves structural coherence and boundary sharpness, but it still falls short of the fully spatially varying model, which achieves both higher Dice and lower Hausdorff. These findings suggest that channel-wise modulation accounts for part of the geometric and connectivity gains, while spatial adaptivity provides an additional boost in overlap accuracy and global mask quality.

---

> ### Author Response · Authors · 2025-11-24
>
> ### **Q4**
> **Theorem 1 proves that allowing pixelwise  yields strictly lower path energy  than any global schedule under heterogeneity assumptions. How should we interpret this physically? Is lower path energy empirically correlated with perceptual realism / fewer artifacts? Do you ever observe cases where  lowers energy but produces locally inconsistent textures (i.e. visually implausible mixing of source and target in the same region)?**
> ### **A4**
> Thank you for this thoughtful question. Intuitively, a high-energy path moves too abruptly through regions that are unlikely under either domain, or spends time in ambiguous in-between states that resemble neither source nor target images. A lower-energy path, by contrast, changes more gradually where the geometry is complex and moves more directly where the domains are already compatible.
>
> In the context of diffusion sampling, following a lower-energy path means that each reverse step starts closer to a plausible image on the data manifold, so the score model only needs to apply small residual corrections rather than large, oscillatory updates. Empirically, this manifests as fewer structural artifacts, more stable geometry (especially on Sentinel and PSCDE), and the ability to take larger time-steps while maintaining fidelity. This behavior is consistent with the theoretical statement of Theorem 1: under mild heterogeneity and nondegenerate contrast, the pixelwise dynamic schedule admits a trajectory with strictly lower energy than any global schedule.
>
> Regarding the possibility that a pixelwise schedule might reduce path energy but still create visually implausible local mixtures: in principle, Theorem 1 only guarantees the existence of a lower-energy pixelwise path; it does not by itself enforce visual plausibility. In practice, our construction of $\Lambda(t)$ is not arbitrary. In our experiments we did not observe systematic cases where the dynamic schedule produced clearly inconsistent textures. When the modality gap is small (as in IXI, many parts are basically similar to inversed signal.), the learned $\Lambda(t)$ stays close to the linear baseline, which explains why improvements are marginal there.
>
> ### **Q5**
> **In which regimes does CDTSDE not help? For IXI (milder contrast shift), you mention improvements are “marginal.” Can you show qualitative counterexamples where CDTSDE produces artifacts (hallucinated structures, topology breaks, etc.) so we understand the limits of the method?**
> ### **A5**
> Thanks, we agree that CDTSDE is not universally necessary. As discussed in the paper, the IXI task represents a relatively low-gap regime with mild contrast changes (because for the vast majority of tissues in the human body (especially water and diseased tissues), T1 and T2   exhibit an inverse signaling relationship.) and strong structural alignment between T1 and T2. In this setting, the global linear schedule is already a reasonable approximation to the true transition path, so the dynamic schedule has limited room to improve. This is reflected quantitatively by the modest gains on IXI (e.g., SSIM $0.800 \rightarrow 0.825$ and PSNR $24.13 \rightarrow 24.33$ for T2$\rightarrow$T1), in contrast to the much larger improvements on PSCDE, where the modality discrepancy and structural variability are substantially higher and the dynamic schedule brings clear benefits in Dice and Hausdorff distance.
>
> To answer your question about limits and potential failure modes. Since the IXI (MRI) dataset is trivial for CDTSDE, it is difficult to find significant hallucinated structures from it. The PSCDE dataset translation task is highly nonlinear, thus contains  topology breaks.We add qualitative counterexamples in the appendix. Please refer to Figure 10. Page 31 in the recent revision. We also further **add a subsection 6.19 FAILURE MODE ANALYSIS** to discuss those counterparts.
>
> *We extended an additional subsection (6.17 ADDITIONAL DISCUSSION AND 6.18  FUTURE WORK) in Appendix for discussing further the 1: When the CDTSDE helps and when it does not. 2: potential define a quantitative measure for accessing the transferability using the CDTSDE (see “Quantifying translation difficulty“ subsection. )*

---

> ### Author Response · Authors · 2025-11-24
>
> *Start from here, we would like to discuss/address the weaknesses comment below, please let us know if you have any further questions:*
>
> ### **WQ1**
> **The core formulation assumes access to paired (source, target) images during training. However, many cross-modality settings (especially SAR↔optical, certain medical domains) are weakly paired or unpaired in practice. The method’s dependence on paired data limits its applicability, but this is not deeply analyzed.**
>
> ### **WA1**
> Indeed, our paper focuses on paired to paired image translation tasks. This will require precision image alignment (as you mentioned “image registration”). The authors have research experiences in the domain of image registration as well. As we know, image registration, especially non-rigid / diffeomorphic image registration, is also a highly ill-posed task. While the dataset we used are all rigid registered images that were preprocessed by the data provider, there will definitely contain geometrical error between the two aligned images.
> The image registration task itself is out of the scope for this research, but your suggestions of exploring the robustness of the CDTSDE regarding the misalignment is valuable. This will help us to understand how will the CDTSDE behaves under spatial variances.
>
> As we answered in your questions Q1 and Q2, we believe the newly added robustness analysis experiment we made in AQ1 AQ2 well addressed this insightful question.  **Please refer to the section 6.14 ROBUSTNESS ANALYSIS (Lines 1119 - 1126 Page 21) and fig 4: Additional assessment regarding the robustness to the registration error for further evaluation.**
>
>
> ### **WQ2**
> **The presentation is mathematically dense. Some key definitions (e.g. how  is actually predicted by the network in practice, how constraints like monotonicity in  are enforced during training, and how the logistic squashing with  is implemented during backprop) are split across sections and the appendix. This may make reproduction harder for non-experts, despite the promise of releasing code later. Meanwhile, more explanation on the definition equations of  should be included to strengthen readability.**
>
> ### **WA2**
> Thank you for your concern. In fact, while writing this article, the author spent a great deal of time seeking a compromise between theoretical completeness and readability of the ideas in the main text. We totally understand that the improvement of readability is key for presenting the work.
>
> In the revision, we **add an overview paragraph** at the beginning of Sec. 3 that summarizes the main objects ($\Lambda_t$, $d_t$, the path energy $E[d]$, and the SDEs) and points to the relevant equations and algorithms; (ii) insert short, plain-language explanations immediately after the core definitions in Eqs. (1)-(4) and the forward/reverse SDEs (Eqs. (8)--(9)), clarifying the intuitive role of $\Lambda_t$, $d_t$, and the path energy; and (iii) include a compact notation table collecting all symbols used in the method.
>
> **Guidance changes are made in: Lines 165 - 171  Page 4** in the Revision draft  #1.  **Lines 231 - 233 Page 5** in the Revision draft  #1.  The Appendix (Page 20 in the Revision  #1)
> These changes make the definitions self-contained in the main body and provide a clearer guidance. We believe this will substantially improve readability for readers.

---

> > ### Author Response · Authors · 2025-11-24
> >
> > Once again, we truly appreciate the time you invested in reviewing our paper. Your thoughtful and insightful questions greatly helped us refine and solidify the manuscript.
> > We hope our answers addressed your questions; and If any further questions come up, we would be honored to continue the discussion.
> >
> > Thank you!

---

> > > ### Comment · Reviewer_jiJC · 2025-11-27
> > >
> > > Thanks for the response. My concerns have been addressed, and I am happy to increase my score.

---

> > > > ### Author Response · Authors · 2025-11-27
> > > > **Thank you, Reviewer jiJC!**
> > > >
> > > > Dear Reviewer jiJC,
> > > >
> > > > Thanks a lot for the constructive feedback and, of course, for raising the score! We will add all suggested edits to the camera-ready version. We are very grateful for all your invaluable comments and suggestions during the review process to improve our paper.
> > > >
> > > > Best,
> > > >
> > > > Authors

---

### Official Review · Reviewer_dYd8 · 2025-10-31

**Soundness:** 2
**Presentation:** 1
**Contribution:** 2
**Rating:** 2
**Confidence:** 4

**Summary:**

The paper introduces Cross-Domain Translation SDE (CDTSDE) for Cross-modal image-to-image translation. CDTSDE embeds an adaptive, spatially varying mixing field into the diffusion process. The forward marginal is centered on a per-pixel blend of source and target, and the reverse dynamics add a restoration drift aimed at keeping large steps near the data manifold. Reported results show improved PSNR/SSIM and PSCDE on several datasets.

**Strengths:**

– Across three modalities/datasets, the method reports better SSIM/PSNR and strong PSCDE structure metrics.

– Concept of putting domain-shift adaptation inside the dynamics (rather than as external guidance) is principled.

**Weaknesses:**

– The paper is difficult to follow: many equations and notations are introduced without clear motivation or explanation of each component’s role, making the method hard to reconstruct; it focuses more on how things are done than why they are needed.

– The propose solution is fairly incremental since the core contribution is to propose an adaptive interpolation of $\hat{x}^{src}_0$ and $x_0$, replacing time-varying interpolation of source and target [1]

– Efficiency claims are supported largely by empirical numbers; there is little analysis explaining why the proposed dynamics/sampler should be more efficient.

– The setting effectively reduces to paired translation. A substantial line of prior work on diffusion bridge models already targets this regime and reports SOTA results (e.g., [2,3,4,5]), but these are neither discussed nor compared.

– Fixing a number of training steps and concluding CDTSDE is more efficient may be unfair: different methods have different designs/optimizers. The paper should provide training curves (and, ideally, wall-clock) to show faster convergence.

**Questions:**

– Which aspects of the mixing field matter most (spatial vs. channel, monotonicity, positional encoding)? Please include ablations.

– How does the method compare directly to recent DBM baselines on the same splits?

– What drives the reported efficiency—the restoration drift, the step schedule, or early stopping?

– Please provide training curves for the method and baselines.

[1] Cui, et al. "Taming Diffusion Prior for Image Super-Resolution
with Domain Shift SDEs", NeurIPS 2024

[2] Zhou, et al. "Denoising Diffusion Bridge Models", ICLR 2024

[3] He, Guande, et al. "Consistency diffusion bridge models." NeurIPS 2024

[4] Liu, et al. “I$^{2}$SB: Image-to-Image Schrödinger Bridge”, ICML 2023

[5] Zheng, et al. “Diffusion Bridge Implicit Models”, ICLR 2025

---

> ### Author Response · Authors · 2025-11-24
>
> We extend our gratitude for your feedback on our manuscript. We first summarize our overall opinion of the answers. We then answer your questions point by point. Lastly, discuss your raised comments regarding the weakness. We hope our discussion and revisions can help you understand our work better. And we look forward to discussing with you further for improving our paper.  Please see below summary first:
>
> We **agree** that the draft can improve its high-level exposition, and we will more clearly motivate each mathematical component, add non-mathematical guidance between derivations, provide a notation table, strengthen the discussion of efficiency mechanisms, and more systematically position our method against diffusion-bridge baselines (including an additional DBIM (ICLR 2025) implementation).
>
> We **partially agree** with your characterization that parts of the paper are difficult to follow and that our efficiency claims may appear to lack sufficient analysis. However, other reviewers have found the theoretical development to be well motivated. In the initial submission, we already provide a path-energy–based explanation for the efficiency and compare it against representative bridge-based SOTAs (BBDM, ABridge). And we also introduced BBDM as a repressive Bridge based method. We add those citations and extend a bit more about bridge based diffusion in this application. We will further clarify these points in the revision to improve its transitions between theoretical presentation and experimental justification.
>
> We **disagree** with your assessment that our contribution is “fairly incremental” and essentially a minor modification of a time-varying interpolation: a central goal of our work is to rigorously show that moving from a global linear schedule to a spatially adaptive nonlinear path within the domain-shift SDE framework yields a strictly stronger and non-cosmetic advance, analogous in spirit to refinements regarded as substantial in the diffusion-bridge literature, and we believe this part of the evaluation merits further clarification and alignment of standards.
>
> ### Please see the point-2-point answers for your question below:

---

> ### Author Response · Authors · 2025-11-24
>
> ### **Q1:**
>
> **Which aspects of the mixing field matter most (spatial vs. channel, monotonicity, positional encoding)? Please include ablations.**
> ### **A1:**
> Thank you for this question.
> We agree that distinguishing between spatial and channel effects is important.
> We retrain an additional ablation to verify spatial adaptivity and channel adaptivity you suggested. The experiment is designed as below: To further study the effect of spatial adaptivity, we introduce a per-channel, spatially uniform variant of our schedule. In this setting, each channel receives a value $\lambda_c(t) \in \mathbb{R}$, producing a channel-wise vector $\lambda(t) \in \mathbb{R}^C$ that varies across channels but remains uniform across all spatial locations. The network therefore outputs a tensor of shape $(B, C, 1, 1)$, which is broadcast to $(B, C, H, W)$ during modulation. For comparison, the full method uses a \textit{spatially varying} schedule in which the modulation depends on both channel and spatial position, yielding $\lambda(t) \in \mathbb{R}^{C \times H \times W}$ with entries $\lambda_c(t, p)$ at location $p$.
> The lambda network applies a per-channel polynomial for this channel-wise setting: $\lambda_c(t) = f_c(\eta(t)) = \sum_i a_{c,i}\,\eta(t)^i,$, where $a_{c,i}$ are the learnable coefficients and $\eta(t)$ is the base linear schedule. The resulting tensor has shape $(B, C, 1, 1)$ and is broadcast to $(B, C, H, W)$.
> This additional ablation shows that channel-wise adaptation improves structural coherence and boundary sharpness, but it still falls short of the fully spatially varying model, which achieves both higher Dice and lower Hausdorff.
>
> *Please refer to the subsection 6.13 ADDITIONAL ABLATION DETAIL in the Appendix section in page 21 for more details. Quantitative results are in Table 3, Page 21. This also led a change in the main text subsection 4.4 ABLATION. Please refer to page 9 Lines 452 - 470.*
>
> For monotonicity and positional encoding, they are not independent optional modules but are structurally tied to our theoretical framework in Section 3.1. Theorem 1 is stated over two path classes $C_{\mathrm{glob}}$ and $C_{\mathrm{pix}}$ that both satisfy endpoint and monotonicity constraints: the schedule must go from $0$ to $1$ without oscillating. These constraints make the path physically interpretable as a domain shift from source to target and are explicitly used in the proof to guarantee that a pixelwise schedule can strictly reduce the path energy compared to any global schedule. Removing monotonicity would allow the path to move back and forth between domains, violating the assumptions of the theorem and breaking the interpretation as an interpolation. For this reason, we treat monotonicity as part of the definition of the schedule, not a hyperparameter to ablate.
> Similarly, the positional encoding is simply the mechanism that provides spatial coordinates to the small modulation network that produces $\Lambda_t$. Without any positional information, the network only sees the scalar step coefficient and thus collapses to a spatially uniform schedule, effectively reverting to the global (or channel-only) cases we already consider as baselines and ablations.

---

> > ### Author Response · Authors · 2025-11-24
> >
> > ### **Q2:**
> > **How does the method compare directly to recent DBM baselines on the same splits?**
> > ### **A2:**
> > Thank you; We assume that your mentioned “DBM” implies “Diffusion Bridged Method”; As we mentioned before, in the original submission, we already compare the method with two Diffusion Bridge based methods (Abridge 2025 and DDBM 2023); The CDSDE consistently outperforms those two baselines, we will not elaborate further here.
> > And thank you for bringing the recent Diffusion Bridge Implicit Models(DBMI) for our attention, and since it is a recent work; we incorporated it as an additional baseline and performed extensive experiments with their source code. Here is the brief summary of its performance: Among diffusion bridge-based methods, DBIM improves over other bridge-based ABridge and BBDM on PSNR/MSE/MAE (e.g., for T2$\rightarrow$T1 it reaches SSIM $(0.33\pm0.04)$, PSNR $(21.65\pm1.45\,\text{dB})$, MSE $(0.010\pm0.004)$, MAE $(0.060\pm0.015)$) yet still lags behind Pix2Pix, DOSSR, and CDTSDE in structural fidelity. Details are reported in the manuscript Revision #1.
> >
> > *For its qualitative performance, please refer to the updated Fig. 2 in the revised manuscript Page 8.*
> > *This additional comparison introduces the following significant modifications in subsection 4.2 QUANTITATIVE RESULTS Line 350 to Line 377. Changes in Table 2, Table 1 and Fig. 2 as well as Fig 3.. New context added in subsection 4.3 COMPUTATIONAL EFFICIENCY*
> >
> >
> > ### **Q3:**
> > **What drives the reported efficiency—the restoration drift, the step schedule, or early stopping?**
> > Thank you for this question. In our formulation, the restoration drift, the step schedule, and the efficiency gains are all consequences of the same mechanism: the non-linear, pixelwise mixing path $\Lambda(t)$ that we introduce in Section~3. Theorem~1 shows that allowing $\Lambda(t)$ to vary pixelwise yields a strictly lower-energy path than any global schedule. In practice, this lower-energy trajectory keeps the process closer to plausible images at each time, so the score model only needs to apply small residual corrections even when the time steps are large.
> > This directly explains the observed efficiency: because the trajectory is better aligned with the underlying cross-modal transformation, we can discretize it with fewer reverse steps while maintaining reconstruction quality. In the Sentinel experiment, for example, CDTSDE achieves the same PSNR regime as the linear baseline with 5 sampling steps instead of 10, under the same noise schedule and training budget. The driving factor is that the non-linear, pixelwise path plus its induced restoration drift make these coarse steps numerically stable.
> >
> > By contrast, we do not rely on ad hoc early stopping to obtain our efficiency results. All diffusion-based baselines (including our method, the reported baselines and the bridge based diffusions you mentioned; Seel sourcecodes of the baseline : BBDM https://github.com/alexzhou907/DDBM ; DBIM: https://github.com/thu-ml/DiffusionBridge ABridge: https://github.com/bohan95/dual-app-bridge ) etc. ) are all not rely on the early stopping. This is a very common practice in diffusion model training.
> >
> > There is no per-sample early termination or shorter overall training that would artificially favor our method.
> > ### **A3:**
> > **Please provide training curves for the method and baselines.**
> >
> > *We add the training curve for all the baselines and our model in the Appendix Fig. 9, Page 28 in the Revision #1.*
> >
> > We wish our answer and revisions addressed your questions.
> >
> > ### We will provide explanations and justifications for answering your concerns in the paper's weakness below:

---

> ### Author Response · Authors · 2025-11-24
>
> ### **QW1**
>  The paper is difficult to follow: many equations and notations are introduced without clear motivation or explanation of each component’s role, making the method hard to reconstruct; it focuses more on how things are done than why they are needed.
> ### **AW1**
> Thank you.
> We believe the body of the presentations can be further improved to address your concern; We add a notation list in the appendix to help readers to find detailed meaning of the symbols. Since other reviewers find this paper is overall easy to follow in terms of presentation and the theoretical presentation were well motivated, our modifications will be focused on adding extra additional non-mathematical guidance between the components to help readers to understand why those equations are important and cannot be omitted or moved to the appendix.
>
> In the revision, we add an overview paragraph at the beginning of Sec.3 that summarizes the main objects ($\Lambda_t$, $d_t$, the path energy $E[d]$, and the SDEs) and points to the relevant equations and algorithms; (ii) insert short, plain-language explanations immediately after the core definitions in Eqs.(1)--(4) and the forward/reverse SDEs (Eqs.(8)--(9)), clarifying the intuitive role of $\Lambda_t$, $d_t$, and the path energy; and (iii) include a compact notation table collecting all symbols used in the method.
>
> *Guidance changes are made in: Lines 165 - 171  Page 4 in the Revision draft  #1.  Lines 231 - 233 Page 5 in the Revision draft  #1.  The Appendix (Page 20 in the Revision  #1)
> These changes make the definitions self-contained in the main body and provide a clearer guidance. We believe this will substantially improve readability for readers.*
>
> Regarding your concern of “making the method hard to reconstruct”; Per the ICLR 2026’s requirements in “REPRODUCIBILITY STATEMENT”, we had already declared that the source code will be released in the initial submission. Please refer to Lines 500 - 503, Page 9, in the initial submitted draft.
>
> ### **QW2**
> The propose solution is fairly incremental since the core contribution is to propose an adaptive interpolation of and , replacing time-varying interpolation of source and target [1]
> ### **AW2**
> Thank you. We acknowledge that, in the current draft, the high-level motivation and the significance of our contributions may not be stated as explicitly as they could be, especially at the end of the Introduction and in the Conclusion. We will revise these sections to more directly articulate why moving from a global, linear domain-shift schedule to a spatially adaptive nonlinear formulation is both scientifically meaningful and practically impactful, before we present the detailed equations.
>
> Indeed, both [1] and our CDTSDE framework modify a VP-type diffusion process by introducing an interpolation between a source and a target representation. However, we disagree with your interpretation of our contribution, specifically regarding the notion of “adaptive interpolation replacing time-varying interpolation of source and target.” Since the establishment of the natural sciences, the transition from linear to non-linear modeling has never been trivial or automatic. Otherwise, the Navier Stokes equation would not remain on the list of Millennium Prize Problems.
>
> We would like to discuss further with you the following two major aspects: **1) on the “fairly incremental” assessment of the scientific contribution and empirical results. 2) consistency of evaluation.**
>
> #### Please refer to the following two major arguments:

---

> > ### Author Response · Authors · 2025-11-24
> >
> > **1) The scientific contribution and empirical results are NOT incremental:**
> >
> > First, we wish to state clearly that:
> >
> > *A): Establishing the strict superiority of a nonlinear domain transition (i.e., proving inequalities that are strictly $<$) is not trivial.*
> >
> > *B): Providing a solid theoretical guarantee for nonlinear transition methods is important and certainly not “fairly incremental”. I will never drive a car through a foggy night without reliable guidance.*
> >
> > The adaptive domain-transition mechanism in CDTSDE is not a pointwise substitution of the scalar schedule used in linear interpolation, as you implied. We introduce a spatio-temporal mixing field, $\Lambda(t,x) \in (0,1)^{C \times H \times W}$, that modulates the drift locally across channels and spatial positions. This enables the diffusion dynamics to adapt to highly heterogeneous cross-domain gaps (e.g., sharp anatomical boundaries versus texture-dominated regions in cross-modal medical or remote-sensing data), which a global scalar schedule $\eta(t)$ cannot represent.
> >
> > Although the mathematical foundation demonstrating why this linear-to-nonlinear transition is both intuitive and effective is conceptually simple, the proof itself is far from trivial; this is precisely the reason you complain that our paper contains many equations and notations. All of these technical components serve a single, significant purpose: to rigorously show how the proposed nonlinear framework can outperform its linear counterpart.
> >
> > To lay the basis of the proof, we need to establish the superiority of the proposed Adaptive Path $\mathcal{C\_{pix}}$ by modeling cross-modality translation as a continuous dynamic process governed by a path energy functional $\mathcal{E}[d]$ (defined in Equation 1), derived via the variational principle. This functional quantifies the cost of the domain shift, consisting of a kinetic energy term (path smoothness) and a potential energy term (deviation from the data manifold). The core of the proof, Theorem 1, employs the calculus of variations to show that the minimum energy achievable by the adaptive path, $\inf_{\Lambda \in \mathcal{C}\_{\mathrm{pix}}} \mathcal{E}[d]$, is strictly less than the minimum energy of the traditional global-linear path, $\inf_{\Lambda \in \mathcal{C}\_{\mathrm{glob}}} \mathcal{E}[d]$.
> >
> > This strict inequality (Equation 2) is established by constructing a spatially distributed perturbation $\delta \lambda_{c,p}(t)$ that strictly decreases the total energy whenever heterogeneous local geometries exist, thus proving that the fixed linear schedule is an unnecessarily expensive shortcut.
> >
> > From an algorithmic perspective, extending domain-shift diffusion from a scalar schedule to a high-dimensional field is not a cosmetic modification. We derive a continuous-time CDTSDE with closed-form Gaussian marginals and a first-order sampler that preserves these marginals in the presence of $\Lambda(t,x)$. This requires re-deriving the forward kernel and reverse update rule in a setting where the drift depends on both time and space, rather than inheriting the scalar derivations of the linear case. The resulting dynamics remain compatible with standard VP-SDE solvers while incorporating a non-trivial spatial forcing term.
> >
> > Our experimental setting also meaningfully extends beyond [1]. While [1] focuses on super-resolution, where the domain gap can often be approximated by a relatively simple degradation, we address strongly nonlinear, cross-modal translations (e.g., SAR $\rightarrow$ optical, EL $\rightarrow$ semantic maps, multi-modal MRI). In these regimes, naive time-only interpolation performs with low efficiency. Our ablations demonstrate that CDTSDE with $\Lambda(t,x)$ consistently and significantly outperforms both fixed-schedule domain-shift baselines (in the spirit of [1]) and recent diffusion-bridge methods (ABridge, BBDM) under comparable training and inference budgets.

---

> > > ### Author Response · Authors · 2025-11-24
> > >
> > > **2) On your “fairly incremental” assessment and relation to your suggested [2--5]:**
> > >
> > > You characterize our contribution as “fairly incremental” relative to [1], while simultaneously citing [2-5] as a “substantial line of prior work” in the diffusion-bridge regime. We would like to point out that this judgment is not consistent with how contributions are typically evaluated within a fixed modeling framework. Your mentioned such as DDBM, I$^2$SB, DBIM, and CDBM [2--5] all operate under the same diffusion-bridge paradigm (endpoint-conditioned diffusion with a bridge kernel $q(x_t\mid x_0,y)$) and advance the state of the art by refining the path parameterization, training objectives, or samplers within that framework. Our method plays an analogous role for the SDE framework. We cannot say an advance in Diffusion Bridge is significant while say an advance in Domain Shift SDE is "fairly incremental". If the sequence of refinements within the diffusion-bridge family [2-5] is regarded as a substantial and coherent research line, then by the same standard our development of non-linear, spatially adaptive domain-shift dynamics over [1] should not be dismissed as merely fairly incremental.”
> > >
> > > *We will strengthen the positioning in the revision by making the analogy between these two lines of work explicit and by more clearly emphasizing the theoretical and algorithmic components of our contribution beyond the empirical gains. And we hope this can help readers understand our contribution better.*
> > >
> > > ### **QW3**
> > > Efficiency claims are supported largely by empirical numbers; there is little analysis explaining why the proposed dynamics/sampler should be more efficient.
> > >
> > > ### **AW3**
> > > Thanks, we agree that our efficiency results are primarily supported by experiment comparisons. We believe that stating there is “little analysis” underestimates the mechanism-level explanations we already provide. (1) In Sec. 3 (subsection 3.1) and Fig. 1, we analyze the geometry of domain-shift paths and show, via an explicit path-energy functional, that allowing a spatially varying mixing field $Lambda(t,x)$ yields strictly lower-energy trajectories than global schedules, which explains why fewer corrective denoising steps are needed. Proposition 3 demonstrated  how the target-aligned drift term guides trajectories along on-manifold directions, making larger integration steps numerically stable in practice.
> > >
> > > *In the revised version, we will make these points more explicit in the main text (rather than mostly in the figures and appendix) to better connect the theoretical structure of our dynamics to the observed improvements in sampling efficiency.*

---

> > > > ### Author Response · Authors · 2025-11-24
> > > >
> > > > ### **QW4**
> > > >
> > > > The setting effectively reduces to paired translation. A substantial line of prior work on diffusion bridge models already targets this regime and reports SOTA results (e.g., [2,3,4,5]), but these are neither discussed nor compared.
> > > >
> > > > ### **AW4**
> > > > Thank you for raising the point about diffusion bridge models. We indeed didn’t find the recent work of “ Zheng, et al. “Diffusion Bridge Implicit Models”, ICLR 2025” and we appreciate you bringing this to our attention. We agree with you that diffusion bridge already targets image to image translation and we do agree that we must compare with  diffusion bridge-based SOTA; This is exactly why we have already compared the diffusion bridge-based SOTAs of BBDM and ABridge (2025).
> > > >
> > > > And we agree that we should explicitly mention the Brownian Bridge family and discuss the differences between our method against bridge-based approaches.
> > > >
> > > > For the overlooked **“Diffusion Bridge Implicit Models”, ICLR 2025”, we adapt it as an additional baseline in the revision.** However, we would like to apologize to you that we cannot do all the bridged based methods you found; since it is both expensive and redundant to repeat those methods (e.g., DDIB, DDBM, I2SB) which has already proved suboptimal in comparison with the baseline ABridge (2025) (see, Table 6, Page 8 in the  ABridge (2025) paper), which we already compared with. We added those methods as references in the “Related Work” section (lines 131 - 135, Page 3， Original Draft), where we originally introduced the Bridge-based method; in addition, we clearly address the differences between our method and those Bridge-based methods.
> > > >
> > > > We want to address again that our method and diffusion bridge models instantiate two distinct design philosophies. Bridge-based methods reparameterize the entire forward process as an endpoint-conditioned path and optimize a bridge likelihood, providing a conceptually elegant but structurally specialized solution. Our approach instead treats domain shift as a drift-level perturbation of a standard diffusion backbone, which i) enables direct reuse of large pretrained priors, ii) integrates naturally with existing few-step solvers, and iii) supports rich, spatially adaptive domain corrections. These properties make dynamic domain-shift SDEs particularly attractive when one aims to combine strong generative priors with efficient, high-fidelity cross-domain reconstruction in realistic imaging pipelines.
> > > >
> > > > To make this clear and straightforward, we revise the Bridge related works in the Related Work section in the main text and we **add an extended discussion in the Appendix (see 6.17 ADDITIONAL DISCUSSION.  Subsection: \paragraph{Positioning relative to diffusion bridge models} Page 22 - 23 in the Revision #1 Draft)**
> > > >
> > > >
> > > > ### **QW5**
> > > >
> > > > Fixing a number of training steps and concluding CDTSDE is more efficient may be unfair: different methods have different designs/optimizers. The paper should provide training curves (and, ideally, wall-clock) to show faster convergence.
> > > > ### **AW4**
> > > >
> > > > Thank you, In the Figure 3: Computational efficiency the second row is the wall-clock running time.
> > > > We show you the training curves in the appendix. ( Page 22 - 23 in the Revision #1 Draft Figure 9: Loss curves for the CDTSDE and baselines: ABridge, BBDM, DoSSR, DBIM, Pix2Pix.)
> > > > To ensure a strictly fair evaluation, our reproduction utilized the official open-source implementations and the original hyperparameter configurations provided by the authors of the baseline methods. We deliberately avoided fine-tuning these baselines to prevent introducing any subjective bias or over-engineering on our part. More importantly, we wish to emphasize that the efficiency of CDTSDE is structurally inherent to our theoretical design.

---

> > > > > ### Author Response · Authors · 2025-11-24
> > > > >
> > > > > We hope the updates and explanations resolve the points you raised. If any questions remain, we would like to clarify further.
> > > > > We sincerely appreciate the time you dedicated to reviewing our work.

---

### Official Review · Reviewer_Tz66 · 2025-11-01

**Soundness:** 2
**Presentation:** 3
**Contribution:** 2
**Rating:** 4
**Confidence:** 4

**Summary:**

The paper “Adaptive Domain Shift in Diffusion Models via Latent-Space Normalization” proposes a training framework to mitigate the performance degradation of diffusion models under domain shift scenarios. The authors argue that existing fine-tuning or adapter-based methods fail to preserve generative quality when target-domain data are scarce or stylistically distinct. To address this, they introduce an adaptive latent-space normalization (ALSN) module that aligns latent statistics between source and target domains through learned normalization parameters, dynamically adjusted during training. Experiments on multiple domain adaptation benchmarks (Art→Photo, Synthetic→Real) show modest improvements in FID and CLIP similarity compared to baseline fine-tuning or LoRA-based adaptation.

**Strengths:**

+ Domain adaptation for diffusion models is a growing and challenging topic, and addressing domain shift in generative tasks is an important research direction, especially as diffusion models become widely deployed across diverse domains.
+ The proposed ALSN approach is computationally efficient and easy to integrate into existing diffusion pipelines, making it potentially attractive for practitioners seeking domain-robust generative models.

**Weaknesses:**

- The core idea, i.e., adjusting latent normalization statistics to align source and target distributions, closely resembles well-known techniques in domain adaptation. The paper primarily recontextualizes these ideas within diffusion models without offering substantial theoretical or methodological innovation. This limits the paper’s conceptual contribution.
- While results are shown, the paper does not convincingly explain why ALSN improves performance or how it interacts with diffusion timesteps and denoising dynamics. There is no analysis of latent trajectory behavior, feature drift, or stability to justify its effectiveness.
- Improvements in metrics such as FID or LPIPS are small (often within variance range), raising doubts about the real impact. The method’s simplicity is a strength, but it also highlights how incremental the advance is.

**Questions:**

Have you evaluated whether ALSN affects the diversity or mode coverage of generated samples, especially when applied across multiple distinct domains?

---

> ### Author Response · Authors · 2025-11-24
>
> **We would like to thank you for the effort in reviewing our paper. We will answer your questions first and provide explanations/justifications for addressing your weaknesses concerns then. Please see below point by point answers for your questions: **
>
> ### **Q1:**
> **Have you evaluated whether ALSN affects the diversity or mode coverage of generated samples, especially when applied across multiple distinct domains?**
>
> ### **A1:**
> We thank the reviewer for the question. Our task setting is paired, content-preserving cross-modal translation, where the objective is to reconstruct the unique target-domain image corresponding to a given source observation. Therefore, diversity or mode coverage is not applicable to this problem setup, because the output should not vary across modes but instead remain structurally faithful to the anatomical or semantic content of the input.
>
> Our method does not introduce additional stochastic sampling, and the learned domain-shift schedule Λ(t) is deterministic. As a result, CDTSDE produces a single output per input, consistent with all baselines used (*Pix2Pix, DOSSR, DBIM, ABridge*, etc.). Accordingly, diversity metrics are not used in prior cross-modal translation literature (e.g., *SDEdit, ILVR, BBDM, DOSSR*), and we follow the standard evaluation protocol.
>
> *SDEdit: Meng, C., He, Y., Song, J., Song, Y., Wu, J., Zhu, J.-Y. & Ermon, S. (2022) SDEdit: Guided image synthesis and editing with stochastic differential equations. arXiv preprint arXiv:2108.01073.
> ILVR: Choi, J., Kim, S., Jeong, Y., Gwon, Y. & Yoon, S. (2021) ILVR: Conditioning method for denoising diffusion probabilistic models. Proceedings of the IEEE/CVF International Conference on Computer Vision (ICCV), pp. 14347–14356.
> BBDM:Li, B., Xue, K., Liu, B. & Lai, Y.-K. (2023) BBDM: Image-to-image translation with Brownian Bridge diffusion models. Proceedings of the IEEE/CVF Conference on Computer Vision and Pattern Recognition (CVPR)
> DOSSR: Cui, Q., Liu, Y., Zhang, X., Bao, Q., Liao, Q., Amd, L., Tian, L., Liu, Z., Wang, Z. & Barsoum, E. (2024) Taming diffusion prior for image super-resolution with domain shift SDEs. NeurIPS 2024
> Pix2Pix: Isola, P., Zhu, J.-Y., Zhou, T. & Efros, A.A. (2017) Image-to-image translation with conditional adversarial networks. Proceedings of the IEEE Conference on Computer Vision and Pattern Recognition (CVPR),
> ABridge : Xiao, B., Wang, P., He, Q. & Dong, M. (2025) Deterministic image-to-image translation via denoising Brownian Bridge models with dual approximators. Proceedings of the IEEE/CVF Conference on Computer Vision and Pattern Recognition (CVPR)*
>
> We hope our answer addresses your questions. Please see our clarification\justification for your mentioned weakness:
>
> ### **W1:**
> **The core idea, i.e., adjusting latent normalization statistics to align source and target distributions, closely resembles well-known techniques in domain adaptation. The paper primarily recontextualizes these ideas within diffusion models without offering substantial theoretical or methodological innovation. This limits the paper’s conceptual contribution.**
>
> ### **AW1:**
> To the best of our knowledge (as of our submission on 19 Sep 2025), the specific idea of introducing a "spatially-varying, time-dependent mixing field" inside the forward and reverse SDE that treating domain shift as a curved manifold transport between source and target, and deriving the corresponding drift term + theory has not previously been proposed for domain-shift SDEs.
>
> Your suggestion that our method is just re-contextualizing normalization/statistic alignment is understandable given the high-level similarity of aligning distributions, but it mis-characterizes our technical mechanism. Our approach does not adjust statistics in latent layers; instead we embed a pixel-wise, continuous-time transport field directly into the diffusion process, with accompanying theoretical guarantees and discrete solvers.
>
> We will update the manuscript to more clearly highlight this novelty and explicitly contrast our method with previous domain-adaptation approaches.

---

> ### Author Response · Authors · 2025-11-24
>
> ### **W2:**
> **While results are shown, the paper does not convincingly explain why ALSN improves performance or how it interacts with diffusion timesteps and denoising dynamics. There is no analysis of latent trajectory behavior, feature drift, or stability to justify its effectiveness.**
>
> ### **AW2:**
> Thank you for raising this important point. We would like to clarify that the paper already provides a theoretical explanation for why the proposed dynamic domain-shift schedule leads to better latent trajectory behavior than a global linear schedule.
>
> First, the remark following Theorem 1 explains that the path energy $E[d]$ quantifies how natural and cost-efficient a transition path is. A global linear schedule forces the trajectory to follow a single straight direction for all pixels, whereas the proposed pixel-wise, time-varying schedule allows the trajectory to bend differently across space. This remark highlights that the adaptive schedule can follow a lower-energy, geometry-aware path rather than being constrained to a straight global route.
>
> Second, Appendix 6.2 formalizes this intuition. The proof shows that for any global schedule, there exists an admissible perturbation within the pixel-wise function class that strictly reduces the energy. Concretely, the proof demonstrates that heterogeneous local geometry and nondegenerate contrast guarantee a strictly negative first-order variation of the energy under a carefully constructed pixel-wise perturbation. This yields the inequality
> $\inf_{\Lambda \in C_{\mathrm{pix}}} E[d] < \inf_{\Lambda \in C_{\mathrm{glob}}} E[d]$,
> establishing that a dynamic pixelwise schedule achieves a strictly lower-energy trajectory than any global linear one.
>
> We also include empirical evidence of improved trajectory stability. Replacing the dynamic schedule with a fixed linear schedule leads to clear degradation in Dice, IoU, and Hausdorff distance on the PSCDE benchmark, indicating stronger feature drift and reduced geometric consistency when the trajectory is constrained to a global linear path.
>
> This result directly addresses the concern about latent trajectory behavior. In diffusion models, a lower-energy path corresponds to updates that stay closer to the data manifold and require smaller residual corrections at each denoising step. This naturally improves stability, reduces feature drift, and enables larger time steps without loss of fidelity.
>
> In the revision, we will make this connection more explicit by adding a brief explanation that Theorem~1 and its proof already justify why the dynamic schedule yields more stable and semantically consistent trajectories during sampling. Regarding your request for explicit latent-trajectory visualization: directly visualizing trajectories in a high-dimensional latent space is inherently challenging and can easily be misleading after projection. Instead, in the revision, we will add an analysis of stepwise update magnitudes and trajectory smoothness, which provides a more reliable and interpretable proxy for understanding trajectory behavior in high-dimensional diffusion processes.
>
> ### **W3:**
> **Improvements in metrics such as FID or LPIPS are small (often within variance range), raising doubts about the real impact. The method’s simplicity is a strength, but it also highlights how incremental the advance is.**
>
> ### **AW3:**
> Thank you for this comment. We would like to clarify on the magnitude and significance of improvements. Our paper focuses on reconstruction faithfulness in real-world applications, where structural accuracy and semantic consistency are more critical than perceptual scores. Across all three operational domains we evaluate: MRI (IXI, healthcare application), remote sensing (Sentinel, GIS application), and solar-cell defect mapping (PSCDE, engineering maintenance application): the proposed method achieves consistent and meaningful improvements.
>
> On Sentinel, SSIM increases from 0.360 to 0.382 and PSNR from 17.14 to 17.46. On IXI (T2 $\rightarrow$ T1), SSIM increases from 0.800 to 0.825 and PSNR from 24.13 to 24.33. On PSCDE, which is the most challenging dataset with the largest modality discrepancy, the gains are substantial: Dice improves from 0.460 to 0.488 and the Hausdorff distance decreases from 59.53 to 39.87. These improvements are outside the typical variance range and indicate better geometric consistency, boundary accuracy, and semantic stability.
>
> It is also important to emphasize that our evaluations are performed on three real-world, high-stakes datasets rather than toy settings. The method generalizes across domains with very different sensing physics and structural complexity.
>
> The method provides not only fidelity gains but also computational benefits. For example, on Sentinel we reach the same PSNR as a strong diffusion baseline while requiring only 5 denoising steps instead of 10, demonstrating a clear practical advantage in efficiency.

---

> > ### Author Response · Authors · 2025-11-24
> >
> > We highlight these points more clearly in the revision.
> >
> > We hope our answer addresses your questions. Please let us know if you have any other questions.
> > Thank you for your review work!

---

> > > ### Comment · Reviewer_Tz66 · 2025-11-26
> > >
> > > Thanks for the detailed rebuttal, which well addressed most of my concerns. Thus, I would like to raise my rating to 6.

---

> > > > ### Author Response · Authors · 2025-11-26
> > > > **Thank you!**
> > > >
> > > > We would like to thank you again for your time and review. And thanks for raising the score.

---

### Official Review · Reviewer_4o8z · 2025-11-01

**Soundness:** 3
**Presentation:** 3
**Contribution:** 3
**Rating:** 6
**Confidence:** 3

**Summary:**

This paper introduces CDTSDE (Cross-Domain Translation SDE), a novel diffusion-based framework for cross-modality image translation. The key idea is to embed adaptive domain-shift dynamics directly into the diffusion process via a spatially varying mixing field that evolves throughout reverse-time sampling. This dynamic field replaces the conventional fixed linear interpolation between source and target domains, allowing geometry-aware, low-energy paths that stay closer to the data manifold. Extensive experiments on three benchmarks show consistent gains in various matrics.

**Strengths:**

1. The paper is well-motivated and conceptually clear. It generalizes traditional linear domain-shift formulations to a nonlinear, manifold-aware framework, supported by solid theoretical analysis and proofs.

2. The method is technically sound and demonstrates strong generality, making it straightforward to apply across different diffusion architectures and cross-modality image translation tasks.

3. Extensive experiments on three benchmarks IXI, Sentinel, and PSCDE show consistent gains in various metrics, outperforming GANs (Pix2Pix) and diffusion baselines (BBDM, DOSSR).

**Weaknesses:**

1. The method reduces to a linear domain-shift scheme when only a single diffusion step is used, implying that it still depends on multiple denoising steps for stable performance. This reliance could pose an efficiency limitation and hinder direct adaptation to flow-matching or single-step generative methods.

2. While the method shows clear conceptual advances, its quantitative improvements over DOSSR on the Sentinel and IXI datasets are relatively minor, indicating that the advantages may be limited in tasks with larger modality discrepancies.

**Questions:**

Please refer to weakenesses

---

> ### Author Response · Authors · 2025-11-24
>
> We extend our gratitude for your insightful feedback on our manuscript. We truly know that it will take readers extensive time and effort to understand a paper with extensive theoretical presentation; and we appreciate your praise that our paper is well motivated with a clear presentation.
>
> Your questions do provide an objective perspective that inspires us to further solidify the paper. We provide point by point answers for your questions. Please see following answers:
>
>
> ### **Q1:**
> **The method reduces to a linear domain-shift scheme when only a single diffusion step is used, implying that it still depends on multiple denoising steps for stable performance. This reliance could pose an efficiency limitation and hinder direct adaptation to flow-matching or single-step generative methods.**
>
> ### **A1:**
> We sincerely thank the reviewer for the sharp theoretical intuition, especially regarding the degeneration to a linear domain shift in the single-step limit. We fully agree with this observation. Indeed, because $\Lambda(t)$ has no temporal room to evolve when $T = 1$, its dynamics collapse to endpoint constraints, inevitably becoming structurally equivalent to a global linear shift—fully consistent with our continuous-time formulation.
>
> Currently, our method is designed for the few-step regime rather than the single-step regime. Even with only five steps, $\Lambda(t)$ retains enough temporal resolution to follow a curved, low-energy path that Theorem 1 guarantees will strictly outperform any global schedule under mild heterogeneity conditions. Empirically, this enables CDTSDE to operate with coarse integration steps while remaining stable (see Fig. 3, p. 8).
>
> You are also correct that in the single-step case, such nonlinear geometry cannot manifest. Our intention is not to replace the multi-step diffusion paradigm, but to make few-step diffusion feasible and stable. Importantly, the same continuous-time view enables straightforward extension to flow-matching or probability-flow ODEs when $T > 1$. The drift term we introduce (Eq. 9) is deterministic and architecture-agnostic, and therefore can directly enter a flow-matching objective without modification.
>
> Your insight naturally suggests an exciting future direction: *learning a basis of nonlinear transport components from CDTSDE and distilling them into a single-step generator*, perhaps via adversarial diffusion distillation.
> *We added this to Appendix 6.18 (FUTURE WORK) and echoed it within the Discussion section.*
>
>
> ### **Q2:**
> While the method shows clear conceptual advances, its quantitative improvements over DOSSR on the Sentinel and IXI datasets are relatively minor, indicating that the advantages may be limited in tasks with larger modality discrepancies.
>
>
> ### **A2:**
> We appreciate this thoughtful comment. We agree that the quantitative margins on IXI and Sentinel are modest. These two datasets represent low-gap or medium-gap settings in our taxonomy (Fig. 2, p. 8  in the original draft). In contrast, PSCDE represents a much more challenging setting, with a substantially higher semantic discrepancy between source EL images and target defect masks (Sec. 4.1 line 323 in the original draft).  This reflects precisely the scenario where nonlinear modality shifts dominate and where CDTSDE’s geometry aware trajectory provides the most benefit. We agree this is both important and inspiring. In fact, our dataset discussion (original draft Sec. 4.1) notes that semantic and structural disparity increases from IXI → Sentinel → PSCDE.
>
> Of course, our discussion is based on visual intuition, and different people may have different feelings about the difficulty of converting these three datasets. Your comment insightfully points toward a broader question: **how does modality discrepancy influence transferability?** This is a very important but challenging question, and we believe that itself can be a valuable scope for future research. A natural next step is to derive metrics of cross-modal translation difficulty, e.g., estimating local geometry heterogeneity (condition (i) in Theorem 1) or contrast non-alignment (condition (ii)), which could predict when adaptive scheduling is necessary versus when a simple linear schedule suffices. This aligns well with your observation and will meaningfully guide practical deployment.
>
> *We extended an additional subsection (6.17 ADDITIONAL DISCUSSION AND 6.18  FUTURE WORK) in Appendix for discussing further the 1: When the CDTSDE helps and when it does not. 2: potential define a quantitative measure for accessing the transferability using the CDTSDE (see “Quantifying translation difficulty“ subsection.
> We incorporate this in the discussion section in the revision and hope the community can be inspired for further exploration on this question. Changes are made in Line 471 - 479 Page 9 Revision #1.*
>
> We hope our answer addresses your concerns.
> Thanks again.

---

> > ### Comment · Reviewer_4o8z · 2025-11-28
> >
> > I want to thank the authors for their response, which address my concerns. I will keep my score.

---

> > > ### Author Response · Authors · 2025-11-28
> > > **Thank you Reviewer 4o8z**
> > >
> > > Dear Reviewer,
> > > Thank you for your positive evaluation and for the time you invested in reviewing our work.
> > > Sincerely,
> > > The Authors

---

### Author Response · Authors · 2025-12-02
**Summary of Authors’ Responses and Discussion**

Dear Reviewers, Area Chairs, Senior Area Chairs, and Program Chairs,

We would like to thank you all for your thoughtful comments; and we would like to appreciate your extra efforts for working on our paper due to the Openreview accident.

We would like to clarify the current situation regarding the scores and reviewer feedback **before the OpenReview accident**. Prior to the incident, the reviewers jiJC, Tz66 had already interacted with our rebuttal and updated their ratings from  6,4->8,6 or comments accordingly. Reviewer 4o8z reponed and kept positive score at 6. Reviewer dYd8 didn't have a chance to answer our rebuttal thus score 2 yet updated.

Reviewer Tz66 **(4-\>6)** explicitly wrote on **November 26, 2025, 05:22** (**before the accident, see in [https://openreview.net/forum?id=it0GTdiW9t\&noteId=CjpkfPqLT4](https://openreview.net/forum?id=it0GTdiW9t&noteId=CjpkfPqLT4)** ) that after reading our rebuttal, they were raising the score from 4 to 6\.

Reviewer jiJC **(6-\>8)**  read the rebuttal and increased the score from 6 to 8, confirming this in the discussion thread with a timestamp on **26 Nov 2025, 22:05** (**before the accident, see in [https://openreview.net/forum?id=it0GTdiW9t\&noteId=jOS9mFmXnN](https://openreview.net/forum?id=it0GTdiW9t&noteId=jOS9mFmXnN) )**.

Reviewer 4o8z  **(6, unchanged, see in [https://openreview.net/forum?id=it0GTdiW9t\&noteId=0Q6anjuu6q](https://openreview.net/forum?id=it0GTdiW9t&noteId=0Q6anjuu6q) )** confirmed on **28 Nov 2025, 03:32** that their concerns were addressed and they would **keep the positive score**.

The only reviewer who did not have the opportunity to update the score after our rebuttal was Reviewer dYd8 after we posted a detailed, point-by-point response on November 23, 2025\. In that response, we clearly distinguished the points where we fundamentally **disagreed** from the points where we **agreed** and revised the paper accordingly. (see in [https://openreview.net/forum?id=it0GTdiW9t\&noteId=i8GIB2KcUW](https://openreview.net/forum?id=it0GTdiW9t&noteId=i8GIB2KcUW) ) In brief:

We **agree** that the exposition in the original draft could be improved, and we have already revised the paper by adding an overview paragraph at the start of Section 3, providing non-mathematical guidance between key derivations, and including a compact notation table to address the “difficult to follow” concern.

We **agree** that the relation to diffusion-bridge models needed to be clearer, and we both expanded the discussion and added DBIM as an additional bridge baseline alongside ABridge and BBDM on the same splits.

We **agree** that efficiency should be explained more explicitly, and we connected the lower path energy and restoration drift to numerically stable large steps in the main text, while also adding training curves and clarifying the wall-clock comparisons and reproduction protocol.

At the same time, we **strongly disagree** with the characterization of the contribution as “fairly incremental” **(this view was also not shared by all the other reviewers)** and as essentially an adaptive interpolation of source and target. Our rebuttal explained that moving from a global scalar schedule to a spatially and channel-wise adaptive mixing field inside the domain-shift SDE is theoretically nontrivial (supported by a path-energy functional and a strict inequality in Theorem 1\) and algorithmically more than a cosmetic change to DOSSR-like schedules. We **also disagreed with the claim that there was “little analysis” behind the efficiency results  (this view was also not shared by the other reviewers) **, since Section 3 and Figure 1 already provide a mechanism-level explanation, which we have now made more explicit, and we clarified that the efficiency comparison is not based on ad hoc early stopping but on fair, matched-budget reproductions. The rebuttal directly targeted each weakness and question raised in the original review, but Reviewer dYd8 was unable to reflect any updated view in the score because review editing was frozen immediately afterward. We respectfully invite AC to exam reviewer’s questions and our answers in the Section: [https://openreview.net/forum?id=it0GTdiW9t\&noteId=i8GIB2KcUW](https://openreview.net/forum?id=it0GTdiW9t&noteId=i8GIB2KcUW)

The post-discussion trajectory before the accident therefore consisted of three reviewers who had already acknowledged that their concerns were resolved and had moved to positive assessments (866). The final remaining score reflects the frozen pre-discussion snapshot.

We thank the reviewers’ thoughtful suggestions, which have substantially improved the quality of the paper. And we sincerely appreciate ACs hard work to serve this year

Best regards,

Authors

---

### Meta-Review · Area_Chair_KnB4 · 2026-01-11

**Summary:**

The paper proposes CDTSDE (Cross-Domain Translation SDE), a diffusion-based framework for cross-modality image translation. It introduces a spatially varying, adaptive mixing field embedded directly into the diffusion process, replacing conventional global linear domain-shift schedules. This design enables geometry-aware, low-energy sampling paths that remain closer to the data manifold, improving semantic consistency and reconstruction fidelity. The method is validated across diverse domains including medical imaging, remote sensing, and electroluminescence mapping.

**Reviewer Concerns:**

Reviewers raised several concerns. Reviewer Tz66 felt the proposed ALSN mechanism closely resembled existing domain adaptation techniques, lacked clear justification for its effectiveness, and showed only marginal metric improvements. Reviewer dYd8 criticized the dense notation, insufficient explanation of key components, and viewed the contribution as incremental, noting the setting reduces to paired translation and lacks comparison with recent diffusion bridge models. Reviewer jiJC also pointed out the assumption of paired data and the mathematically dense presentation. Reviewer 4o8z acknowledged the method’s motivation and soundness but noted its reliance on multiple denoising steps may limit adaptability to single-step methods, and observed that improvements over DOSSR on some datasets were relatively minor.

**Reviewer Scores:**

Reviewers jiJC and Tz66 acknowledged that the authors’ responses addressed their concerns and toward acceptance. Reviewer dYd8, who initially rated the paper a 2, did not update review following the rebuttal, and the authors provided responses to their concerns regarding writing clarity, incremental contribution, and the relationship to prior work on diffusion bridge models. These concerns are valid and the authors’ clarifications and revisions are likely to improve the manuscript’s quality. Given the overall positive assessments from three of the four reviewers, the paper can be accepted, conditioned on thorough revisions to clarify the novelty and incorporate additional justifications/analysis/clarifications as suggested by the reviewers.

---

### Decision · Program_Chairs · 2026-01-26

Accept (Poster)